# AUTODA-TIMESERIES: AUTOMATED DATA AUGMENTATION FOR TIME SERIES

**Zijun Dou**[1], **Zhenhe Yao**[1], **Zhe Xie**[1], **Xidao Wen**[2], **Tong Xiao**[1,*] **Dan Pei**[1]
[1]Tsinghua University, Beijing, China
[2]Alibaba Cloud Computing Company, Hangzhou, China
`douzj25@mails.tsinghua.edu.cn`

 GitHub: `https://github.com/NetManAIOps/AutoDA-Timeseries`
 Project: `https://netmanaiops.github.io/AutoDA-Timeseries`

## ABSTRACT

Data augmentation is a fundamental technique in deep learning, widely applied in both representation learning and automated data augmentation (AutoDA). In representation learning, augmentations are used to construct contrastive views for learning task-agnostic embeddings. While in AutoDA, the augmentations are directly optimized to improve downstream task performance. However, both paradigms have key limitations: representation learning typically follows a two-stage pipeline with limited adaptability, and current AutoDA frameworks are largely designed for image data, rendering them ineffective for capturing time series–specific features. To address these issues, we propose **AutoDA-Timeseries**, the first general-purpose AutoDA framework tailored for time series. AutoDA-Timeseries incorporates time series features into augmentation policy design and adaptively optimizes both augmentation probability and intensity in a single-stage, end-to-end manner. We conduct extensive experiments on five mainstream tasks, including classification, long-term forecasting, short-term forecasting, regression, and anomaly detection, showing that AutoDA-Timeseries consistently outperforms strong baselines across diverse models and datasets.

## 1 INTRODUCTION

Data augmentation refers to a series of transformations that generate high-quality artificial data by manipulating existing samples, serving as a fundamental approach in deep learning to improve model performance and robustness (Shorten & Khoshgoftaar, 2019; Wang et al., 2024). Existing applications of data augmentation can be broadly categorized into two paradigms. The first paradigm is *representation learning*, where augmentations are used to construct contrastive samples, enabling models to learn task-agnostic representations (Chen et al., 2020; He et al., 2020). The second paradigm is *automated data augmentation (AutoDA)*, which focuses on automatically searching or generating augmentation strategies that directly optimize downstream model performance while reducing the reliance on manual design and tuning (Cubuk et al., 2019; 2020).

In time series analysis tasks, data augmentation is equally indispensable due to data insufficiency and homogeneity (Wen et al., 2020; Iwana & Uchida, 2021; Iglesias et al., 2023). As illustrated in Figure 1, these two application paradigms differ in their training pipelines when applied to time series analysis tasks. In the representation learning paradigm, the encoder is first pretrained with contrastive learning on augmented views, and then transferred to downstream tasks through a separate fine-tuning stage, where the downstream model adapts to the learned representations (Yue et al., 2022; Luo et al., 2023). However, a key limitation of representation learning lies in the adaptability of downstream models to the learned representations. For instance, recurrent neural networks (RNNs) are inherently designed for sequence-to-sequence prediction (Sutskever et al., 2014), excelling at modeling long-term dependencies and dynamic evolution rather than capturing invariant

---

*Corresponding author.

representations emphasized by contrastive learning (Chen et al., 2020). In contrast, AutoDA follows a one-stage scheme where augmentations are jointly optimized with the downstream task. Augmentation policies, including the choice probability and intensity of transformation, are adaptively tuned during training, producing high-quality and diverse samples tailored to the downstream task and directly enhancing downstream performance.

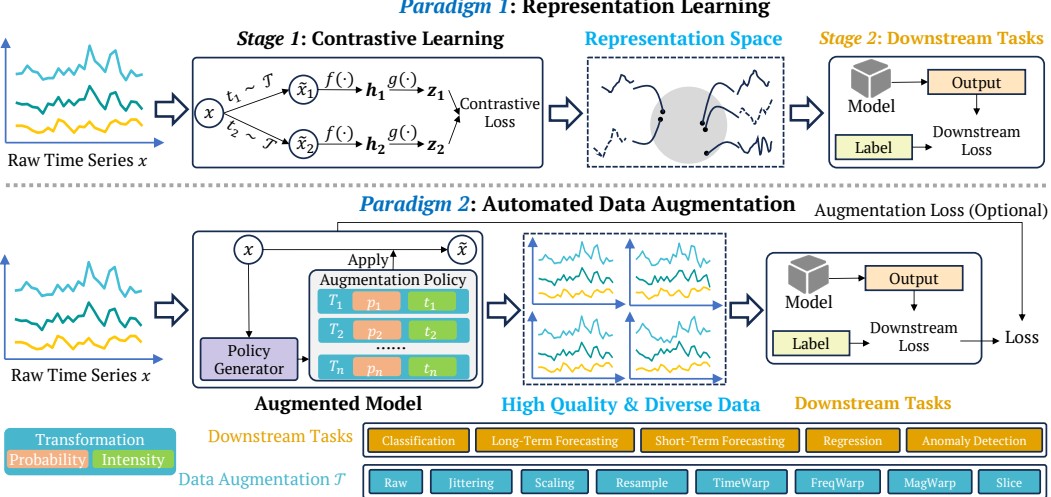

Figure 1: Two application paradigms of time series data augmentation: representation learning and AutoDA.

While representation learning frameworks suffer from limitations in adapting to downstream models, AutoDA provides a promising alternative by jointly optimizing augmentations with downstream model training. However, existing AutoDA approaches have been predominantly developed for image data and are not directly applicable to time series due to the inherent differences between modalities. Even state-of-the-art (SOTA) of these AutoDA frameworks, including RandAugment (Cubuk et al., 2020), TrivialAugment (Müller & Hutter, 2021), UniformAugment (LingChen et al., 2020), and A2Aug (Li & Li, 2023), still face the following key challenges in the context of automated time series augmentation:

- **Limited task generalization**. Most existing AutoDA methods are validated on a single task. This narrow evaluation setting overlooks the fact that augmentation policies may not generalize well when applied to different time series tasks with distinct objectives.

- **Neglect of time series characteristics**. Existing AutoDA frameworks ignore time series-specific features (e.g., autocorrelation, distribution, high-order features) when generating augmentation policies. Their assumption that transformations preserve semantic validity as in image domains fails for time series modality where critical time series features govern augmentation effectiveness, and modality-agnostic approaches risk distorting intrinsic data properties, yielding suboptimal strategies. For instance, frequency-warping-based augmentations blindly applied without considering autocorrelation patterns may disrupt temporal dependencies, degrading downstream classification or forecasting model performance.

- **Lack of adaptive policy learning**. Previous SOTA AutoDA frameworks rely on uniform sampling to determine both the types and intensities of augmentation transformations, treating all transformations equally important without considering their varying impacts on time series data. This uniform design fails to account for the fact that different transformations and intensities may contribute unevenly to the effectiveness of the augmentation policy, potentially leading to suboptimal or inappropriate augmentation policies.

To address these challenges, we propose a general-purpose automated data augmentation framework for time series, **AutoDA-Timeseries**. It employs an augmentation data generator that learns a combination distribution of selection probability and the intensity for each augmentation transformation,

conditioned on the time series features. AutoDA-Timeseries offers several advantages: **First**, it provides *a unified one-stage framework* that jointly optimizes augmentation policies with downstream task objectives, ensuring broad applicability across *diverse time series tasks*. **Second**, when choosing the optimal augmentation policy for each time series, it integrates multiple *time series features*, making it suitable for automated augmentation in the time series domain. **Finally**, the framework performs *adaptive* augmentation of both probability and intensity, which can more properly reflect the distribution of the optimal augmentation policy.

To summarize, our key contributions are as follows:

- **Comprehensive revisit of data augmentation application paradigms**: we analyze the limitations of existing paradigms, representation learning and automated data augmentation, highlighting their restricted adaptability and the absence of time series-specific design.

- **AutoDA-Timeseries framework**: we propose the *first* general-purpose automated data augmentation framework for time series, which incorporates time series-specific features into augmentation selection and jointly optimizes both augmentation model and downstream model in a single-stage, end-to-end manner.

- **Extensive empirical validation**: we conduct extensive experiments on five mainstream tasks, demonstrating the superiority, robustness, and generalization of AutoDA-Timeseries through detailed evaluations and visualizations.

## 2 RELATED WORK

Time series augmentation refers to a collection of advanced techniques designed to artificially expand and diversity existing time series datasets. Previous studies have surveyed various time series augmentation transformations proposed for different downstream tasks, such as classification and segmentation (Iwana & Uchida, 2021; Wen et al., 2020; Alomar et al., 2023; Iglesias et al., 2023; Mohammadi Foumani et al., 2024), or forecasting and anomaly detection (Wen et al., 2020; Iglesias et al., 2023; Semenoglou et al., 2023). Representative transformations include jittering (Salamon & Bello, 2017), rotation (Ohashi et al., 2017), scaling (Ohashi et al., 2017), slicing (Pan et al., 2020), permuting (Um et al., 2017), time warping (Le Guennec et al., 2016), magnitude warping (Demir et al., 2021), and several other techniques (Wen et al., 2020). Beyond the level of individual transformations, recent research has further explored two broader paradigms for leveraging data augmentation: *representation learning* and *automated data augmentation (AutoDA)*.

Representation learning aims to learn task-agnostic representations that can transfer across diverse downstream tasks. TS2Vec introduces hierarchical contrastive objectives together with contextual consistency (Yue et al., 2022). InfoTS leverages the information bottleneck principle and employs adaptive augmentations to generate diverse views, thereby learning more discriminative representations (Luo et al., 2023). AutoTCL proposes a contrastive learning framework with parametric augmentations (Zheng et al., 2024). AutoCL adaptively adjusts augmentation strength through cross-scale temporal consistency constraints (Jing et al., 2024). CAAP learns an adversarial augmentation policy that produces task-aware perturbations guided by contrastive objectives (Chang et al., 2024). FreRA leverages frequency-domain statistics to adaptively decide augmentation direction and intensity (Tian et al., 2025). Despite their effectiveness, most representation learning frameworks adopt a two-stage pipeline. In the first stage, multiple augmented views of the same time series are generated, and an encoder is trained using contrastive objectives to obtain task-agnostic representations. In the second stage, the pretrained encoder is transferred and adapted to downstream models. However, these two stages are decoupled: the augmentation strategy and representation learning in Stage 1 are optimized entirely for the contrastive objective and cannot perceive feedback from the downstream model in Stage 2, particularly when the downstream model is not explicitly designed to leverage such representations. As a result, the learned representations may not always align well with the objectives or architectures of downstream models, which limits the performance gains in practical scenarios.

AutoDA is proposed to generate optimal augmentation policies, mainly in computer vision (CV) domain (Yang et al., 2023). Early studies proposed **two-stage proxy-based** frameworks, such as TANDA (Ratner et al., 2017) and AutoAugment (Cubuk et al., 2019), where a smaller proxy model

was trained to evaluate candidate policies. Although effective, these methods are computationally expensive and often fail to generalize due to the mismatch between proxy and downstream models (Cubuk et al., 2020). ReAugment uses a variational masked autoencoder (VMAE) to reconstruct masked raw samples and learn their underlying data distribution, then applies reinforcement learning to adjust the VMAE's latent variable to generate augmented sequences that preserve the original structure (Yuan et al., 2024). More recent work has shifted toward **one-stage non-proxy** AutoDA frameworks, which directly optimize augmentation policies with the downstream task. Representative approaches include RandAugment (Cubuk et al., 2020), TrivialAugment (Müller & Hutter, 2021), UniformAugment (LingChen et al., 2020), and A2Aug (Li & Li, 2023). These methods eliminate proxy models and instead rely on the simple randomization or ensemble strategies to reduce cost while improving downstream performance. However, applying such frameworks to time series remains challenging, as they lack adaptive augmentation mechanisms and ignore modality-specific features that are crucial for preserving intrinsic patterns (Christ et al., 2018; Lubba et al., 2019).

## 3 METHODOLOGY

### 3.1 PROBLEM STATEMENT

Let $\mathcal{D} = \{\mathbf{D}_1, \mathbf{D}_2, \ldots, \mathbf{D}_m\}$ be a time series dataset, where $\mathbf{D}_i$ ($i = 1 \ldots m$) is a univariate or multivariate time series. Let $\mathcal{M}$ denote a downstream model (e.g., a classifier) whose trainable parameters are denoted as $\theta_{\mathcal{M}}$. We consider a set of time series augmentation transformations $\mathcal{T} = \{T_1, T_2, \ldots, T_n\}$, where $T_j$ ($j = 1 \ldots n$) is an augmentation operator that can be applied to a given time series $\mathbf{D}_i$ to produce an augmented view of $\mathbf{D}_i$. Our goal is to design an **automated time series augmentation framework** $A_\theta$ parameterized by $\theta$ that outputs a policy $P_i = A_\theta(\mathbf{D}_i)$ for each $\mathbf{D}_i \in \mathcal{D}$. $P_i$ consists of two vectors: (i) a *probability* vector $p_i$, where $p_{i,j} \in [0, 1]$ is the probability $T_j$ is selected; and (ii) a *intensity* vector $t_i$, where $t_{i,j} \geq 0$ is the intensity of $T_j$. After applying $P_i$ to $\mathbf{D}_i$, we can obtain the augmented time series $P_i(\mathbf{D}_i)$. By performing this operation for the entire dataset $\mathcal{D}$, we obtain the *augmented dataset* $A_\theta(\mathcal{D})$, that is,

$$A_\theta(\mathcal{D}) = \{P_i(\mathbf{D}_i) | \mathbf{D}_i \in \mathcal{D}\}. \tag{1}$$

We then train the downstream model $\mathcal{M}$ on the augmented dataset by minimizing a task-related loss as follows:

$$\theta_{\mathcal{M}}^* = \arg\min_{\theta_{\mathcal{M}}} L(\theta_{\mathcal{M}}, A_\theta(\mathcal{D})), \tag{2}$$

where $L$ is the loss function of the specific task (e.g., mean squared error for forecasting, cross-entropy for classification, etc.). Finally, we evaluate the trained model $\mathcal{M}$ using the *original* dataset $\mathcal{D}$, aiming to achieve superior performance with respect to the loss function $L$. The objective thus becomes finding the optimal parameter $\theta^*$ for the augmentation framework $A_\theta$:

$$\theta^* = \arg\min_{\theta} L(\theta_{\mathcal{M}}^*, \mathcal{D}). \tag{3}$$

The automated time series augmentation is formulated as a joint optimization over both the augmentation framework's parameters $\theta$ and the downstream model's parameters $\theta_{\mathcal{M}}$.

### 3.2 AUTODA-TIMESERIES OVERVIEW

As shown in the Figure 2, a time series feature-aware augmented data generator (denoted as $A_\theta$) is composed of multiple stacked *Augmentation Layers* $A_{\theta_k}^{(k)}$, each of which is responsible for selecting and applying one of the available transformations in the set $\mathcal{T} = \{T_1, T_2, \ldots, T_n\}$.

The $k$-th augmentation layer generates an augmentation policy consisting of (i) a series of probability $p_{i,j}^{(k)}$ indicating the likelihood of choosing transformation $T_j$ and (ii) a series of intensity $t_{i,j}^{(k)}$ to apply a chosen transformation. By stacking these augmentation layers, the framework can explore a variety of transformation sequences, allowing for more diverse and potentially useful augmented data. The final output augmented time series is used to train a single downstream model in a single-stage, end-to-end manner, with a composite loss to update the parameters in the augmented data generator together with the downstream model.

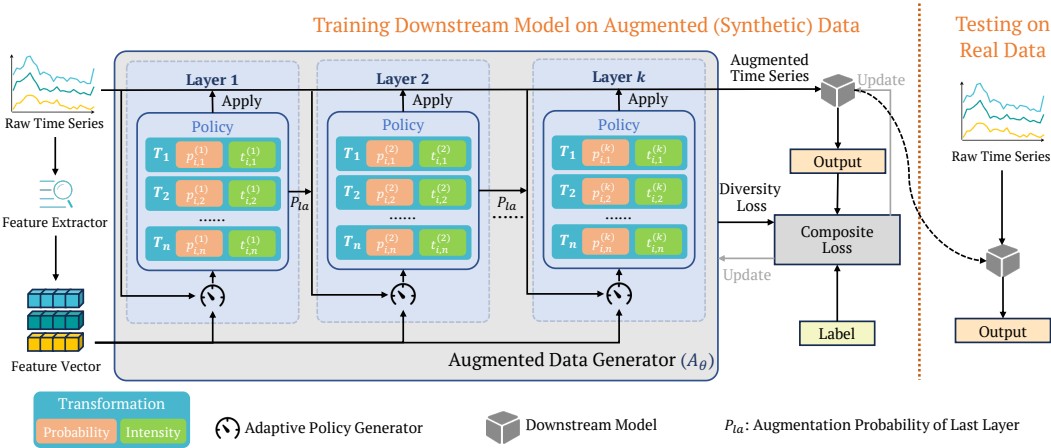

Figure 2: Overall architecture of AutoDA-Timeseries.

## 3.3 TIME SERIES FEATURE EXTRACTION

Following the prior work (Qiu et al., 2024), we extracted 24 descriptive statistics for each time series in the original dataset, forming a feature vector $\mathbf{F}_i = f_e(D_i)$, where $f_e(\cdot)$ denotes our feature extraction function. These features are effective across various time series classification and forecasting tasks (Lubba et al., 2019; Qiu et al., 2024). In our design, the feature vector $\mathbf{F}_i$ remains unchanged and static across layers to preserve the global context of the original time series, preventing distortion from sequential augmentations while stabilizing training.

## 3.4 STACKED AUGMENTATION LAYERS

Our framework $A_\theta$ is composed of $K$ sequential augmentation layers: $A_\theta = A_{\theta_1}^{(1)} \circ A_{\theta_2}^{(2)} \circ \cdots \circ A_{\theta_K}^{(K)}$. Each layer $A_{\theta_k}^{(k)}$ receives (i) the input time series $\mathbf{D}_i^{(k-1)}$ from previous layer (raw time series $\mathbf{D}_i$ for the first layer), (ii) the previous probability vector $p_i^{(k-1)}$ (initialized as zeros), and (iii) the global feature vector $\mathbf{F}_i$. It then generates the probability $p_i^{(k)}$ and intensity $t_i^{(k)}$ via MLPs $f_p^{(k)}$ and $f_t^{(k)}$:

$$p_i^{(k)} = f_p^{(k)}\big(p_i^{(k-1)}, \mathbf{F}_i\big), \tag{4}$$

$$t_i^{(k)} = f_t^{(k)}\big(p_i^{(k-1)}, \mathbf{F}_i\big). \tag{5}$$

A transformation $T_{r_k}$ is then sampled in each layer by a Gumbel-Softmax (Jang et al., 2016) approximation (denoted $\sigma_{gs}$), which ensures that the framework remains differentiable. The selected transformation $T_{r_k}$ is applied to $\mathbf{D}_i^{(k-1)}$ with intensity $t_{i,r_k}^{(k)}$ to generate the augmented time series:

$$T_{r_k} = \sigma_{gs}\big(\mathcal{T}, p_i^{(k)}\big), \tag{6}$$

$$\mathbf{D}_i^{(k)} = T_{r_k}\big(\mathbf{D}_i^{(k-1)}, t_{i,r_k}^{(k)}\big). \tag{7}$$

By stacking these augmentation layers, the framework performs sequential transformations. The final output $\mathbf{D}_i^{(K)} = A_\theta(\mathbf{D}_i)$ is fed to the downstream model. All layer parameters are jointly optimized with the downstream model via a composite loss backpropagation.

## 3.5 STRATEGIES FOR EXPLORATION AND EXPLOITATION

To balance exploration (experimenting with diverse transformations) and exploitation (converging on effective augmentations), we incorporate the following strategies:

### 3.5.1 LEARNABLE GUMBEL-SOFTMAX TEMPERATURE

We adopt a learnable temperature parameter in the Gumbel-Softmax distribution to control the randomness of transformation sampling (Jang et al., 2016). Each augmentation layer maintains its own

temperature, and all temperatures are optimized purely via backpropagation. A higher temperature encourages exploration by making the selection probabilities more uniform, while gradually lowering the temperature increases determinism and helps the model converge to the most promising transformation choices.

### 3.5.2 COMPOSITE LOSS FUNCTION

To maintain diversity in the transformation probability distribution, we encourage the augmentation layer to output diverse transformation probabilities. Therefore, in addition to the task-specific loss, we introduce diversity loss terms. To address the weight setting problem for multiple losses, inspired by previous work (Liebel & Körner, 2018), we employ learnable weights in the final composite loss $L_{\text{composite}}$ as follows:

$$L_{\text{composite}} = \sum_{z=1,2,3} \left[ \frac{1}{2w_z^2} L_z + \ln(1 + w_z^2) \right], \tag{8}$$

where: (1) $L_1$ is the task-specific loss, e.g., mean squared error for forecasting, or cross-entropy for classification. (2) $L_2$ is an intra-layer diversity loss to encourage diverse transformations within a layer, which is defined as:

$$L_2 = \sum_{k=1}^{K} \mathbb{E}_i \left[ H(p_i^{(k)}) \right], \tag{9}$$

where $K$ is the number of augmentation layers, $\mathbb{E}_i[\cdot]$ means averaging over samples, $p_i^{(k)} \in \mathbb{R}^n$ is the augmentation probability vector for $\mathbf{D}_i$ at the $k$-th layer, and $H(p_i^{(k)})$ denotes the Shannon entropy calculated as follows:

$$H(p_i^{(k)}) = -\sum_{j=1}^{n} p_{i,j}^{(k)} \log(p_{i,j}^{(k)} + \epsilon), \tag{10}$$

where $n$ is the number of transformations, and $\epsilon$ is a small constant added for numerical stability (set to $10^{-10}$ in our implementation). (3) $L_3$ is an inter-layer diversity loss, which measures the divergence between the augmentation probability distribution of the current layer and that of the previous layer and is defined as:

$$L_3 = \sum_{k=2}^{K} \mathbb{E}_i \left[ \text{KL}(p_i^{(k-1)} \| p_i^{(k)}) \right], \tag{11}$$

where $\text{KL}(p_i^{(k-1)} \| p_i^{(k)}) = \sum_{j=1}^{n} p_{i,j}^{(k-1)} \left[ \log \left( p_{i,j}^{(k-1)} \right) - \log \left( p_{i,j}^{(k)} \right) \right]$. (4) $w_z^2$s are learnable weights to achieve trade-off between diversity and task performance during the training.

This composite loss enables the augmented data generator and the downstream model to be jointly optimized in a fully end-to-end manner.

### 3.5.3 RAW TRANSFORM BIAS

To avoid overfitting to augmented data, we add a bias term $p_{rb}$ that selects the raw data with probability $p_{rb}$:

$$T_{r_k} = \begin{cases} \sigma_{gs}(\mathcal{T}, p_i^{(k)}) & \text{with probability } (1 - p_{rb}), \\ T_1 & \text{with probability } p_{rb}, \end{cases}$$

where $T_1$ denotes the Raw (no transformation) operator.

## 4 EXPERIMENTS

We conduct extensive experiments to systematically evaluate the effectiveness of AutoDA-Timeseries on five mainstream time series analysis tasks: classification, long-term forecasting, short-term forecasting, regression, and anomaly detection. Beyond quantitative comparisons with state-of-the-art baselines, we also provide in-depth analyses and insights into AutoDA-Timeseries.

## 4.1 EXPERIMENT SETUP

**Implementation** Table 1 summarizes the benchmarks, evaluation metrics, and representative downstream models for each task. Following the prior work (Zheng et al., 2024), we evaluate on representative downstream models and extend the scope by incorporating both classical and advanced architectures, covering convolutional, recurrent, Transformer-based, and generative paradigms, to assess the generalizability of AutoDA-Timeseries. More detailed descriptions can be found in Appendix A.

Table 1: Summary of benchmarks, evaluation metrics, and representative downstream models.

| Tasks | Benchmarks | Metrics | Downstream Models |
|---|---|---|---|
| Classification | UEA (26 subsets) | Accuracy | TCN, ROCKET |
| Forecasting | **Long-term**: ETT (4 subsets), Exchange, Weather | MSE, MAE | RNN, Autoformer |
| | **Short-term**: M4 (6 subsets) | SMAPE, MASE, OWA | |
| Regression | UEA & UCR (6 subsets) | MSE, MAE | CNN, MLP |
| Anomaly Detection | MSL, SMAP, SMD | Precision, Recall, F1-score | UNet, VAE |

**Baselines** We compare AutoDA-Timeseries with three groups of baselines to ensure a comprehensive and fair evaluation. We use *NoAug* as the **control group**, which does not apply any augmentation. For **representation learning**, we adopt InfoTS (Luo et al., 2023), AutoTCL (Zheng et al., 2024), and TS2Vec (Yue et al., 2022), which leverage data augmentation to construct contrastive views and learn task-agnostic representations in a two-stage manner. For **automated data augmentation**, we consider four state-of-the-art methods: RandAugment (Cubuk et al., 2020), UniformAugment (LingChen et al., 2020), TrivialAugment (Müller & Hutter, 2021), and A2Aug (Li & Li, 2023). More detailed descriptions of these baselines can be found in Appendix B.

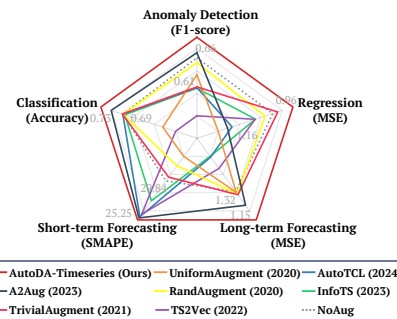

Figure 3: Overall comparison of AutoDA-Timeseries with baselines across five time series tasks.

## 4.2 RESULTS

Figure 3 presents an overall comparison of AutoDA-Timeseries with state-of-the-art baselines across five time series tasks. We observe that AutoDA-Timeseries consistently achieves the best performance, covering the largest area in the radar plot. Next, we provide a more detailed analysis for each task.

### 4.2.1 CLASSIFICATION

**Setups** Time series classification aims to assign a discrete label to each sample, which can be either a univariate or multivariate time series (Ismail Fawaz et al., 2019). We evaluate 26 subsets selected from the UEA archive (Bagnall et al., 2018), covering diverse domains such as audio recognition, human activity recognition, and healthcare monitoring. Following the prior work (Liu et al., 2024), we use accuracy as the evaluation metric, and adopt TCN (Bai et al., 2018) and ROCKET (Dempster et al., 2020) as representative downstream models.

**Results** As shown in Figure 4, AutoDA-Timeseries achieves the best accuracy, reaching 0.730 (+6.7%) with TCN and 0.721 (+5.2%) with ROCKET, significantly surpassing the *NoAug* control. Traditional AutoDA methods (RandAugment, UniformAugment, and TrivialAugment) yield limited or even negative gains, highlighting the gap in directly transferring image-based augmentation policies to time series. Representation learning methods show instability: TS2Vec suffers severe

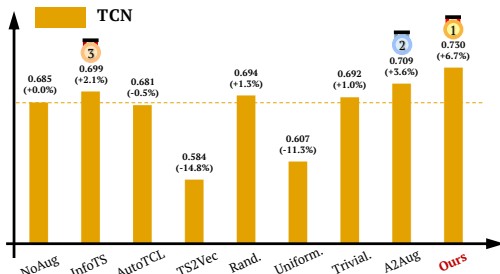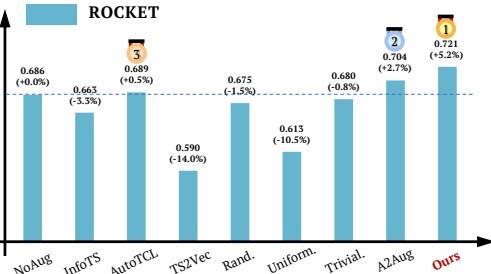

Figure 4: Classification accuracy comparison of AutoDA-Timeseries and baselines on TCN (orange) and ROCKET (blue). "∗." in the method names denotes ∗Augment. Full results are provided in Table 15 and Table 16 in the Appendix.

degradation, while AutoTCL and InfoTS achieve only marginal gains. These results suggest that the augmentation policies of AutoDA-Timeseries can consistently boost classification accuracy and generalize across different downstream models.

### 4.2.2 LONG- AND SHORT-TERM FORECASTING

**Setups** Time series forecasting is a fundamental task with wide applications in weather, traffic, energy, and finance. We evaluate AutoDA-Timeseries on both long- and short-term forecasting. For long-term forecasting, we use ETT (4 subsets) (Zhou et al., 2021), Exchange (Lai et al., 2018), and Weather (Wetterstation), with MSE and MAE as metrics, following the prior work (Wu et al., 2022). For short-term forecasting, we adopt the M4 competition setup with six subsets (Spyros Makridakis, 2018), using SMAPE, MASE, and OWA as metrics. Representative downstream models include RNN-based forecasters and Autoformer (Wu et al., 2021).

**Results** As shown in Tables 2 and 3, AutoDA-Timeseries achieves the best results on both long- and short-term forecasting. For long-term forecasting, AutoDA-Timeseries attains the lowest MSE and MAE on both RNN and Autoformer. We also observe that representation learning suffers larger relative degradation on RNN than on Autoformer, as Autoformer is more compatible with learned representations. For short-term forecasting, AutoDA-Timeseries again outperforms all baselines on RNN and Autoformer.

Table 2: Comparison of long-term forecasting performance across baselines and AutoDA-Timeseries. "∗." in the method names denotes ∗Augment. Full results are provided in Table 17 and Table 18 in the Appendix.

| Downstream Model | Metrics | Methods | | | | | | | | |
|---|---|---|---|---|---|---|---|---|---|---|
| | | NoAug | InfoTS | AutoTCL | TS2Vec | Rand. | Uniform. | Trivial. | A2Aug | **Ours** |
| RNN | MSE | 0.5408 | 1.5163 | 1.4888 | 1.3851 | 0.5114 | 0.4416 | 0.5193 | 0.6342 | **0.3968** |
| | MAE | 0.5381 | 1.5423 | 1.5167 | 1.4151 | 0.5117 | 0.4389 | 0.5148 | 0.6347 | **0.3930** |
| Autoformer | MSE | 2.4274 | 2.2761 | 2.2872 | 2.1240 | 2.4055 | 2.5116 | 2.3758 | 2.0155 | **1.9098** |
| | MAE | 2.4883 | 2.3323 | 2.2626 | 2.1779 | 2.4655 | 2.5755 | 2.4254 | 2.0617 | **1.9548** |

### 4.2.3 REGRESSION

**Setups** Time series regression predicts a continuous scalar from an input time series, differing from classification (discrete labels) and forecasting (future values) (Tan et al., 2021). In particular, it generalizes forecasting by relaxing the requirement that the target must depend primarily on recent values, and has broad applications such as heart rate estimation from physiological signals (Reiss et al., 2019) or crop yield prediction from satellite observations (Yebra et al., 2018). We evaluate six subsets from the UEA & UCR archives (Tan et al., 2020), using MSE and MAE as metrics, with CNN and MLP as downstream models.

Table 3: Comparison of short-term forecasting performance across baselines and AutoDA-Timeseries. "∗." in the method names denotes ∗Augment.

| Downstream Model | Metrics | Methods | | | | | | | | |
|---|---|---|---|---|---|---|---|---|---|---|
| | | NoAug | InfoTS | AutoTCL | TS2Vec | Rand. | Uniform. | Trivial. | A2Aug | **Ours** |
| RNN | SMAPE | 11.384 | 12.454 | 13.143 | 13.832 | 12.910 | 11.962 | 11.482 | 11.980 | **11.068** |
| | MASE | 1.774 | 1.864 | 2.027 | 2.624 | 2.536 | 1.778 | 1.736 | 1.985 | **1.644** |
| | OWA | 0.883 | 0.981 | 1.009 | 1.142 | 1.139 | 0.906 | 0.877 | 0.961 | **0.838** |
| Autoformer | SMAPE | 57.854 | 47.219 | **38.875** | 39.389 | 63.573 | 69.034 | 59.541 | 39.456 | 39.425 |
| | MASE | 14.865 | 15.216 | 10.406 | 7.790 | 48.076 | 16.301 | 15.729 | 7.818 | **7.762** |
| | OWA | 6.020 | 3.359 | 4.154 | **3.482** | 14.915 | 6.807 | 6.308 | 3.499 | 3.490 |

**Results** Regression inherently relies on precise continuous value mappings, making it highly sensitive to the quality of augmented data. As shown in Table 4, AutoDA-Timeseries achieves state-of-the-art performance across diverse regression datasets, verifying the effectiveness of its task-adaptive augmentation strategy.

Table 4: Comparison of regression performance across baselines and AutoDA-Timeseries. "∗." in the method names denotes ∗Augment. Full results are provided in Table 19 and Table 20 in the Appendix.

| Downstream Model | Metrics | Methods | | | | | | | | |
|---|---|---|---|---|---|---|---|---|---|---|
| | | NoAug | InfoTS | AutoTCL | TS2Vec | Rand. | Uniform. | Trivial. | A2Aug | **Ours** |
| CNN | MSE | 0.9285 | 1.1025 | 1.1290 | 1.0892 | 1.0951 | 1.4714 | **0.8875** | 1.6016 | 0.8921 |
| | MAE | 0.6821 | 0.7386 | 0.7343 | 0.7211 | 0.7545 | 0.7477 | 0.6814 | 0.7160 | **0.6731** |
| MLP | MSE | 1.2937 | 1.4036 | 1.4197 | 1.3441 | 1.2196 | 1.4032 | 1.2744 | 1.2157 | **1.0350** |
| | MAE | 0.7010 | 0.7352 | 0.7348 | 0.7320 | 0.6695 | 0.7433 | 0.6729 | 0.6652 | **0.6420** |

### 4.2.4 ANOMALY DETECTION

**Setups** Time series anomaly detection aims to identify rare or abnormal patterns that deviate from normal temporal dynamics. The main challenge lies in the scarcity and diversity of anomaly samples, making data augmentation particularly crucial. We follow standard benchmarks (Hundman et al., 2018; Su et al., 2019) and use F1-score as the primary metric. Representative models include UNet (Gao et al., 2020) and VAE (Xu et al., 2018).

**Results** As shown in Table 5, anomaly detection is highly sensitive to augmentation, since inappropriate transformations may erase or mimic rare anomalies, making them harder to detect. Nevertheless, AutoDA-Timeseries consistently achieves superior results on both models, showing that adaptive policies enhance model robustness and generalize to augmentation-sensitive tasks.

Table 5: Comparison of anomaly detection performance across baselines and AutoDA-Timeseries. "∗." in the method names denotes ∗Augment. Full results are provided in Table 21 and Table 22 in the Appendix.

| Downstream Model | Metrics | Methods | | | | | | | | |
|---|---|---|---|---|---|---|---|---|---|---|
| | | NoAug | InfoTS | AutoTCL | TS2Vec | Rand. | Uniform. | Trivial. | A2Aug | **Ours** |
| UNet | F1 | 0.6991 | 0.6912 | 0.6944 | 0.6173 | 0.6844 | 0.7171 | 0.6886 | 0.6993 | **0.7478** |
| VAE | F1 | 0.5592 | 0.4887 | 0.4871 | 0.4914 | 0.5610 | 0.4973 | 0.4945 | 0.5591 | **0.5761** |

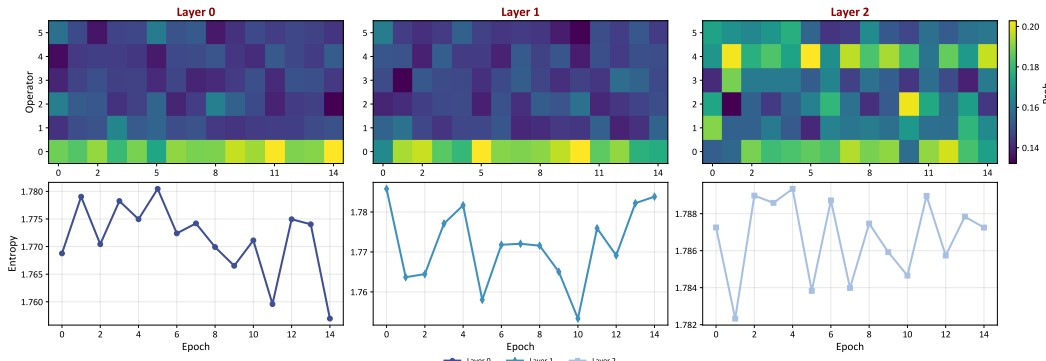

Figure 5: Adaptive augmentation policy. Top: operator distribution over training epochs. Bottom: entropy dynamics showing convergence in lower layers and diversity in higher layers.

## 4.3 MODEL ANALYSIS

**Adaptive Augmentation Policy Visualization** We investigate how augmentation policies evolve during training by visualizing augmentation operator probabilities and entropy across layers in the augmentation process (Figure 5).

The results reveal a clear layer-wise differences. Layer 0 rapidly converges to a few operators (e.g., Raw augmentation), reflecting deterministic exploitation, while upper layers maintain higher entropy and more diverse policies. This pattern illustrates the exploitation-exploration trade-off (Sutton et al., 1998), where lower layers stabilize the augmentation policies and upper layers remain adaptive, providing a complementary balance between stability and diversity.

**Feature-Space Consistency under Augmentation** We examine whether augmentations preserve time series features as shown in Figure 6. The catch22 features of augmented data remain highly consistent with those of the raw data, indicating that AutoDA-Timeseries maintains essential characteristics and further supports our motivation of incorporating time series feature extraction (Section 3.3).

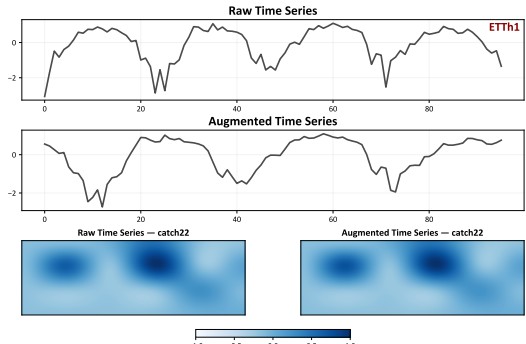

Figure 6: Feature-space consistency under augmentation.

## 5 CONCLUSION AND FUTURE WORK

In this paper, we proposed AutoDA-Timeseries, a general-purpose framework that adaptively learns augmentation policies conditioned on time series features and jointly optimizes them with downstream models. Experiments across diverse tasks verify its superiority and clear advantages over existing augmentation paradigms. In future work, we aim to extend the framework to real-world time series applications, which often involve diverse domains and complex dynamics.

## 6 ETHICS STATEMENT

This work focuses on methodological advances in automated data augmentation for time series analysis and does not involve sensitive personal data. All experiments are conducted on publicly available benchmark datasets. We confirm compliance with the ICLR Code of Ethics.

## 7 REPRODUCIBILITY STATEMENT

We ensure reproducibility through detailed descriptions of our method and experiments. The model architecture is formally defined in the main text, and all implementation details are provided in the Appendix, including dataset descriptions, evaluation metrics, hyperparameters, training configurations, and experimental settings. All experiments are conducted under consistent protocols for fair comparison.

## ACKNOWLEDGMENTS

This work is supported by the National Key Research and Development Program of China (No.2024YFB4505903) and the China Postdoctoral Science Foundation (Grant No.2025M771563). We thank Prof. Changhua Pei (Computer Network Information Center, Chinese Academy of Sciences) for helpful discussions and technical guidance.

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

## A  IMPLEMENTATION DETAILS

All experiments were conducted on a workstation equipped with a single NVIDIA GeForce RTX 3080 Ti GPU. To evaluate the effectiveness of AutoDA-Timeseries, we conduct experiments on a wide range of benchmark datasets across five mainstream tasks, including classification, long-term forecasting, short-term forecasting, regression, and anomaly detection. The detailed statistics of the datasets are provided in Tables 6, 7, and 8.

We conducted experiments across multiple datasets with $K = \{1, 2, 3, 4, 5\}$, and selected the hyperparameter value that performs well on most datasets, namely $K = 3$. Therefore, in the main experiments, all datasets use $K = 3$.

## B  BASELINE DESCRIPTIONS

To comprehensively evaluate the performance of the AutoDA-Timeseries framework, the following baselines were applied to the same downstream models:

- NoAug: No augmentation was applied; the downstream model was trained directly on the raw dataset.

Table 6: Summary of benchmark datasets for time series classification.

| Datasets | Code | Classes | Dims | Length | Test Size | Train Size | Type |
|---|---|---|---|---|---|---|---|
| ArticularyWordRecognition | AWR | 25 | 9 | 144 | 300 | 275 | Motion |
| AtrialFibrillation | AF | 3 | 2 | 640 | 15 | 15 | ECG |
| BasicMotions | BM | 4 | 6 | 100 | 40 | 40 | HAR |
| Cricket | CR | 12 | 6 | 1197 | 72 | 108 | HAR |
| DuckDuckGeese | DDG | 5 | 1345 | 270 | 50 | 50 | Audio |
| EigenWorms | EW | 5 | 6 | 17984 | 131 | 128 | Motion |
| Epilepsy | EP | 4 | 3 | 206 | 138 | 137 | HAR |
| ERing | ER | 6 | 4 | 65 | 270 | 30 | HAR |
| EthanolConcentration | EC | 4 | 3 | 1751 | 263 | 261 | Spectro |
| FaceDetection | FD | 2 | 144 | 62 | 3524 | 5890 | EEG |
| FingerMovements | FM | 2 | 28 | 50 | 100 | 316 | EEG |
| HandMovementDirection | HMD | 4 | 10 | 400 | 74 | 160 | EEG |
| Handwriting | HW | 26 | 3 | 152 | 850 | 150 | HAR |
| Heartbeat | HB | 2 | 61 | 405 | 205 | 204 | Audio |
| Libras | LIB | 15 | 2 | 45 | 180 | 180 | HAR |
| LSST | LSST | 14 | 6 | 36 | 2466 | 2459 | Astronomy |
| MotorImagery | MI | 2 | 64 | 3000 | 100 | 278 | EEG |
| NATOPS | NATOPS | 6 | 24 | 51 | 180 | 180 | HAR |
| PEMS-SF | PEMS-SF | 7 | 963 | 144 | 173 | 267 | Transportation |
| PenDigits | PD | 10 | 2 | 8 | 3498 | 7494 | Motion |
| PhonemeSpectra | PS | 39 | 11 | 217 | 3353 | 3315 | Audio |
| RacketSports | RS | 4 | 6 | 30 | 152 | 151 | HAR |
| SelfRegulationSCP1 | SCP1 | 2 | 6 | 896 | 293 | 268 | EEG |
| SelfRegulationSCP2 | SCP2 | 2 | 7 | 1152 | 180 | 200 | EEG |
| StandWalkJump | SWJ | 3 | 4 | 2500 | 15 | 12 | ECG |
| UWaveGestureLibrary | UWGL | 8 | 3 | 315 | 320 | 120 | HAR |

Table 7: Summary of benchmark datasets for time series regression.

| Datasets | Code | Dims | Length | Test Size | Train Size | Type |
|---|---|---|---|---|---|---|
| AppliancesEnergy | AE | 24 | 144 | 42 | 96 | Energy |
| FloodModeling1 | FM1 | 1 | 266 | 202 | 471 | Environment |
| FloodModeling2 | FM2 | 1 | 266 | 167 | 389 | Environment |
| FloodModeling3 | FM3 | 1 | 266 | 184 | 429 | Environment |
| LiveFuelMoistureContent | LFMC | 7 | 365 | 1510 | 3493 | Environment |
| IEEEPPG | IEEEPPG | 5 | 1000 | 1328 | 1768 | Healthcare |

- TS2Vec (Yue et al., 2022): TS2Vec is a universal representation learning framework designed for time series, which enables representation learning across multiple semantic levels. It achieves this by hierarchically distinguishing positive and negative samples at both the instance and temporal dimensions, thereby capturing rich contextual information for diverse downstream tasks.

- InfoTS (Luo et al., 2023): InfoTS is a contrastive learning-based method for time series augmentation. It generates two augmented views of the input using parameterized transformations and learns representations by maximizing mutual information between them. InfoTS applies instance-level contrastive loss to retain fine-grained semantic identity, particularly useful for downstream classification tasks.

- AutoTCL (Zheng et al., 2024): AutoTCL proposes a parametric framework for time series contrastive learning. It constructs two views using a learnable augmentation module, and maximize their alignment via InfoNCE loss. The augmentation parameters are optimized with a bi-level meta-learning strategy to enhance task performance.

- RandAugment (Cubuk et al., 2020): RandAugment is a proxy-free automated augmentation framework that has achieved state-of-the-art (SOTA) performance in image classification tasks, significantly optimizing performance compared to proxy-based frameworks.

Table 8: Summary of benchmark datasets for time series forecasting and anomaly detection. The "Dataset Size" column reports the number of samples in the training, validation, and testing splits, respectively.

| Tasks | Datasets | Dims | Length | Dataset Size | Type (Frequency) |
|---|---|---|---|---|---|
| Long-term Forecasting | ETTm1, ETTm2 | 7 | {96, 192, 336, 720} | (34465, 11521, 11521) | Electricity (15 mins) |
| | ETTh1, ETTh2 | 7 | {96, 192, 336, 720} | (8545, 2881, 2881) | Electricity (15 mins) |
| | Weather | 21 | {96, 192, 336, 720} | (36792, 5271, 10540) | Weather (10 mins) |
| | Exchange | 8 | {96, 192, 336, 720} | (5120, 665, 1422) | Exchange rate (Daily) |
| Short-term Forecasting | M4-Yearly | 1 | 6 | (23000, 0, 23000) | Demographic |
| | M4-Quarterly | 1 | 8 | (24000, 0, 24000) | Finance |
| | M4-Monthly | 1 | 18 | (48000, 0, 48000) | Industry |
| | M4-Weekly | 1 | 13 | (359, 0, 359) | Macro |
| | M4-Daily | 1 | 14 | (4227, 0, 4227) | Micro |
| | M4-Hourly | 1 | 48 | (414, 0, 414) | Other |
| Anomaly Detection | MSL | 55 | 100 | (44653, 11664, 73729) | Spacecraft |
| | SMAP | 25 | 100 | (108146, 27037, 427617) | Spacecraft |
| | SMD | 38 | 100 | (566724, 141681, 708420) | Server Machine |

- TrivialAugment (Müller & Hutter, 2021): TrivialAugment is a tuning-free, proxy-free automated augmentation framework that has demonstrated SOTA performance in image classification tasks.

- UniformAugment (LingChen et al., 2020): UniformAugment is a proxy-free AutoDA framework achieving high efficiency and comparable performance in image classification tasks with theoretical supports.

- A2Aug (Li & Li, 2023): A2Aug is a proxy-free AutoDA framework that trains multiple downstream models in parallel with different augmentation transforms and combines their outputs via ensemble learning, achieving SOTA performance in image classification tasks.

For CV-based AutoDA baseline, we did not naively apply these methods. Instead, we performed a rigorous time-series–specific adaptation of each method. Specifically, we made three categories of modifications:

- **Replacing image operations with time-series transformations.** The original RandAugment/TrivialAugment families rely on image operations such as rotation, shear, and color jitter, which are not meaningful for time series. To ensure fairness, we replaced their augmentation set with standard time-series transformations such as jittering, scaling, time-warping, etc. This guarantees that all baselines and our method use the same valid augmentation set.

- **Preserving each method's original sampling logic.** We strictly retained the core augmentation-selection mechanisms of each method: RandAugment preserves its $N$ random operations + global magnitude $M$ formulation. UniformAugment uniformly samples both operations and magnitudes. TrivialAugment samples one operation and magnitude per sample, following its original design. A2Aug learns augmentation weights jointly and ensembles operator logits adaptively.

- **Ensuring identical downstream settings for all baselines.** For fair comparison, all baselines use the same downstream models (RNN, Autoformer, etc.), the same data splits, sequence lengths, and batch sizes as AutoDA-Timeseries.

For time series-based representation learning baselines, we take the following measures to ensure fairness:

- First, all representation learning methods (TS2Vec, InfoTS, AutoTCL) are implemented using their official open-source repositories. We strictly follow their default hyperparameter configurations, including the number of training epochs, batch size, optimizer settings, and the built-in augmentation pipeline. We do not modify any internal architectural components or training procedures. This ensures that the results are fully reproducible and not influenced by implementation choices on our side.

- Second, unlike image-based AutoDA baselines, time-series representation learning methods already include augmentation operators specifically designed for sequential data. To

ensure fairness, we preserve the exact augmentation transformations defined in their official codebases. This prevents any methodological bias that might arise from altering or replacing their augmentation primitives.

- Finally, to guarantee fairness in downstream evaluation, all baselines adopt the same downstream configuration used in AutoDA-Timeseries. The downstream model architecture is kept identical across all methods, and every method is evaluated under the same data splits. For representation learning baselines, we follow their standard protocol: the encoder is first pretrained, and then frozen during downstream training while only the prediction head is optimized.

## C   ABLATION STUDIES

To verify the effectiveness of our key insights and the architecture designs introduced in Section 3, we conducted ablation studies. We remove each component from a complete AutoDA-Timeseries and evaluate their impacts by performance degradation.

The results are presented in Figure 7 and Table 9. As shown in Figure 7, most points lie above the diagonal, indicating that incorporating time series features, joint optimization, dynamic temperature, and composite loss consistently improves the performance of AutoDA-Timeseries on the classification task compared to their ablated versions. These results validate the necessity of the overall framework design, showing that each component contributes positively to the final performance, while removing any of them leads to performance degradation.

As shown in Table 9, first, disabling *Time Series Features* increased the MSE by up to 14.4%, which underscores the need for these features to guide augmentation, verifying our insight of performing augmentation policy generation conditioned on time series features. Second, removing *Joint Optimization* of probabilities and intensities led to an increase in MSE of up to 7.6%, which emphasizes the importance of generating the optimal combination of transformation types and strengths. Finally, the exploration-exploitation balancing strategies, including *Dynamic Temperature* and *Composite Loss*, all demonstrate clear effectiveness, reducing MSE by up to 7.4% and 8.1% relative to their ablated counterparts. Overall, these findings emphasize that each component is essential for the framework's performance and effectiveness in automated time series augmentation.

To verify that AutoDA-Timeseries meaningfully adapts augmentation strategies to time series dynamics, we conducted an ablation study based on the three major categories of Catch22 features as shown in Table 10. Specifically, different subsets of Catch22 capture key dynamic properties of time series:

- Linear and nonlinear autocorrelation features (e.g., CO_f1ecac, CO_FirstMin_ac) describe temporal dependency;
- Distribution-shape features (e.g., DN_HistogramMode_5, DN_HistogramMode_10) reflect whether the sequence contains sparse anomalies, skewness, or kurtosis;
- Differential-based features (e.g., SB_BinaryStats_diff_longstretch0) capture local fluctuations, short-term variation strength, and long-term stationarity.

We removed each of these feature categories respectively and evaluated performance across 9 UEA datasets using ROCKET as the downstream model. The results show that removing any category consistently leads to a noticeable performance drop. This indicates that temporal-structure features play an essential role in guiding the learned augmentation policy: the model does not simply apply general augmentation patterns, but instead relies on time-series-specific dynamic attributes to adjust both augmentation selection probabilities and augmentation strengths, thereby achieving true adaptation to time series dynamics.

## D   HYPER-PARAMETER SENSITIVITY

AutoDA-Timeseries involves two key hyper-parameters: the number of augmentation layers $K$ and the raw transform bias $p_{rb}$, which jointly determine the size of the augmentation search space and the proportion of raw samples retained during training. Specifically, the former controls how many

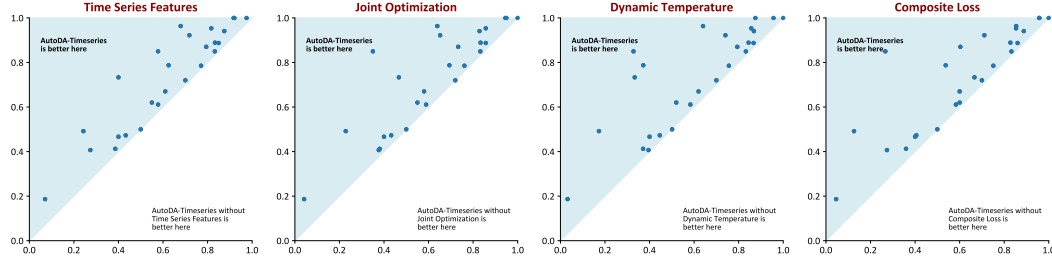

Figure 7: Ablation study of AutoDA-Timeseries on TCN for classification.

Table 9: Ablation study of AutoDA-Timeseries on RNN for long-term forecasting.

| Design Metrics | | w/o Time Series Features MSE | MAE | w/o Joint Optimization MSE | MAE | w/o Dynamic Temperature MSE | MAE | w/o Composite Loss MSE | MAE | AutoDA-Timeseries MSE | MAE |
|---|---|---|---|---|---|---|---|---|---|---|---|
| ETTh1 | 96 | 0.5071 | 0.4738 | 0.4884 | 0.4728 | 0.4903 | 0.4721 | 0.4906 | 0.4730 | 0.4849 | 0.4715 |
| | 192 | 0.5697 | 0.4989 | 0.5592 | 0.5310 | 0.5540 | 0.5019 | 0.5529 | 0.5262 | 0.5536 | 0.5046 |
| | 336 | 0.5876 | 0.5133 | 0.5468 | 0.4956 | 0.6049 | 0.5227 | 0.5843 | 0.5440 | 0.5552 | 0.4902 |
| | 720 | 0.5804 | 0.5209 | 0.5648 | 0.5174 | 0.6232 | 0.5458 | 0.6026 | 0.5635 | 0.5777 | 0.5161 |
| | Avg | 0.5612 | 0.5017 | **0.5398** | 0.5042 | 0.5681 | 0.5106 | 0.5576 | 0.5267 | 0.5429 | **0.4956** |
| ETTm1 | 96 | 0.6013 | 0.5031 | 0.4749 | 0.4758 | 0.4635 | 0.4486 | 0.5090 | 0.4957 | 0.4714 | 0.4500 |
| | 192 | 0.7106 | 0.5381 | 0.5179 | 0.5072 | 0.5207 | 0.4987 | 0.5336 | 0.5110 | 0.5107 | 0.4630 |
| | 336 | 0.7354 | 0.5542 | 0.5574 | 0.5246 | 0.5689 | 0.4900 | 0.5552 | 0.5210 | 0.5628 | 0.4881 |
| | 720 | 0.7612 | 0.5737 | 0.6083 | 0.5499 | 0.6203 | 0.5544 | 0.6219 | 0.5579 | 0.6071 | 0.5237 |
| | Avg | 0.7021 | 0.5423 | 0.5396 | 0.5144 | 0.5434 | 0.4979 | 0.5549 | 0.5214 | **0.5380** | **0.4812** |
| Exchange | 96 | 0.1236 | 0.2528 | 0.1188 | 0.2413 | 0.1319 | 0.2494 | 0.1299 | 0.2551 | 0.1086 | 0.2328 |
| | 192 | 0.2238 | 0.3412 | 0.2607 | 0.3632 | 0.2201 | 0.3353 | 0.2135 | 0.3287 | 0.2049 | 0.3234 |
| | 336 | 0.3745 | 0.4458 | 0.3945 | 0.4605 | 0.3610 | 0.4354 | 0.3989 | 0.4632 | 0.3582 | 0.4360 |
| | 720 | 0.9984 | 0.7622 | 0.9839 | 0.7570 | 0.9810 | 0.7562 | 0.9866 | 0.7582 | 0.6920 | 0.6493 |
| | Avg | 0.4301 | 0.4505 | 0.4395 | 0.4555 | 0.4235 | 0.4441 | 0.4322 | 0.4513 | **0.3409** | **0.4104** |
| Weather | 96 | 0.1842 | 0.2353 | 0.2066 | 0.2440 | 0.2066 | 0.2427 | 0.2053 | 0.2490 | 0.1736 | 0.2191 |
| | 192 | 0.2286 | 0.2676 | 0.2483 | 0.2824 | 0.2407 | 0.2825 | 0.2474 | 0.2882 | 0.2263 | 0.2636 |
| | 336 | 0.2948 | 0.3153 | 0.3168 | 0.3287 | 0.3132 | 0.3236 | 0.3032 | 0.3209 | 0.2761 | 0.3050 |
| | 720 | 0.3678 | 0.3617 | 0.4195 | 0.3992 | 0.3554 | 0.3517 | 0.3739 | 0.3692 | 0.3536 | 0.3534 |
| | Avg | 0.2689 | 0.2950 | 0.2978 | 0.3136 | 0.2790 | 0.3001 | 0.2825 | 0.3068 | **0.2574** | **0.2853** |

transformations are applied sequentially to each sample, where larger values increase data diversity but may also introduce excessive noise. The latter assigns a probability to directly selecting the raw input, which acts as a regularizer to prevent overfitting to overly augmented samples.

As shown in Figure 8, both hyper-parameters have limited impact on performance across different tasks. Specifically, increasing $k$ yields stable results, with moderate values providing the best trade-off between diversity and reliability. For the raw transform bias, incorporating a small proportion of raw samples consistently stabilizes training and avoids degradation, highlighting the importance of balancing augmented and authentic data. Overall, these results indicate that AutoDA-Timeseries is robust to the choice of hyper-parameters.

In addition, our current implementation initializes the augmentation distribution using a uniform prior with an additional raw-transform bonus. To evaluate the sensitivity of our framework to this initialization, we conduct a controlled study on both classification (ROCKET-based, evaluated with accuracy) and long-term forecasting (RNN-based, evaluated with MSE and MAE) settings under four initial augmentation distributions:

- Uniform distribution + raw-transform bonus (the default setting in our paper);
- Pure uniform distribution;
- Random distribution sampled from Dirichlet($\alpha = 1$), which centers around the uniform distribution with moderate variance;
- Random distribution sampled from Dirichlet($\alpha = 2$), which produces samples closer to uniform but still with variability.

As shown in Tables 11 and 12, the final performance differences remain consistently small across all augmentation distributions. These results demonstrate that our framework is insensitive to the

Table 10: Ablation study on Catch22 feature groups.

| Dataset | Ours | Remove Autocorrelation | Remove Distribution | Remove Differencing |
|---|---|---|---|---|
| AWR | **0.9800** | 0.9400 | 0.9467 | **0.9800** |
| BM | **1** | **1** | **1** | **1** |
| CR | **1** | 0.9861 | 0.9444 | **1** |
| EP | **0.9783** | 0.9275 | 0.9348 | 0.9203 |
| ER | **0.9741** | 0.9185 | 0.9296 | 0.9407 |
| HB | **0.7756** | 0.7610 | 0.7415 | 0.7512 |
| RS | **0.8947** | **0.8947** | 0.8421 | 0.8026 |
| SCP1 | **0.8840** | 0.8464 | 0.8294 | 0.8328 |
| UWGL | **0.9313** | 0.9031 | 0.9125 | 0.9188 |
| Average Accuracy | **0.9353** | 0.9086 | 0.8979 | 0.9052 |

choice of the initial augmentation distribution, and that the learned augmentation policy remains stable and robust regardless of how the distribution is initialized. This indicates that the training dynamics of AutoDA-Timeseries are sufficiently strong to overcome any prior biases introduced at initialization.

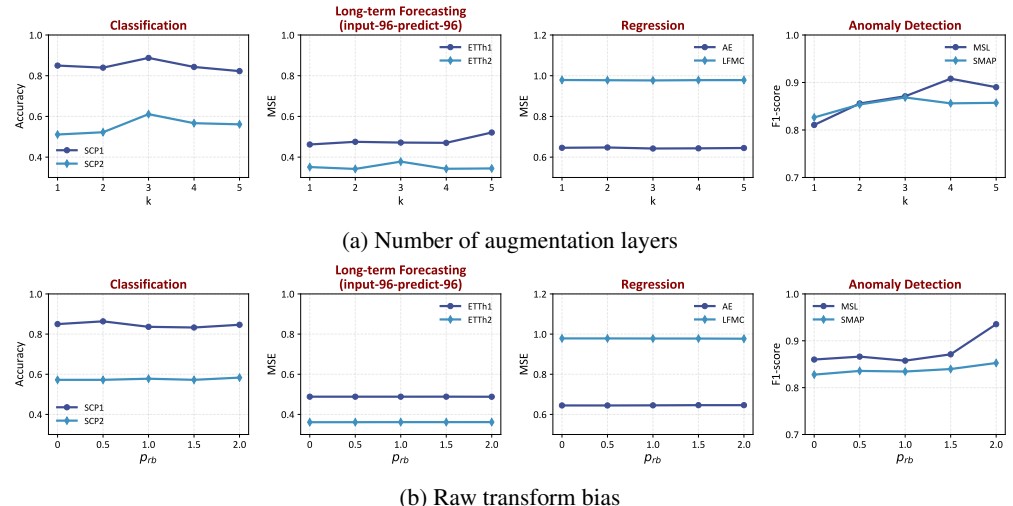

(a) Number of augmentation layers

(b) Raw transform bias

Figure 8: Performance of AutoDA-Timeseries under different hyper-parameter settings across representative tasks.

Table 11: Sensitivity of AutoDA-Timeseries to initial augmentation distributions on classification.

| Dataset | Unifrom + Raw Bonus (Ours) | Uniform Dist. | Random Dist ($\alpha = 1$) | Random Dist ($\alpha = 2$) |
|---|---|---|---|---|
| AWR | **0.9800** | 0.9500 | 0.9733 | 0.9633 |
| BM | **1** | **1** | **1** | **1** |
| CR | **1** | 0.9861 | 0.9861 | **1** |
| EP | **0.9783** | 0.9638 | 0.9420 | 0.9638 |
| ER | **0.9741** | 0.9222 | 0.9556 | 0.9296 |
| HB | **0.7756** | **0.7756** | 0.7415 | 0.7366 |
| RS | 0.8947 | 0.8817 | 0.8882 | **0.9013** |
| SCP1 | 0.8840 | 0.8703 | **0.8980** | 0.8805 |
| UWGL | 0.9313 | 0.9188 | 0.9270 | **0.9345** |
| Average Accuracy | **0.9353** | 0.9187 | 0.9235 | 0.9233 |

# E  GENERALIZATION ACROSS DATASETS

To further examine the generalizability of AutoDA-Timeseries, we conduct transfer experiments across datasets, as summarized in Table 13. Specifically, we train the downstream model together

Table 12: Sensitivity of AutoDA-Timeseries to initial augmentation distributions on long-term forecasting.

| Dataset | Pred_len | Metric | Unifrom + Raw Bonus (Ours) | Uniform Dist. | Random Dist ($\alpha = 1$) | Random Dist ($\alpha = 2$) |
|---|---|---|---|---|---|---|
| ETTh1 | 96 | MSE | **0.4849** | 0.5034 | 0.5098 | 0.5137 |
| | | MAE | **0.4715** | 0.4789 | 0.4795 | 0.4803 |
| | 192 | MSE | 0.5536 | **0.5308** | 0.5387 | 0.5420 |
| | | MAE | 0.5046 | **0.4912** | 0.4951 | 0.4956 |
| | 336 | MSE | **0.5552** | 0.5658 | 0.5728 | 0.5762 |
| | | MAE | **0.4902** | 0.5027 | 0.5065 | 0.5069 |
| | 720 | MSE | 0.5777 | **0.5536** | 0.5692 | 0.5734 |
| | | MAE | 0.5161 | **0.5003** | 0.5135 | 0.5139 |
| ETTh2 | 96 | MSE | **0.3336** | 0.3590 | 0.3559 | 0.3559 |
| | | MAE | **0.3779** | 0.3918 | 0.3914 | 0.3914 |
| | 192 | MSE | **0.4229** | 0.4250 | 0.4273 | 0.4273 |
| | | MAE | **0.4238** | 0.4277 | 0.4320 | 0.4321 |
| | 336 | MSE | **0.4340** | 0.4453 | 0.4597 | 0.4593 |
| | | MAE | **0.4392** | 0.4503 | 0.4582 | 0.4582 |
| | 720 | MSE | **0.4213** | 0.4472 | 0.4546 | 0.4498 |
| | | MAE | **0.4431** | 0.4590 | 0.4591 | 0.4505 |
| ETTm1 | 96 | MSE | **0.4714** | 0.4787 | 0.4761 | 0.4763 |
| | | MAE | **0.4500** | 0.4581 | 0.4556 | 0.4539 |
| | 192 | MSE | 0.5107 | **0.5077** | 0.5081 | 0.5280 |
| | | MAE | 0.4630 | **0.4601** | 0.4735 | 0.4691 |
| | 336 | MSE | 0.5628 | **0.5612** | 0.5615 | 0.6471 |
| | | MAE | 0.4881 | **0.4851** | 0.4897 | 0.5084 |
| | 720 | MSE | 0.6071 | 0.6132 | **0.6023** | 0.6459 |
| | | MAE | 0.5237 | 0.5305 | **0.5188** | 0.5584 |
| ETTm2 | 96 | MSE | **0.2019** | 0.2108 | 0.2156 | 0.2156 |
| | | MAE | 0.2850 | **0.2842** | 0.2896 | 0.2856 |
| | 192 | MSE | **0.2601** | 0.2875 | 0.2827 | 0.2826 |
| | | MAE | **0.3225** | 0.3296 | 0.3265 | 0.3265 |
| | 336 | MSE | **0.3136** | 0.3611 | 0.3480 | 0.3485 |
| | | MAE | **0.3545** | 0.3727 | 0.3635 | 0.3639 |
| | 720 | MSE | **0.4197** | 0.4582 | 0.4328 | 0.4517 |
| | | MAE | **0.4165** | 0.4388 | 0.4358 | 0.4368 |

with augmentation policies on ETTh1 and directly evaluate the trained model on ETTh2 and ETTm2, comparing with NoAug and UniformAugment baselines (the latter is included because it is the second-best method under the RNN downstream model, only inferior to ours). As shown in the upper block (ETTh1 → ETTh2), AutoDA-Timeseries consistently outperforms the baselines across all forecasting horizons, achieving the lowest average MSE and MAE, which demonstrates that the models trained with our framework generalize well to datasets with similar distribution. In the more challenging setting of ETTh1 → ETTm2, where the source and target distributions differ substantially, the performance gap narrows, yet AutoDA-Timeseries remains competitive and clearly superior to UniformAugment. These results highlight that AutoDA-Timeseries not only enhances performance within a single dataset but also exhibits strong potential for cross-dataset generalization, validating its robustness and applicability in real-world scenarios.

## F  MODEL EFFICIENCY

To further evaluate the practicality of AutoDA-Timeseries, we conduct efficiency experiments considering three factors: parameter size, training time (ms/iter), and accuracy. As shown in Figure 9, AutoDA-Timeseries achieves a favorable balance between accuracy and efficiency. Compared with NoAug, AutoDA-Timeseries brings consistent accuracy improvements with only moderate increases in parameter size and training time. Compared with more complex baselines such as AutoTCL, TS2Vec, and A2Aug, AutoDA-Timeseries delivers higher accuracy with significantly lower computational overhead. Although simple augmentation baselines such as UniformAugment exhibit shorter training times, they fail to match the performance of AutoDA-Timeseries. Overall, these

Table 13: Generalization performance of AutoDA-Timeseries on RNN under cross-dataset transfer settings.

| Settings | | NoAug | | UniformAugment | | AutoDA-Timeseries | |
| --- | --- | --- | --- | --- | --- | --- | --- |
| | | MSE | MAE | MSE | MAE | MSE | MAE |
| ETTh1 → ETTh2 | 96 | 0.4761 | 0.4602 | 0.6486 | 0.5577 | 0.4431 | 0.4409 |
| | 192 | 0.5418 | 0.4944 | 0.7172 | 0.5882 | 0.5146 | 0.4788 |
| | 336 | 0.5566 | 0.5116 | 0.7366 | 0.6091 | 0.5374 | 0.4996 |
| | 720 | 0.5416 | 0.5115 | 0.7727 | 0.6315 | 0.5354 | 0.5052 |
| | Avg | 0.5290 | 0.4944 | 0.7188 | 0.5966 | **0.5076** | **0.4811** |
| ETTh1 → ETTm2 | 96 | 0.8668 | 0.6702 | 1.1370 | 0.7578 | 0.8663 | 0.6724 |
| | 192 | 1.0198 | 0.7361 | 1.3273 | 0.8305 | 1.0087 | 0.7338 |
| | 336 | 1.2997 | 0.8369 | 1.6327 | 0.9313 | 1.2791 | 0.8317 |
| | 720 | 1.7294 | 0.9623 | 2.1243 | 1.0623 | 1.7105 | 0.9576 |
| | Avg | 1.2289 | 0.8014 | 1.5553 | 0.8955 | **1.2162** | **0.7989** |

results highlight the advantage of AutoDA-Timeseries in achieving an effective accuracy-efficiency trade-off, demonstrating its practicality for real-world time series applications.

To complete our analysis, we next present a formal analysis of AutoDA-Timeseries's computational complexity. During training, the computational complexity of the augmented model is $O(K \times B \times d \times L)$, and the memory $O(B \times d \times L)$. During inference, our framework does not invoke the augmented model; only the downstream model is used, resulting in zero additional runtime or memory overhead. We provide a detailed derivation below.

The computational cost of AutoDA-Timeseries comes from two components: the policy generator (probability and strength generators) and the augmentation operators in the augmentation set $\mathcal{T}$.

**Time Complexity**. The policy generator takes as input the flattened feature vector in $\mathbb{R}^{C \times d}$ and the probability vector in $\mathbb{R}^n$ from the previous layer. Both are length-independent vectors. Therefore, the cost for this part is $O(B \times (Cd + n))$, where $B$ is the batch size, $d$ is the channel dimension, $C$ is the number of feature dimensions, and $n$ is the size of the augmentation set. This cost does not depend on the sequence length $L$ and is significantly smaller than the cost of applying the augmentation operators.

The dominant cost comes from the augmentation operators. Most transformations used in AutoDA-Timeseries (such as Jittering, Scaling, TimeWarp, and Resample) involve pointwise operations or a single interpolation along the temporal axis. Since the operations are performed over all $L$ time steps and across all $d$ channels, their cost per layer is $O(dL)$. Combining the two parts, the total time complexity of $K$ stacked augmentation layers is $O(K \times B \times d \times L) + O(B \times (Cd + n)) = O(K \times B \times d \times L)$, because the second term is much smaller than the first.

**Memory Complexity**. The memory cost consists of three components.

- The policy generator parameters. The MLP weights have size $O(Cd + n)$, which does not depend on $L$ and is much smaller than the parameter size of downstream models such as Autoformer, VAE, or TCN;

- The intermediate tensors during augmentation. At each layer the model stores the original time series and the augmented time series. The extra memory cost is $O(B \times d \times L)$. Augmentation layers operate sequentially rather than in parallel. Therefore, the total additional memory cost remains $O(B \times d \times L)$, the same as a single layer;

- Downstream model memory, which is shared across all methods and does not affect the relative cost of AutoDA-Timeseries.

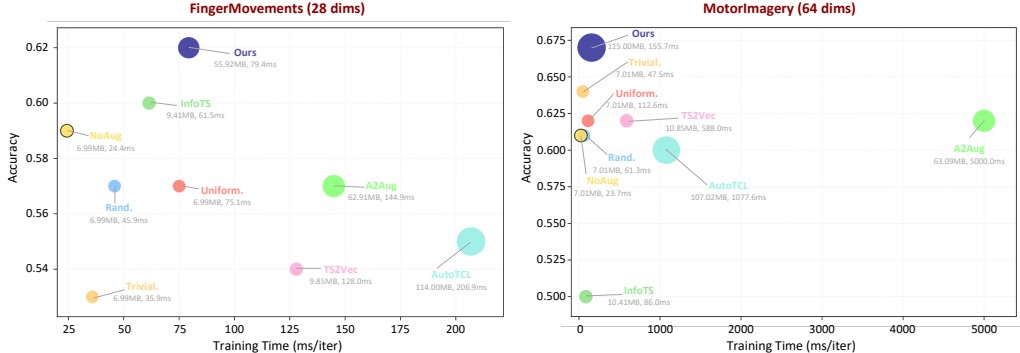

(a) Efficiency comparison on FingerMovements dataset (28 dimensions)

(b) Efficiency comparison on MotorImagery dataset (64 dimensions)

Figure 9: Model efficiency comparison across datasets. The x-axis represents training time per iteration, the y-axis shows accuracy, and the bubble size reflects model parameter size.

## G   ROBUSTNESS UNDER LIMITED TRAINING SAMPLES

In practical scenarios, obtaining sufficient labeled training samples is often challenging. A data augmentation method that maintains strong performance under limited samples demonstrates better generalization under limited training data (Wen et al., 2020). To evaluate this property, we varied the training data ratio from 10%, 30%, 50%, 70%, to 100% of the original training set, and compared the performance of NoAug with AutoDA-Timeseries. We selected three representative tasks, including classification, long-term forecasting, and anomaly detection. The model architectures and hyperparameters were kept fixed, and we reported Accuracy, MSE, and F1-score for each task.

As shown in Figure 10, AutoDA-Timeseries consistently outperforms NoAug under different fractions of training data. The advantage is particularly evident in low-data regimes, where augmentation substantially narrows the performance gap caused by limited supervision. Even when more data are available, AutoDA-Timeseries remains competitive, indicating that learned augmentation strategies not only alleviate data scarcity but also enhance robustness across varying data scales. This finding suggests that AutoDA-Timeseries is not merely a remedy for data scarcity but a general mechanism to enhance model generalization in diverse real-world scenarios.

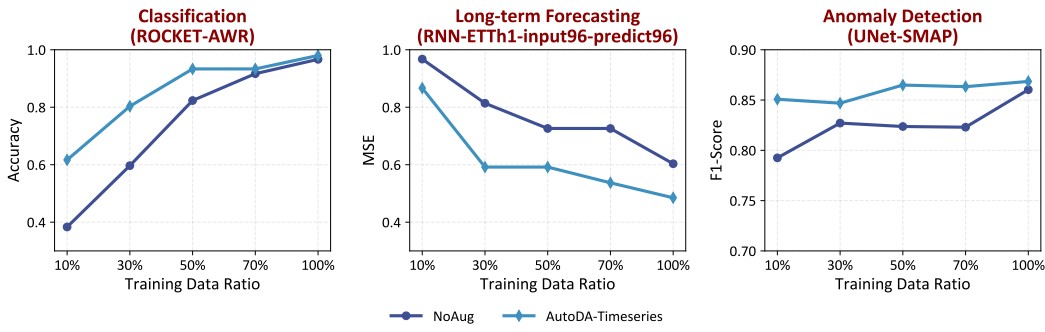

Figure 10: Performance comparison between NoAug and AutoDA-Timeseries under varying training data ratios (10%, 30%, 50%, 70%, 100%) across three representative tasks: classification (Accuracy), long-term forecasting (MSE), and anomaly detection (F1-score).

## H   WEIGHT DISTRIBUTION ANALYSIS

To further understand the effect of AutoDA-Timeseries on downstream model training, we analyze the weight distributions of models trained with and without augmentation, as they provide a

compact characterization of model stability and generalization. Figure 11 presents kernel density estimates of model parameters across five representative tasks, including classification (SelfRegulationSCP2), long-term forecasting (ETTh1), short-term forecasting (M4), regression (FloodModeling2), and anomaly detection (MSL). The distributions remain largely consistent in shape and centered around zero, indicating that AutoDA-Timeseries does not introduce abnormal parameter shifts or bias. Meanwhile, the five tasks exhibit distinct distributional patterns: ETTh1 and FloodModeling2 show narrow and almost sparse distributions, while M4 presents wider tails that reflect higher complexity. These results demonstrate that AutoDA-Timeseries adapts effectively to diverse scenarios while preserving distributional stability.

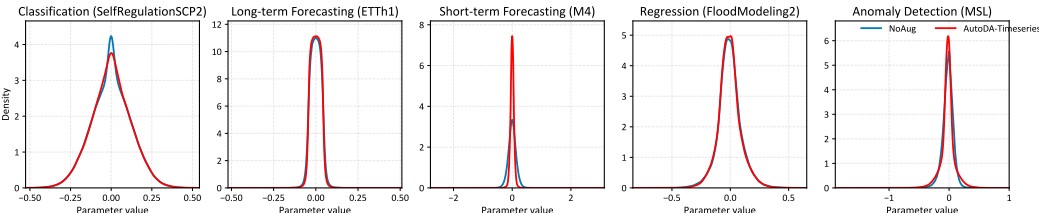

Figure 11: Comparison of weight distributions between models trained without augmentation (blue, NoAug) and with AutoDA-Timeseries (red) across five tasks, demonstrating that AutoDA-Timeseries preserves stable parameter distributions while adapting to task-specific characteristics.

## I    PRIVACY ANALYSIS

To assess whether releasing augmented datasets could expose sensitive information from the original time series, we conducted statistical attack experiments to compare privacy vulnerability by uniform (employed in RandAugment, UniformAugment, and TrivialAugment) or biased (employed in AutoDA-Timeseries) augmentation selections. We try to reconstruct the original time series from an augmented time series dataset generated from a set of augmentation transforms applied to the original seed time series, and evaluate the privacy vulnerability by the RMSE between the ground truth original time series and the reconstructed time series.

As detailed in Appendix I.1, a more deterministic reconstruction can be performed with known equal probabilities of augmentation, while in contrast, reconstruction with unknown probabilities of augmentation has to be modeled as a mixture-model estimation.

As shown in Table 14, four groups of reconstruction are performed for comparison. To ensure fairness, $G1$, $G3$, and $G4$ utilize the same augmented time series dataset, and we control the $G3$ and $G4$ to iterate with the same time consumption. Due to the context limitation, more details can be found in Appendix I.2.

The results are presented in Figure 12. First, $G1$ and $G4$, which simulate reconstructing from a dataset generated by previous SOTA AutoDA frameworks, demonstrate a lower RMSE and time consumption than $G2$ and $G3$. This indicates the risk to data privacy when releasing augmented datasets with a fixed uniform augmentation policy. Second, the accuracy difference between $G3$ and $G4$ shows that the estimation of seed data can be easily misled when the augmentation probabilities are also jointly estimated for a mixture model estimation, proving the effectiveness for augmenting the time series without a fixed augmentation probability. Last, the RMSEs in $G2$ are higher than $G3$ and $G4$ with the same estimation model, indicating that the non-uniform augmentation probability in augmentation policy does increase the difficulty of reconstructing the seed data.

### I.1    RECONSTRUCT A SINGLE TIME SERIES FROM AUGMENTED TIME SERIES

This section discusses how to reconstruct the seed time series from augmented time series data based on a seed time series and a set of augmentation transformations $\mathcal{T} = \{T_j, j = 1, 2, \ldots, n\}$ when randomly sampling augmentation transforms and intensities.

Denote the original time series as $c$. Suppose the probability of selecting augmentation transform $T_i$ is $p_i$, the distribution of the augmented time series generated by $T_i$ is $Y_i$, and the distribution of the

Table 14: Reconstruction experiment group setup. AugProbs is whether the augmentation probabilities are equal, and ProbDist is whether this probability distribution is fixed and known to the attacker.

| Group | Estimation | ProbDist | AugProbs |
|---|---|---|---|
| Group 1 ($G1$) | Deterministic | Fixed | Uniform |
| Group 2 ($G2$) | Mixture-model | Unfixed | Non-uniform |
| Group 3 ($G3$) | Mixture-model | Unfixed | Uniform |
| Group 4 ($G4$) | Mixture-model | Fixed | Uniform |

(a) Reconstruction RMSE  (b) Estimation time

Figure 12: Reconstruction RMSE and time consumption to reconstruct the original time series.

entire generated dataset is $X_g$. Then:

$$E(X_g) = \sum p_i E(Y_i(c)),$$

where $E(X_g)$ is precisely the weighted mean expectation of all time series in the generated dataset, denoted $mean(X_g) = \mu_g$. Next, the variance is given by:

$$Var(X_g) = E\big(Var(X_g \mid p_i)\big) + Var\big(E(X_g \mid p_i)\big),$$

where

$$E\big(Var(X_g \mid p_i)\big) = \sum p_i\, Var\big(Y_i(c)\big)$$

represents the weighted mean variance of all subsets generated by different augmentation transforms.

In previous AutoDA frameworks (Cubuk et al., 2020; Müller & Hutter, 2021; LingChen et al., 2020), the transformation operators are predefined and fixed. Consequently, the distributions $Y_i$ can be easily derived apart from an unknown intensity range parameter $t$. Therefore, the distribution of $X_g$ is determined by the original seed data $c$, the intensity range parameter $t$, and the probability distribution $\{p_i\}$. These three can be viewed as the prior for $X_g$ and hence can be estimated with the observed samples of $X_g$, which correspond exactly to the time series in the augmented dataset. As a result, if the augmentation transforms are selected with equal probabilities, $t$ can be easily estimated, and the seed data $c$ can be reconstructed accordingly, jeopardizing data privacy.

For illustration, consider a toy example with a specific seed time series $c$ and an augmentation transform set comprising three transformations:

- *Raw* transform: $Y_1(c) = c$
- *Scaling* transform: $Y_2(c) = c \cdot s$, where the scaling factor $s$ follows a uniform distribution $s \sim U[2t-1, 2t+1]$
- *Jittering* transform: $Y_3(c) = c + n$, where the noise $n$ follows a Gaussian distribution $n \sim \mathcal{N}(0, t^2)$

The expectation and variance of each subset are then:

$$E(Y_1) = c,$$
$$Var(Y_1) = 0,$$
$$E(Y_2) = 2t \cdot c,$$
$$Var(Y_2) = 4t^2 c^2,$$
$$E(Y_3) = c,$$
$$Var(Y_3) = t^2.$$

Hence, if the augmentation transforms are chosen with equal probability $p_1 = p_2 = p_3 = \frac{1}{3}$, the expectation and variance of the entire augmented dataset are:

$$E(X_g) = \frac{c + 2tc + c}{3} = \frac{2t+2}{3}c,$$

$$Var(X_g) = (0 + \tfrac{1}{3}c^2 + t^2)/3 + (c^2 + 4t^2c^2 + c^2)/3 - \left(\frac{2t+2}{3}c\right)^2 = \frac{1}{3}t^2 + \frac{8t^2 - 8t + 3}{9}c^2.$$

Since the average and variance of the augmented dataset can be computed easily, $t$ can be estimated, and subsequently $c$ can be inferred. By contrast, if the probability of augmentation selection is not equal, the model forms a mixture model, making estimation significantly more complex.

**Abstractly**, when the selection probabilities are not necessarily equal, one must re-estimate the prior from observations of the distribution involving $\{p_i\}$, $c$, and $t$. However, when the selection probabilities are assumed to be equal, $X_g$ reduces to a distribution that contains only the unknown priors $c$ and $t$, which substantially reduces the difficulty of accurate prior estimation.

### I.2 RECONSTRUCTION EXPERIMENT SETTINGS

Given a predefined set of augmentation transformations, we apply these transformations to time series in the original dataset. Two types of datasets are generated with different strategies:

- The generated time series data of all different transformations are directly mixed into the dataset $\mathcal{D}_1$ with equal probability.
- The generated time series data of all transformations are mixed into a synthetic dataset $\mathcal{D}_2$ according to a given probability vector.

From $\mathcal{D}_1$ and $\mathcal{D}_2$, the original seed metrics as prior parameters are estimated. For $\mathcal{D}_1$, we use Newton's method to estimate the intensity range parameters and the original seed metrics, denoted as *Group1*. For $\mathcal{D}_2$, since the probability vector prior is unknown, it formulates a mixture model estimation. Thus, the Expectation-Maximization (EM) algorithm is applied to estimate the probability and the prior parameters of the corresponding distribution iteratively. We generated a dataset with unevenly sampled transformations and performed EM (denoted as *Group2*).

To ensure fairness in comparison, we also established a comparison group of applying EM on $\mathcal{D}_1$, learning the probability on its own (denoted as *Group3*), or estimating with a fixed probability (denoted as *Group4*).

In the experiment, specific formal modifications have been made to some augmentation transformations to unify the problem form and accelerate the calculation. For example, the *Raw* transformation is replaced with a *Jittering* transformation with a minimal Gaussian noise. In addition, we have performed standard normalization on the original time series in advance to avoid the problem of inconsistent scales.

## J FULL RESULTS

To provide a complete view of the experiment outcomes, we report the detailed results of all downstream models across different tasks. Specifically, the classification results using TCN and ROCKET are presented in Tables 15 and 16, while the long-term forecasting results with RNN and Autoformer are summarized in Tables 17 and 18. For regression, we present the detailed results of CNN and MLP in Tables 19 and 20. Finally, the anomaly detection results with UNet and VAE are provided in Tables 21 and 22.

## K SHOWCASES

### K.1 SHOWCASE OF FORECASTING CASES

To provide an intuitive understanding of how different augmentation strategies influence forecasting performance, we present case studies on the ETTh1 dataset with a horizon of 96 steps, where the downstream model is RNN. As shown in Figure 13, the predictions from models trained with AutoDA-Timeseries better capture the temporal dynamics and align more closely with the ground truth compared to those from other baselines.

### K.2 SHOWCASE OF AUGMENTATION CASES

We visualize augmentation cases on the SCP1 dataset to provide qualitative insights. Figure 14 and Figure 15 present two different samples, each showing the evolution of augmented time series across

Table 15: Detailed classification results with TCN across baselines and AutoDA-Timeseries. "∗." in the method names denotes ∗Augment.

| Datasets / Methods | NoAug | InfoTS (2023) | AutoTCL (2024) | TS2Vec (2022) | Rand. (2020) | Uniform. (2020) | Trivial. (2021) | A2Aug (2023) | **Ours** |
|---|---|---|---|---|---|---|---|---|---|
| AWR | 0.8933 | 0.9767 | 0.9800 | 0.9367 | 0.9133 | 0.9100 | 0.9433 | **0.9833** | 0.9533 |
| AF | 0.3333 | 0.3333 | **0.4667** | 0.3333 | 0.4000 | **0.4667** | 0.3333 | 0.3333 | **0.4667** |
| BM | 1 | 1 | 1 | 0.5000 | 1 | 1 | 1 | 1 | 1 |
| CR | 0.9861 | 0.9861 | 0.9583 | 0.9583 | 0.9772 | 0.9028 | 1 | 0.9861 | 1 |
| DDG | 0.7200 | 0.5600 | 0.6000 | 0.2800 | **0.7400** | 0.7000 | 0.7000 | 0.6000 | 0.7200 |
| EW | 0.8168 | 0.8015 | 0.8168 | 0.7939 | 0.8321 | 0.6718 | 0.7634 | 0.8092 | **0.8702** |
| EP | 0.9783 | 0.9348 | 0.9420 | 0.9420 | 0.9783 | 0.9420 | 0.9783 | 0.9855 | 1 |
| ER | 0.7593 | 0.9185 | 0.8963 | 0.1667 | 0.8778 | 0.8815 | 0.8778 | 0.9037 | **0.9222** |
| EC | 0.4030 | 0.2548 | 0.2890 | 0.3080 | 0.3004 | 0.2776 | 0.3118 | 0.3156 | **0.4068** |
| FD | 0.5000 | **0.6302** | 0.5499 | 0.5182 | 0.5000 | 0.5006 | 0.5000 | 0.5000 | 0.5000 |
| FM | 0.5900 | 0.6000 | 0.5500 | 0.5400 | 0.5700 | 0.5700 | 0.5300 | 0.5700 | **0.6200** |
| HMD | **0.4730** | **0.4730** | 0.4324 | 0.1758 | 0.4189 | 0.4054 | 0.4324 | 0.4595 | **0.4730** |
| HW | 0.5847 | 0.3647 | 0.4600 | 0.2753 | 0.5588 | 0.0812 | 0.6118 | **0.6671** | 0.4918 |
| HB | **0.7854** | 0.7610 | 0.7512 | 0.7317 | 0.7659 | 0.7512 | 0.7756 | 0.7756 | **0.7854** |
| LIB | 0.8222 | 0.8278 | 0.6667 | 0.7222 | 0.7667 | 0.1389 | 0.8222 | **0.9111** | 0.8500 |
| LSST | 0.3990 | 0.6310 | 0.5114 | 0.6196 | 0.4185 | 0.3491 | 0.4091 | **0.6403** | 0.4124 |
| MI | 0.6100 | 0.5000 | 0.6000 | 0.6200 | 0.6100 | 0.6200 | 0.6400 | 0.6200 | **0.6700** |
| NATOPS | 0.8333 | **0.9389** | 0.8389 | 0.8944 | 0.8500 | 0.8444 | 0.8389 | 0.8334 | 0.8889 |
| PEMS-SF | 0.8324 | 0.7861 | 0.6821 | 0.5491 | 0.7977 | 0.4046 | 0.7514 | **0.8728** | 0.8497 |
| PD | 0.8645 | 0.9237 | 0.9423 | 0.9140 | 0.9525 | 0.8716 | 0.9580 | 0.8971 | **0.9634** |
| PS | 0.2320 | **0.2741** | 0.0954 | 0.1700 | 0.1497 | 0.0790 | 0.1968 | 0.2103 | 0.1867 |
| RS | 0.9079 | 0.8882 | 0.8950 | 0.7566 | 0.9145 | 0.8882 | 0.9013 | 0.9211 | **0.9408** |
| SCP1 | 0.8396 | 0.8703 | 0.8700 | 0.8567 | 0.8396 | 0.8601 | 0.8805 | 0.8669 | **0.8874** |
| SCP2 | 0.5389 | 0.5667 | 0.5667 | 0.4611 | 0.5889 | 0.5667 | 0.5500 | 0.5611 | **0.6111** |
| SWJ | 0.3333 | 0.4667 | 0.4667 | 0.3333 | 0.4667 | 0.4000 | 0.4667 | 0.4000 | **0.7333** |
| UWGL | 0.7656 | **0.9094** | 0.8840 | 0.8156 | 0.8531 | 0.7063 | 0.8061 | 0.8219 | 0.7875 |
| Average Accuracy | 0.6847 | 0.6991 | 0.6812 | 0.5836 | 0.6939 | 0.6073 | 0.6915 | 0.7094 | **0.7304** |

three layers. The results indicate that the augmentation process preserves the global structure while introducing diverse variations, demonstrating the effectiveness of AutoDA-Timeseries in generating meaningful augmented data.

Table 16: Detailed classification results with ROCKET across baselines and AutoDA-Timeseries. "∗." in the method names denotes ∗Augment.

| Datasets / Methods | NoAug | InfoTS (2023) | AutoTCL (2024) | TS2Vec (2022) | Rand. (2020) | Uniform. (2020) | Trivial. (2021) | A2Aug (2023) | Ours |
|---|---|---|---|---|---|---|---|---|---|
| AWR | 0.9667 | 0.9767 | 0.9800 | 0.9833 | 0.9567 | 0.9433 | 0.9733 | **0.9900** | 0.9800 |
| AF | 0.4000 | **0.5333** | **0.5333** | 0.4000 | 0.4000 | 0.3333 | 0.4000 | 0.4000 | 0.4667 |
| BM | **1** | 0.7750 | **1** | **1** | **1** | **1** | **1** | **1** | **1** |
| CR | 0.9583 | 0.8889 | 0.8889 | 0.5972 | 0.9306 | 0.8334 | 0.9861 | 0.9861 | **1** |
| DDG | 0.6600 | **0.7000** | **0.7000** | 0.2600 | 0.6600 | 0.6200 | 0.5800 | **0.7000** | **0.7000** |
| EW | 0.6107 | 0.6565 | 0.6183 | 0.5954 | 0.6031 | 0.5649 | 0.6565 | 0.7252 | **0.7328** |
| EP | 0.9638 | 0.5725 | 0.9275 | 0.8551 | 0.9058 | 0.7971 | 0.9565 | 0.9420 | **0.9783** |
| ER | 0.9444 | 0.9593 | 0.9556 | 0.8 | 0.9296 | 0.9037 | 0.9222 | 0.9556 | **0.9741** |
| EC | 0.2928 | 0.4297 | **0.4373** | 0.4297 | 0.2928 | 0.2852 | 0.3042 | 0.2548 | 0.3156 |
| FD | 0.6200 | 0.6393 | 0.6348 | 0.5497 | 0.6379 | 0.6266 | 0.6263 | **0.6510** | 0.6328 |
| FM | 0.5900 | 0.6300 | 0.6200 | 0.6000 | 0.6300 | 0.5800 | 0.6100 | 0.6100 | **0.6500** |
| HMD | 0.5270 | 0.5135 | 0.5405 | 0.1351 | 0.5135 | 0.5000 | **0.5541** | 0.5000 | **0.5541** |
| HW | 0.3600 | 0.2200 | 0.2212 | 0.1600 | 0.3047 | 0.1141 | 0.3447 | **0.4800** | 0.3588 |
| HB | 0.7610 | 0.7415 | 0.7366 | 0.6341 | **0.7805** | 0.7512 | 0.7561 | 0.7659 | 0.7756 |
| LIB | 0.6889 | 0.8500 | **0.8556** | 0.7056 | 0.5833 | 0.3500 | 0.6389 | 0.8222 | 0.7167 |
| LSST | 0.6006 | 0.3978 | 0.5016 | 0.6156 | 0.5921 | 0.5393 | 0.6123 | **0.6415** | 0.5933 |
| MI | 0.5800 | 0.5500 | 0.5800 | 0.6100 | 0.5600 | 0.5500 | 0.5600 | 0.5500 | **0.6500** |
| NATOPS | **0.9167** | 0.9000 | 0.9056 | 0.8833 | 0.8889 | 0.8278 | **0.9167** | **0.9167** | **0.9167** |
| PEMS-SF | 0.5376 | **0.7919** | 0.7746 | 0.3584 | 0.3873 | 0.1792 | 0.4682 | 0.6301 | 0.5607 |
| PD | 0.9634 | 0.9663 | 0.9696 | 0.9574 | 0.9520 | 0.9180 | 0.9634 | 0.9691 | **0.9711** |
| PS | 0.1837 | 0.1062 | 0.1118 | 0.1288 | 0.1697 | 0.1184 | 0.1828 | **0.2120** | 0.1670 |
| RS | 0.8750 | 0.7434 | 0.8816 | 0.7895 | 0.8421 | 0.8289 | 0.8421 | 0.8684 | **0.8947** |
| SCP1 | 0.8737 | 0.7406 | 0.7372 | 0.5290 | 0.8771 | 0.8532 | 0.8771 | 0.8567 | **0.8840** |
| SCP2 | 0.5500 | 0.4778 | 0.4833 | 0.5278 | 0.5389 | 0.5389 | 0.5278 | 0.5556 | **0.6111** |
| SWJ | 0.5333 | 0.4667 | 0.4667 | 0.3333 | **0.7333** | 0.5333 | 0.5333 | 0.4000 | **0.7333** |
| UWGL | 0.8688 | 0.8406 | 0.8438 | 0.8875 | 0.8844 | 0.8594 | 0.8906 | 0.9156 | **0.9313** |
| Average Accuracy | 0.6856 | 0.6630 | 0.6887 | 0.5895 | 0.6752 | 0.6134 | 0.6801 | 0.7038 | **0.7211** |

Table 17: Detailed long-term forecasting results with RNN across baselines and AutoDA-Timeseries. "∗." in the method names denotes ∗Augment.

| Methods | | NoAug | | InfoTS (2023) | | AutoTCL (2024) | | TS2Vec (2022) | | Rand. (2020) | | Uniform. (2020) | | Trivial. (2021) | | A2Aug (2023) | | Ours | |
|---|---|---|---|---|---|---|---|---|---|---|---|---|---|---|---|---|---|---|---|
| Metrics | | MSE | MAE | MSE | MAE | MSE | MAE | MSE | MAE | MSE | MAE | MSE | MAE | MSE | MAE | MSE | MAE | MSE | MAE |
| ETTh1 | 96 | 0.6034 | 0.5234 | 0.9196 | 0.7253 | 0.8481 | 0.6904 | 0.6950 | 0.6141 | 0.5067 | 0.4882 | 0.5044 | 0.4809 | 0.6222 | 0.5465 | 0.6221 | 0.5614 | **0.4849** | **0.4715** |
| | 192 | 0.6314 | 0.5394 | 0.9829 | 0.7631 | 1.0001 | 0.7744 | 0.8538 | 0.6969 | 0.5392 | 0.5053 | 0.5362 | 0.4978 | 0.6615 | 0.5670 | 0.8087 | 0.6374 | 0.5536 | 0.5046 |
| | 336 | 0.5591 | 0.4909 | 1.0161 | 0.7772 | 1.1044 | 0.8068 | 1.0128 | 0.7856 | 0.6173 | 0.5278 | 0.5642 | 0.5090 | 0.6996 | 0.5821 | 0.6959 | 0.6165 | 0.5552 | 0.4902 |
| | 720 | 0.6498 | 0.5550 | 1.1145 | 0.8243 | 1.1164 | 0.8273 | 1.1999 | 0.8308 | 0.6226 | 0.5462 | 0.6596 | 0.5694 | 0.7020 | 0.5962 | 0.7665 | 0.6735 | 0.5777 | 0.5161 |
| | Avg | 0.6109 | 0.6134 | 1.0083 | 1.0378 | 1.0173 | 1.0736 | 0.9404 | 1.0222 | 0.5715 | 0.5930 | 0.5661 | 0.5867 | 0.6713 | 0.6877 | 0.7233 | 0.7570 | **0.5429** | **0.5622** |
| ETTh2 | 96 | 0.4103 | 0.4206 | 1.3967 | 0.9731 | 1.5150 | 1.0029 | 1.0807 | 0.8338 | 0.4822 | 0.4594 | 0.3918 | 0.4034 | 0.5275 | 0.4727 | 0.7067 | 0.5417 | **0.3336** | **0.3779** |
| | 192 | 0.6240 | 0.5461 | 1.8304 | 1.1280 | 2.7557 | 1.3260 | 1.7323 | 1.0532 | 0.5981 | 0.5093 | 0.4651 | 0.4430 | 0.6047 | 0.5068 | 1.2000 | 0.6587 | **0.4229** | **0.4238** |
| | 336 | 0.7327 | 0.5923 | 2.3618 | 1.3095 | 2.3520 | 1.1934 | 1.9501 | 1.1481 | 0.6260 | 0.5289 | 0.4991 | 0.4695 | 0.5782 | 0.5110 | 1.1345 | 0.6889 | **0.4340** | **0.4392** |
| | 720 | 0.7419 | 0.5981 | 3.3260 | 1.5914 | 2.2807 | 1.2913 | 3.4461 | 1.5999 | 0.6262 | 0.5084 | 0.5255 | 0.4877 | 0.5408 | 0.4980 | 0.7155 | 0.6156 | **0.4213** | **0.4431** |
| | Avg | 0.6272 | 0.6995 | 2.2287 | 2.5061 | 2.2807 | 2.5359 | 2.0523 | 2.3762 | 0.5669 | 0.5952 | 0.4704 | 0.4966 | 0.5628 | 0.5746 | 0.9392 | 1.0167 | **0.4030** | **0.4261** |
| ETTm1 | 96 | 0.7152 | 0.5315 | 0.7686 | 0.6304 | 0.7387 | 0.6327 | 0.5347 | 0.5128 | 0.7454 | 0.5465 | 0.5396 | 0.4725 | 0.6846 | 0.5271 | 0.5489 | 0.5058 | **0.4714** | **0.4500** |
| | 192 | 0.7856 | 0.5521 | 0.8374 | 0.6752 | 0.7922 | 0.6620 | 0.6322 | 0.5636 | 0.5981 | 0.5621 | 0.4844 | 0.7650 | 0.5537 | 0.5858 | 0.5216 | **0.5107** | **0.4630** |
| | 336 | 0.8251 | 0.5704 | 0.8773 | 0.6974 | 0.8543 | 0.6969 | 0.7580 | 0.6367 | 0.9121 | 0.6033 | 0.5681 | 0.4923 | 0.8330 | 0.5821 | 0.7301 | 0.5872 | **0.5628** | **0.4881** |
| | 720 | 0.8740 | 0.5959 | 0.9345 | 0.7272 | 0.9600 | 0.7524 | 0.8575 | 0.6967 | 0.9820 | 0.6351 | **0.6040** | **0.5106** | 0.9094 | 0.6201 | 0.6301 | 0.5535 | 0.6071 | 0.5237 |
| | Avg | 0.8000 | 0.8282 | 0.8545 | 0.8831 | 0.8363 | 0.8688 | 0.6956 | 0.7492 | 0.8714 | 0.9134 | 0.5685 | 0.5781 | 0.7980 | 0.8358 | 0.6237 | 0.6487 | **0.5380** | **0.5602** |
| ETTm2 | 96 | 0.2648 | 0.3390 | 0.4430 | 0.5272 | 0.9089 | 0.7701 | 0.8549 | 0.7113 | 0.2383 | 0.3200 | 0.2188 | 0.3017 | 0.2263 | 0.3087 | 0.3262 | 0.3857 | **0.2019** | **0.2850** |
| | 192 | 0.3512 | 0.3868 | 0.7770 | 0.7042 | 0.9957 | 0.8069 | 1.2348 | 0.9199 | 0.3011 | 0.3559 | 0.2769 | 0.3365 | 0.2857 | 0.3426 | 0.4879 | 0.4859 | **0.2601** | **0.3225** |
| | 336 | 0.4352 | 0.4307 | 1.4088 | 0.9832 | 1.0756 | 0.8514 | 1.4640 | 1.0058 | 0.4257 | 0.4249 | 0.3392 | 0.3733 | 0.3398 | 0.3738 | 0.6555 | 0.5522 | **0.3136** | **0.3545** |
| | 720 | 0.5422 | 0.4829 | 2.5102 | 1.3234 | 1.8328 | 1.1009 | 2.4442 | 1.3223 | 0.5411 | 0.4820 | 0.5460 | 0.4654 | **0.4193** | **0.4166** | 0.8572 | 0.6174 | 0.4197 | 0.4165 |
| | Avg | 0.3984 | 0.4429 | 1.2848 | 1.5653 | 1.2033 | 1.3014 | 1.4995 | 1.7143 | 0.3452 | 0.3874 | 0.3178 | 0.3483 | 0.5817 | 0.6669 | **0.2988** | **0.3311** |
| Exchange | 96 | 0.1687 | 0.2995 | 1.7382 | 1.0084 | 1.9485 | 1.1527 | 1.8184 | 1.0493 | 0.1540 | 0.2756 | 0.1540 | 0.2833 | 0.1572 | 0.2813 | 0.3931 | 0.4316 | **0.1086** | **0.2328** |
| | 192 | 0.2726 | 0.3835 | 1.8373 | 1.0733 | 2.1392 | 1.1981 | 2.0228 | 1.1680 | 0.2700 | 0.3676 | 0.2453 | 0.3629 | 0.2643 | 0.3693 | 0.4089 | 0.4511 | **0.2049** | **0.3234** |
| | 336 | 0.4378 | 0.4931 | 2.2536 | 1.2062 | 2.2780 | 1.2362 | 0.4031 | 0.4623 | 0.3826 | 0.4607 | 0.4401 | 0.4838 | 1.0091 | 0.7111 | **0.3582** | **0.4360** |
| | 720 | 1.0198 | 0.7766 | 2.7453 | 1.3081 | 2.6330 | 1.2828 | 2.4236 | 1.2374 | 0.8142 | 0.6828 | 0.9486 | 0.7415 | 1.0740 | 0.7864 | **0.6692** | **0.6393** | 0.6920 | 0.6493 |
| | Avg | 0.4747 | 0.5767 | 2.1436 | 2.2787 | 2.2660 | 2.3718 | 2.1357 | 2.2415 | 0.4103 | 0.4958 | 0.4326 | 0.5255 | 0.4839 | 0.5928 | 0.6201 | 0.6957 | **0.3409** | **0.4184** |
| Weather | 96 | 0.2561 | 0.2801 | 1.2829 | 0.8239 | 1.0355 | 0.7405 | 0.6114 | 0.5761 | 0.2005 | 0.2452 | 0.1856 | 0.2355 | 0.1948 | 0.2383 | 0.1997 | 0.2492 | **0.1736** | **0.2191** |
| | 192 | 0.3021 | 0.3154 | 1.3174 | 0.8486 | 1.0782 | 0.7719 | 0.7302 | 0.6208 | 0.2432 | 0.2792 | 0.2342 | 0.2740 | 0.2414 | 0.2786 | 0.2802 | 0.3134 | **0.2263** | **0.2636** |
| | 336 | 0.3516 | 0.3464 | 1.7156 | 1.0304 | 1.4528 | 0.9040 | 1.1253 | 0.8158 | 0.2885 | 0.3099 | 0.2858 | 0.3081 | 0.2996 | 0.3185 | 0.3481 | 0.3841 | **0.2761** | **0.3050** |
| | 720 | 0.4254 | 0.3916 | 1.9970 | 1.1336 | 1.7517 | 1.0259 | 1.4813 | 0.9688 | 0.3558 | 0.3524 | 0.3613 | 0.3554 | 0.3926 | 0.3765 | 0.4400 | 0.4494 | **0.3536** | **0.3534** |
| | Avg | 0.3338 | 0.3597 | 1.5782 | 1.6767 | 1.3296 | 1.4276 | 0.9871 | 1.1123 | 0.2720 | 0.2958 | 0.2667 | 0.2938 | 0.2821 | 0.3112 | 0.3170 | 0.3561 | **0.2574** | **0.2853** |
| 1st Count | | 0 | | 0 | | 0 | | 0 | | 1 | | 4 | | 1 | | 2 | | 52 | |

Table 18: Detailed long-term forecasting results with Autoformer across baselines and AutoDA-Timeseries. "∗." in the method names denotes ∗Augment.

| Methods | | NoAug | | InfoTS (2023) | | AutoTCL (2024) | | TS2Vec (2022) | | Rand. (2020) | | Uniform. (2020) | | Trivial. (2021) | | A2Aug (2023) | | Ours | |
|---|---|---|---|---|---|---|---|---|---|---|---|---|---|---|---|---|---|---|---|
| Metrics | | MSE | MAE | MSE | MAE | MSE | MAE | MSE | MAE | MSE | MAE | MSE | MAE | MSE | MAE | MSE | MAE | MSE | MAE |
| ETTh1 | 96 | 1.0263 | 0.7891 | 0.9841 | 0.7953 | 1.0033 | 0.8193 | 0.8845 | 0.7419 | 1.0260 | 0.7941 | 1.0410 | 0.8222 | 1.2331 | 0.8827 | 0.9523 | 0.7763 | 0.8732 | 0.7458 |
| | 192 | 0.9639 | 0.7761 | 0.9752 | 0.8126 | 0.9453 | 0.7937 | 0.8865 | 0.7404 | 1.0308 | 0.7904 | 1.0458 | 0.8234 | 1.1358 | 0.8145 | 0.9878 | 0.7885 | 0.9008 | 0.7534 |
| | 336 | 1.0260 | 0.8071 | 0.9689 | 0.8024 | 0.9465 | 0.7928 | 0.9277 | 0.7409 | 1.0334 | 0.7899 | 1.0530 | 0.8263 | 1.2993 | 0.8882 | 0.9864 | 0.7791 | 1.0216 | 0.8191 |
| | 720 | 0.9688 | 0.7799 | 0.9452 | 0.7815 | 0.9263 | 0.7664 | 1.0308 | 0.8376 | 1.0245 | 0.7850 | 1.0575 | 0.8270 | 1.2908 | 0.8784 | 0.9392 | 0.7763 | 1.0215 | 0.8154 |
| | Avg | 0.9963 | 0.9862 | 0.9684 | 0.9631 | 0.9554 | 0.9394 | 0.9324 | 0.9483 | 1.0287 | 1.0296 | 1.0493 | 1.0521 | 1.2398 | 1.2420 | 0.9664 | 0.9711 | 0.9543 | 0.9813 |
| ETTh2 | 96 | 3.8442 | 1.5563 | 3.1187 | 1.3570 | 3.0869 | 1.3503 | 3.0069 | 1.3180 | 3.2213 | 1.4085 | 3.2230 | 1.4388 | 2.8329 | 1.3110 | 2.8021 | 1.2940 | 2.4034 | 1.2302 |
| | 192 | 2.7073 | 1.2526 | 3.1001 | 1.3430 | 3.1326 | 1.3536 | 3.2592 | 1.3796 | 3.2799 | 1.4174 | 3.3293 | 1.4499 | 3.2522 | 1.3950 | 2.9939 | 1.3380 | 2.7126 | 1.3442 |
| | 336 | 0.9961 | 0.7931 | 3.0804 | 1.3340 | 3.1086 | 1.3462 | 2.9728 | 1.3322 | 3.2731 | 1.4067 | 3.2295 | 1.4515 | 3.2216 | 1.3838 | 3.2867 | 1.4342 | 2.7465 | 1.3848 |
| | 720 | 2.5324 | 1.2130 | 3.0378 | 1.3246 | 3.0566 | 1.3329 | 3.1065 | 1.3330 | 3.2630 | 1.4038 | 3.2516 | 1.4560 | 3.0298 | 1.4151 | 3.2314 | 1.4569 | 2.8374 | 1.4446 |
| | Avg | 2.5200 | 2.0786 | 3.0843 | 3.0728 | 3.0962 | 3.0993 | 3.0864 | 3.1128 | 3.2593 | 3.2720 | 3.2334 | 3.2368 | 3.0841 | 3.1679 | 3.0785 | 3.1707 | 2.6750 | 2.7655 |
| ETTm1 | 96 | 1.8310 | 1.1192 | 1.0957 | 0.7893 | 1.1103 | 0.8027 | 0.8148 | 0.7252 | 1.2186 | 0.8663 | 1.2275 | 0.8792 | 1.2325 | 0.8709 | 1.0236 | 0.7758 | 0.8689 | 0.7072 |
| | 192 | 1.7354 | 1.0828 | 1.1138 | 0.8536 | 1.1029 | 0.7935 | 0.8867 | 0.7623 | 1.2134 | 0.8631 | 1.2192 | 0.8771 | 1.2252 | 0.8678 | 1.0293 | 0.7854 | 1.1311 | 0.8390 |
| | 336 | 1.6885 | 1.0605 | 1.1021 | 0.7957 | 1.1519 | 0.8530 | 0.8935 | 0.7615 | 1.2129 | 0.8622 | 1.2157 | 0.8765 | 1.2237 | 0.8674 | 0.9422 | 0.7738 | 1.0529 | 0.8140 |
| | 720 | 1.6893 | 1.0515 | 1.0674 | 0.8193 | 1.1063 | 0.7960 | 0.9424 | 0.7811 | 1.2135 | 0.8637 | 1.2177 | 0.8791 | 1.2294 | 0.8729 | 1.1114 | 0.8154 | 1.2272 | 0.8876 |
| | Avg | 1.7361 | 1.7044 | 1.0948 | 1.0944 | 1.1179 | 1.1204 | 0.8844 | 0.9075 | 1.2146 | 1.2133 | 1.2200 | 1.2175 | 1.2277 | 1.2261 | 1.0266 | 1.0276 | 1.0700 | 1.1371 |
| ETTm2 | 96 | 2.7817 | 1.3155 | 3.1061 | 1.3551 | 3.0206 | 1.4202 | 2.3535 | 1.3076 | 2.6729 | 1.2963 | 2.8379 | 1.3706 | 3.5723 | 1.4780 | 2.8887 | 1.3317 | 2.5533 | 1.2498 |
| | 192 | 3.6055 | 1.5007 | 3.1309 | 1.3587 | 3.1359 | 1.3587 | 2.5970 | 1.3592 | 3.1856 | 1.4181 | 3.2365 | 1.4555 | 3.3879 | 1.4786 | 3.1428 | 1.3996 | 2.7021 | 1.3589 |
| | 336 | 4.0774 | 1.6337 | 3.1818 | 1.3692 | 3.1447 | 1.3679 | 2.2067 | 1.1345 | 3.1680 | 1.4233 | 3.3635 | 1.4706 | 3.3197 | 1.4834 | 3.2642 | 1.3716 | 2.9326 | 1.3434 |
| | 720 | 3.0671 | 1.4349 | 3.1932 | 1.3696 | 3.0466 | 1.3929 | 3.3414 | 1.4279 | 2.9395 | 1.4353 | 3.3176 | 1.4655 | 3.2869 | 1.4884 | 3.1796 | 1.4394 | 2.6495 | 1.2698 |
| | Avg | 3.3829 | 3.5833 | 3.1530 | 3.1686 | 3.0870 | 3.1091 | 2.6247 | 2.7150 | 2.9915 | 3.0977 | 3.1889 | 3.3059 | 3.3917 | 3.3315 | 3.1188 | 3.1955 | 2.7094 | 2.7614 |
| Exchange | 96 | 3.0696 | 1.4659 | 4.7716 | 1.7676 | 4.7654 | 1.8026 | 4.7650 | 1.7686 | 2.5649 | 1.2911 | 2.2821 | 1.2414 | 1.5506 | 1.0461 | 2.5712 | 1.3783 | 1.4143 | 0.9761 |
| | 192 | 2.0449 | 1.1978 | 4.7517 | 1.7609 | 4.5271 | 1.5486 | 4.6947 | 1.7496 | 2.0318 | 1.1312 | 2.9111 | 1.4173 | 2.2307 | 1.2602 | 2.0573 | 1.2145 | 1.4095 | 0.9194 |
| | 336 | 1.8875 | 1.1624 | 4.7606 | 1.7500 | 4.7626 | 1.7528 | 4.7386 | 1.7508 | 1.9907 | 1.1883 | 2.6509 | 1.3539 | 2.2047 | 1.1812 | 1.6807 | 1.0388 | 1.5898 | 0.9914 |
| | 720 | 2.7725 | 1.3706 | 4.7999 | 1.7619 | 4.8315 | 1.7682 | 4.8138 | 1.7665 | 2.9641 | 1.3662 | 3.0918 | 1.4682 | 2.2418 | 1.2271 | 2.3173 | 1.2722 | 1.6965 | 1.0897 |
| | Avg | 2.4436 | 2.2350 | 4.7710 | 4.7707 | 4.4717 | 4.3737 | 4.7530 | 4.7490 | 2.3879 | 2.3289 | 2.7340 | 2.8846 | 2.0570 | 2.2257 | 2.1566 | 2.0184 | 1.5275 | 1.5653 |
| Weather | 96 | 3.6475 | 1.5226 | 0.4995 | 0.5151 | 0.6277 | 0.6023 | 0.3700 | 0.4085 | 3.7655 | 1.5601 | 3.6130 | 1.5279 | 1.3235 | 1.4159 | 1.6439 | 1.0193 | 1.2919 | 0.9329 |
| | 192 | 3.1727 | 1.4315 | 0.6173 | 0.5979 | 0.4497 | 0.4468 | 0.4087 | 0.4189 | 3.4958 | 1.5027 | 3.6112 | 1.5176 | 3.5880 | 1.5015 | 1.7755 | 1.0565 | 2.9921 | 1.3855 |
| | 336 | 3.5969 | 1.5274 | 0.6121 | 0.5925 | 0.6172 | 0.5957 | 0.4556 | 0.4541 | 3.5772 | 1.5229 | 3.4884 | 1.5027 | 3.3572 | 1.4611 | 1.7180 | 1.0404 | 2.7707 | 1.3381 |
| | 720 | 3.5249 | 1.5348 | 0.6123 | 0.5940 | 0.4358 | 0.4500 | 0.6198 | 0.5466 | 3.3654 | 1.4862 | 3.8626 | 1.5746 | 2.9488 | 1.3717 | 1.8466 | 1.0787 | 3.0353 | 1.4080 |
| | Avg | 3.4855 | 3.4315 | 0.5853 | 0.6139 | 0.5326 | 0.5009 | 0.4635 | 0.4947 | 3.5510 | 3.4795 | 3.6438 | 3.6541 | 3.2544 | 3.2980 | 1.7460 | 1.7800 | 2.5225 | 2.9327 |
| 1st Count | | 6 | | 1 | | 6 | | 29 | | 0 | | 0 | | 0 | | 0 | | 17 | |

Table 19: Detailed regression results with CNN across baselines and AutoDA-Timeseries. "∗." in the method names denotes ∗Augment.

| Datasets | Metrics | NoAug | InfoTS (2023) | AutoTCL (2024) | TS2Vec (2022) | Rand. (2020) | Uniform. (2020) | Trivial. (2021) | A2Aug (2023) | Ours |
|---|---|---|---|---|---|---|---|---|---|---|
| AE | MSE | 0.6424 | 0.6463 | 0.6461 | 0.6457 | 0.6424 | 0.6424 | 0.6425 | 0.6458 | 0.6423 |
| | MAE | 0.6347 | 0.6465 | 0.6461 | 0.6458 | 0.6361 | 0.6349 | 0.6362 | 0.6458 | 0.6375 |
| FM1 | MSE | 0.7370 | 0.7965 | 0.7414 | 0.7982 | 0.7370 | 0.7390 | 0.7387 | 4.8129 | 0.6602 |
| | MAE | 0.6480 | 0.6473 | 0.6507 | 0.6591 | 0.6528 | 0.6555 | 0.6539 | 0.8405 | 0.6264 |
| FM2 | MSE | 0.7813 | 0.5699 | 2.1929 | 0.9154 | 0.5273 | 3.1468 | 0.4536 | 0.4685 | 0.3875 |
| | MAE | 0.3195 | 0.4305 | 0.4185 | 0.3684 | 0.2547 | 0.4643 | 0.2879 | 0.2652 | 0.2204 |
| FM3 | MSE | 0.8647 | 1.4001 | 1.4228 | 1.3215 | 1.2162 | 1.5773 | 0.8622 | 1.1040 | 1.2221 |
| | MAE | 0.6891 | 0.8537 | 0.8423 | 0.7890 | 0.8296 | 0.9066 | 0.7082 | 0.7535 | 0.8040 |
| LFMC | MSE | 0.9789 | 0.9790 | 0.9786 | 0.9791 | 1.7238 | 0.9789 | 0.9789 | 0.9789 | 0.9768 |
| | MAE | 0.7544 | 0.7541 | 0.7494 | 0.7563 | 1.0769 | 0.7537 | 0.7544 | 0.7541 | 0.7484 |
| IEEEPPG | MSE | 1.5666 | 1.7569 | 1.7577 | 1.8753 | 1.7238 | 1.7439 | 1.6492 | 1.5992 | 1.4636 |
| | MAE | 1.0466 | 1.0993 | 1.0990 | 1.1079 | 1.0769 | 1.0709 | 1.0480 | 1.0371 | 1.0018 |
| Avg MSE | | 0.9285 | 1.0248 | 1.2899 | 1.0892 | 1.0951 | 1.4714 | 0.8875 | 1.6016 | 0.8921 |
| Avg MAE | | 0.6821 | 0.7386 | 0.7343 | 0.7211 | 0.7545 | 0.7477 | 0.6814 | 0.7160 | 0.6731 |
| 1st Count | | 2 | 0 | 0 | 0 | 0 | 0 | 2 | 0 | 10 |

Table 20: Detailed regression results with MLP across baselines and AutoDA-Timeseries. "∗." in the method names denotes ∗Augment.

| Datasets | Metrics | NoAug | InfoTS (2023) | AutoTCL (2024) | TS2Vec (2022) | Rand. (2020) | Uniform. (2020) | Trivial. (2021) | A2Aug (2023) | Ours |
|---|---|---|---|---|---|---|---|---|---|---|
| AE | MSE | 0.6433 | 0.6425 | 0.6424 | 0.6435 | 0.6438 | 0.6425 | 0.6425 | 0.6438 | **0.6415** |
|  | MAE | 0.6331 | 0.6318 | 0.6325 | **0.6298** | 0.6406 | 0.6335 | 0.6371 | 0.634 | 0.6354 |
| FM1 | MSE | **0.2787** | 0.6555 | 0.6619 | 0.6332 | 0.3529 | 0.7047 | 0.3384 | 0.4155 | 0.2788 |
|  | MAE | 0.4062 | 0.5960 | 0.6148 | 0.6018 | 0.4774 | 0.6518 | 0.4455 | 0.4733 | **0.3885** |
| FM2 | MSE | 3.1369 | 3.0971 | 3.0911 | 2.7806 | 2.3691 | 2.9632 | 2.9757 | 2.4156 | **1.7873** |
|  | MAE | 0.5184 | 0.5029 | 0.4051 | 0.4494 | **0.3179** | 0.4729 | 0.4364 | 0.3604 | 0.3223 |
| FM3 | MSE | 0.8488 | 1.2171 | 1.2412 | 1.2197 | 1.0149 | 1.2935 | 0.8554 | 0.9558 | **0.7675** |
|  | MAE | 0.7416 | 0.8065 | 0.8744 | 0.8433 | 0.7046 | 0.8217 | 0.6666 | 0.6534 | **0.6530** |
| LFMC | MSE | 0.9790 | 0.9789 | 0.9789 | 0.9789 | 0.9790 | 0.9789 | 0.9790 | 0.9789 | **0.9673** |
|  | MAE | 0.7530 | 0.7546 | 0.7545 | 0.7546 | 0.7528 | 0.7544 | 0.7533 | 0.7543 | **0.7505** |
| IEEEPPG | MSE | 1.8752 | 1.8306 | 1.9025 | 1.8085 | 1.9581 | 1.8361 | 1.8551 | 1.8846 | **1.7675** |
|  | MAE | 1.1534 | 1.1195 | 1.1273 | 1.1131 | 1.1239 | 1.1256 | **1.0984** | 1.1157 | 1.1022 |
| Avg MSE |  | 1.2937 | 1.4036 | 1.4197 | 1.3441 | 1.2196 | 1.4032 | 1.2744 | 1.2157 | **1.0350** |
| Avg MAE |  | 0.7010 | 0.7352 | 0.7348 | 0.7320 | 0.6695 | 0.7433 | 0.6729 | 0.6652 | **0.6420** |
| 1st **Count** |  | 1 | 0 | 0 | 1 | 1 | 0 | 1 | 0 | **10** |

Table 21: Detailed anomaly detection results with UNet across baselines and AutoDA-Timeseries. "∗." in the method names denotes ∗Augment.

| Datasets | | MSL | | | SMAP | | | SMD | | | Avg F1 |
|---|---|---|---|---|---|---|---|---|---|---|---|
| Metrics | | P | R | F1 | P | R | F1 | P | R | F1 | |
| NoAug | | 0.6215 | 0.9475 | 0.7506 | 0.7734 | 0.9692 | 0.8603 | 0.3290 | 0.9323 | 0.4864 | 0.6991 |
| InfoTS | (2023) | 0.6226 | 0.9475 | 0.7515 | 0.7734 | 0.9646 | 0.8585 | 0.3207 | 0.8371 | 0.4637 | 0.6912 |
| AutoTCL | (2024) | 0.6287 | 0.9458 | 0.7553 | 0.7677 | 0.9350 | 0.8431 | 0.3279 | 0.9297 | 0.4848 | 0.6944 |
| TS2Vec | (2022) | 0.618 | 0.9387 | 0.7453 | 0.6856 | 0.5722 | 0.6238 | 0.3268 | 0.9236 | 0.4828 | 0.6173 |
| Rand. | (2020) | 0.6283 | 0.9436 | 0.7544 | 0.7714 | 0.8971 | 0.8295 | 0.3176 | 0.898 | 0.4692 | 0.6844 |
| Uniform. | (2020) | 0.7144 | 0.9884 | 0.8293 | 0.7841 | 0.9392 | 0.8547 | 0.3249 | 0.8318 | 0.4673 | 0.7171 |
| Trivial. | (2021) | 0.6207 | 0.9448 | 0.7492 | 0.7679 | 0.9347 | 0.8431 | 0.3203 | 0.9080 | 0.4736 | 0.6886 |
| A2Aug | (2023) | 0.6217 | 0.9475 | 0.7508 | 0.7743 | 0.9737 | 0.8626 | 0.3279 | 0.9279 | 0.4846 | 0.6993 |
| **Ours** | | 0.7772 | 0.9906 | **0.8710** | 0.7888 | 0.9661 | **0.8685** | 0.3491 | 0.9045 | **0.5038** | **0.7478** |

Table 22: Detailed anomaly detection results with VAE across baselines and AutoDA-Timeseries. "∗." in the method names denotes ∗Augment.

| Datasets | | MSL | | | SMAP | | | SMD | | | Avg F1 |
|---|---|---|---|---|---|---|---|---|---|---|---|
| Metrics | | P | R | F1 | P | R | F1 | P | R | F1 | |
| NoAug | | 0.9015 | 0.4041 | 0.5581 | 0.9717 | 0.8652 | 0.9153 | 0.1507 | 0.3159 | 0.2041 | 0.5592 |
| InfoTS | (2023) | 0.9026 | 0.3962 | 0.5507 | 0.9948 | 0.5557 | 0.7131 | 0.1491 | 0.3137 | 0.2022 | 0.4887 |
| AutoTCL | (2024) | 0.9012 | 0.3894 | 0.5438 | 0.9948 | 0.5558 | 0.7132 | 0.1509 | 0.3158 | 0.2042 | 0.4871 |
| TS2Vec | (2022) | 0.9084 | 0.4017 | 0.5571 | 0.9948 | 0.5558 | 0.7132 | 0.1508 | 0.3154 | 0.2040 | 0.4914 |
| Rand. | (2020) | 0.9021 | 0.4151 | 0.5685 | 0.9863 | 0.8447 | 0.9100 | 0.1512 | 0.3163 | 0.2046 | 0.5610 |
| Uniform. | (2020) | 0.9043 | 0.4203 | 0.5739 | 0.9949 | 0.5559 | 0.7133 | 0.1512 | 0.3164 | 0.2046 | 0.4973 |
| Trivial. | (2021) | 0.9011 | 0.4125 | 0.5660 | 0.9948 | 0.5558 | 0.7132 | 0.1509 | 0.3160 | 0.2043 | 0.4945 |
| A2Aug | (2023) | 0.9001 | 0.4102 | 0.5635 | 0.9905 | 0.8393 | 0.9087 | 0.1517 | 0.3168 | 0.2052 | 0.5591 |
| **Ours** | | 0.9032 | 0.4224 | **0.5756** | 0.9731 | 0.9225 | **0.9471** | 0.1521 | 0.3172 | **0.2056** | **0.5761** |

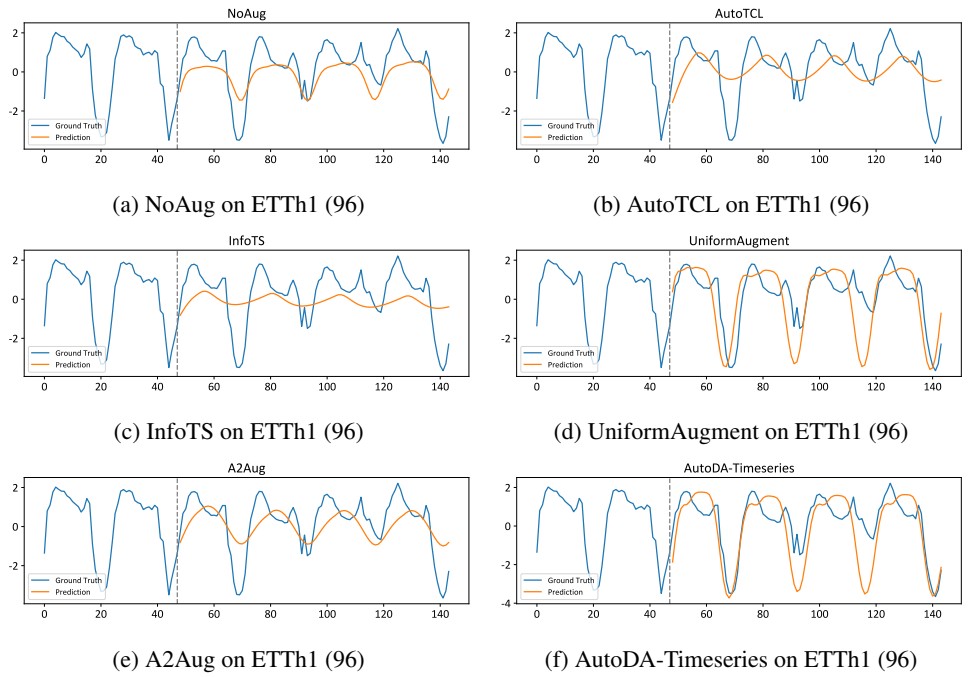

Figure 13: Forecasting showcase on ETTh1 dataset with horizon 96 using RNN as the downstream model.

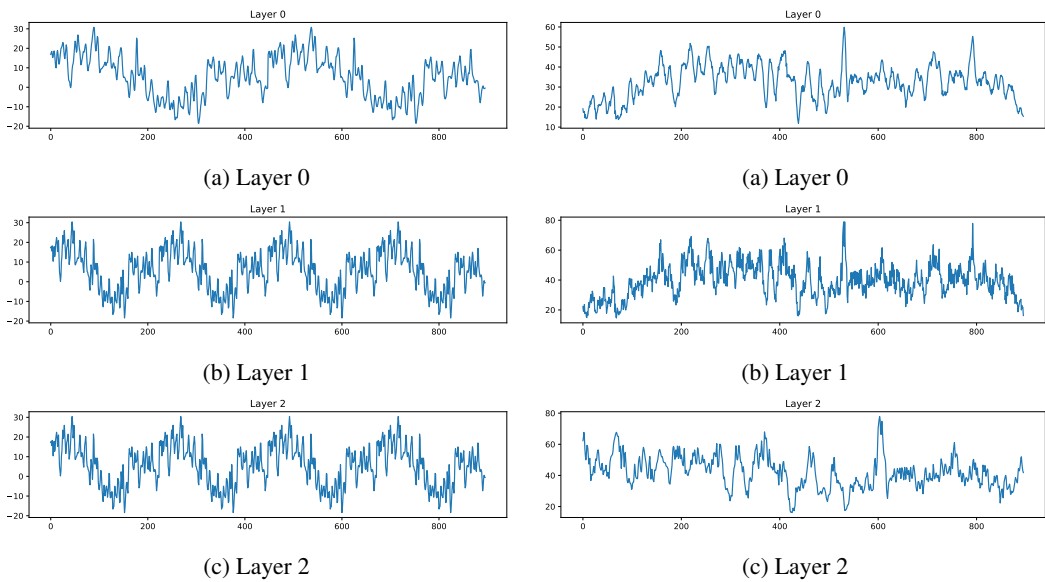

Figure 14: SCP1 augmentation showcase 1 across three layers.

Figure 15: SCP1 augmentation showcase 2 across three layers.

