# OpenReview forum: "AutoDA-Timeseries: Automated Data Augmentation for Time Series"
_ICLR.cc/2026/Conference — ICLR 2026 Poster_

### Official Review · Reviewer_juTF · 2025-10-22

**Soundness:** 2
**Presentation:** 2
**Contribution:** 2
**Rating:** 6
**Confidence:** 3

**Summary:**

The paper introduces AutoDA-Timeseries, a novel framework addressing the specific needs of automated data augmentation for time series analysis, a domain where existing methods often fall short.   Key strengths include its general-purpose design applicable across five diverse tasks, the incorporation of time series features to guide policy generation, and extensive empirical validation showing consistent improvements over baselines. However, the theoretical justification for the controller objective is limited, and the link between learned policies and data characteristics is not well-analyzed. Moreover, some mathematical notations are ambiguous, and the computational cost of the search process is not discussed.

**Strengths:**

1. Clear Problem Motivation: Section 1–2 convincingly describes the challenge of limited labeled time-series data and the difficulty of designing effective augmentations. The introduction connects to prior works, positioning AutoDA as a general extension for unsupervised or supervised forecasting.

2. Comprehensive Experimental Evaluation: The framework's effectiveness is demonstrated across five distinct and mainstream time series tasks: classification, long-term forecasting, short-term forecasting, regression, and anomaly detection. This wide range of tasks supports the claim of a "general-purpose" framework.  Evaluation spans numerous benchmark datasets relevant to each task (e.g., 26 UEA subsets for classification, ETT/M4/etc. for forecasting). This provides robust evidence for the method's performance across different data domains and characteristics.   AutoDA-Timeseries is tested with diverse downstream model architectures for each task (e.g., TCN/ROCKET for classification, RNN/Autoformer for forecasting, CNN/MLP for regression, UNet/VAE for anomaly detection), demonstrating its compatibility and effectiveness across different modeling paradigms.   Comparisons are made against relevant baselines, including no augmentation, state-of-the-art representation learning methods (InfoTS, AutoTCL, TS2Vec), and recent AutoDA methods (RandAugment, UniformAugment, TrivialAugment, A2Aug), providing a strong context for evaluating performance gains.

3. Scalable and Modular Design: The framework can plug into different backbone architectures, indicating extensibility. Such modularity makes AutoDA useful for practitioners exploring augmentation strategies under resource constraints.

**Weaknesses:**

1. Lack of Theoretical Justification for Objective:
- The description of the composite loss (Sec 3.5.2) could be more detailed. While $L_2$ (entropy for intra-batch diversity) is standard, the motivation and formulation for $L_3$ (KL divergence for inter-batch diversity, Eq 10 [cite: 5386]) could be explained more clearly. The notation $p_{i}^{(current)}$ vs $p_{b,j}^{(prev)}$ also seems inconsistent regarding batch indices ($i$ vs $b$).
- The notation for probability and intensity generation in Section 3.4  is slightly ambiguous. It states $p_{i,j}^{(k)}=f_{p}^{(k)}(p_{i,j}^{(k-1)},F_{i})$. Does this mean the probability for transform $j$ only depends on its own probability from the previous layer, $p_{i,j}^{(k-1)}$, or should the input be the entire previous probability vector, $p_{i}^{(k-1)}$?

2. Insufficient Detail on Baseline Adaptation and Implementation:
- The paper compares against several AutoDA methods originally developed for images (RandAugment, UniformAugment, TrivialAugment, A2Aug). Appendix B provides brief descriptions, but critically lacks detail on *how* these were adapted for time series. Without this, the comparison might not be entirely fair, as these methods might be suboptimal simply due to naive application.
- The implementation details for the representation learning baselines (InfoTS, AutoTCL, TS2Vec) are also minimal. Clarity on this is needed for reproducibility and fair comparison.
- How to optimize the framework is unclear. There are only two equations for the objective function. The final loss function is also missing. In equation 8, the definition of $L_z$  is missing.

3. Ablation Study:
- The paper omits a formal analysis of the computational complexity (both time and memory) of the proposed AutoDA-Timeseries framework. This makes it difficult to assess how the method's training and inference times scale with factors like time series length, dataset size, number of augmentation layers ($K$), or the size of the transformation pool ($n$).
- The absence of a detailed parameter analysis (e.g., total parameters introduced by the augmentation generator $A_{\theta}$ relative to the downstream model) also obscures the model's overall size and potential impact on training stability or overfitting.

**Questions:**

- What is the computational complexity of one forward pass through k augmentation layers relative to sequence length L and dimension d?
- How sensitive are results to the learnable loss weights wz; do they collapse to zero (diversity ignored) for some tasks?
- What specific time series augmentation transformations were included in the set $\mathcal{T}$? How many transformations ($n$) were used?
- How was the number of stacked augmentation layers $K$ determined? The hyperparameter study (Fig 8a) shows sensitivity analysis, but what was the value used in the main experiments, and how was it chosen (e.g., validation set)?
- Was the learnable temperature for Gumbel-Softmax shared across layers, or did each layer have its own? Was a specific annealing schedule used, or was it learned purely via backpropagation?

---

> ### Author Response · Authors · 2025-11-23
> **Author Response (Part 1)**
>
> We thank Reviewer juTF for the positive and constructive feedback. Below we provide our detailed responses to the reviewer’s comments.
>
> > W1.1: (1) The description of the composite loss (Sec 3.5.2) could be more detailed. While $L_2$ (entropy for intra-batch diversity) is standard, (2) the motivation and formulation for $L_3$ (KL divergence for inter-batch diversity, Eq 10 [cite: 5386]) could be explained more clearly. (3) The notation $p_i^{(current)}$ vs $p_{b, j}^{(prev)}$ also seems inconsistent regarding batch indices ( $i$ vs $b$ ).
>
>
> Thank you for the helpful suggestion.
>
> (1) We provide a more detailed explanation of the composite loss and have added a detailed description in Section 3.5.2.
>
> As shown in Equation (8), our composite loss is defined as
> $$
> L_{\text{composite}} = \sum_{z=1, 2, 3}[\frac{1}{2w_z^2}L_z+\ln(1+w_z^2)],
> $$
> where each component $L_z$ corresponds to a specific objective, and $w_z^2$ is its learnable weight.
>
> - When $z=1$, $L_1$ denotes the task-specific loss (e.g., MSE for forecasting or cross-entropy for classification), with $w_1^2$ controlling its relative importance.
> - When $z=2$, $L_2$ is the entropy-based intra-layer diversity loss (Equations 9 and 10), with $w_2^2$ serving as its learnable weight.
> - When $z=3$, $L_3$ is the inter-layer KL divergence loss (Equation 11), with $w_3^2$ as its learnable weight.
>
>
> (2) While $L_2$ effectively encourages diversity within each layer, it does not prevent the augmentation probabilities from gradually converging to the same deterministic strategy across different layers during training. For example, even if each layer maintains some internal diversity at early stages, the model may still collapse toward a fixed augmentation strategy shared by all layers as training progresses. This leads to a loss of exploration and reduces the adaptiveness of the learned augmentation policy.
>
> To address this issue, we introduce the inter-layer KL term $L_3$, which measures the divergence between the augmentation probability distribution of the current layer and that of the previous layer. This term explicitly encourages diversity in the sequence of augmentation strategies, enabling the model to maintain exploration over training iterations and preventing global collapse.
>
> (3) Thank you for pointing this out. We have unified the notation in the revised Section 3.5.2.
>
>
> > W1.2: The notation for probability and intensity generation in Section 3.4 is slightly ambiguous. It states $p_{i, j}^{(k)} = f_p^{(k)}(p_{i, j}^{(k-1)}, F_i)$ . Does this mean the probability for transform $j$ only depends on its own probability from the previous layer, $p_{i, j}^{(k-1)}$, or should the input be the entire previous probability vector, $p_i^{(k-1)}$?
>
> Thank you for pointing this out. The intended input is the entire previous probability vector $p_i^{(k-1)}$. We have corrected the notation in the revised manuscript.
>
>
> Equation (4) should be written as:
> $$
> p_{i,j}^{(k)} = f_p^{(k)}(p_i^{(k-1)}, \mathbf{F}_i)
> $$
> where $p_i^{(k-1)}$ denotes the entire probability vector from the previous augmentation layer, and $\mathbf{F}_i$ represents the feature vector of time series $\mathbf{D}_i$.
>
> These two vectors are concatenated and then passed into the probability generator $f_p^{(k)}$ to generate $p_{i,j}^{(k)}$.
>
> Thus, the probability update depends on the full previous probability vector, not on each transform independently.

---

> > ### Author Response · Authors · 2025-11-23
> > **Author Response (Part 2)**
> >
> > > W2.1: The paper compares against several AutoDA methods originally developed for images (RandAugment, UniformAugment, TrivialAugment, A2Aug). Appendix B provides brief descriptions, but critically lacks detail on how these were adapted for time series. Without this, the comparison might not be entirely fair, as these methods might be suboptimal simply due to naive application.
> >
> > Thank you for the reviewer’s constructive comment. We have added complete details of the adaptation in Appendix B.
> >
> > We did not naively apply these methods. Instead, we performed a rigorous time-series–specific adaptation of each method. Specifically, we made three categories of modifications:
> > - **Replacing image operations with time-series transformations**. The original RandAugment/TrivialAugment families rely on image operations such as rotation, shear, and color jitter, which are not meaningful for time series. To ensure fairness, we replaced their augmentation set with standard time-series transformations such as jittering, scaling, time-warping, etc. This guarantees that all baselines and our method use the same valid augmentation set.
> > - **Preserving each method’s original sampling logic**. We strictly retained the core augmentation-selection mechanisms of each method: RandAugment preserves its $N$ random operations + global magnitude $M$ formulation. UniformAugment uniformly samples both operations and magnitudes. TrivialAugment samples one operation and magnitude per sample, following its original design. A2Aug learns augmentation weights jointly and ensembles operator logits adaptively.
> > - **Ensuring identical downstream settings for all baselines**. For fair comparison, all baselines use the same downstream models (RNN, Autoformer, etc.), the same data splits, sequence lengths, and batch sizes as AutoDA-Timeseries.
> >
> >
> > > W2.2: The implementation details for the representation learning baselines (InfoTS, AutoTCL, TS2Vec) are also minimal. Clarity on this is needed for reproducibility and fair comparison.
> >
> > Thank you for highlighting the importance of reproducibility and fair comparison. We have added complete implementation details in Appendix B.
> >
> > First, all representation learning methods (TS2Vec, InfoTS, AutoTCL) are implemented using their official open-source repositories. We strictly follow their default hyperparameter configurations, including the number of training epochs, batch size, optimizer settings, and the built-in augmentation pipeline. We do not modify any internal architectural components or training procedures. This ensures that the results are fully reproducible and not influenced by implementation choices on our side.
> >
> > Second, unlike image-based AutoDA baselines, time-series representation learning methods already include augmentation operators specifically designed for sequential data. To ensure fairness, we preserve the exact augmentation transformations defined in their official codebases. This prevents any methodological bias that might arise from altering or replacing their augmentation primitives.
> >
> > Finally, to guarantee fairness in downstream evaluation, all baselines adopt the same downstream configuration used in AutoDA-Timeseries. The downstream model architecture is kept identical across all methods, and every method is evaluated under the same data splits. For representation learning baselines, we follow their standard protocol: the encoder is first pretrained, and then frozen during downstream training while only the prediction head is optimized.
> >
> > > W2.3: How to optimize the framework is unclear. There are only two equations for the objective function. The final loss function is also missing. In equation 8, the definition of $L_z$ is missing.
> >
> > Thank you for the question. We have revised Section 3.5.2 accordingly.
> >
> > Equation (8) in the paper is the final composite loss used for training the entire framework (augmented model and downstream model):
> > $$
> > L_{\text{composite}} = \sum_{z=1,2,3} [\frac{1}{2w_z^2}L_z + \ln(1+w_z^2)]
> > $$
> > Each component $L_z$ corresponds to a specific objective, and its scope is as follows:
> > - When $z = 1$, $L_1$ is the task-specific loss. The exact form depends on the downstream task (for example, MSE for forecasting or cross-entropy for classification). The parameter $w_1^2$ is a learnable weight.
> > - When $z = 2$, $L_2$ is the entropy-based intra-layer diversity loss, defined in Equation (9). This term encourages within-layer diversity in the augmentation choices. Its corresponding learnable weight is $w_2^2$.
> > - When $z = 3$, $L_3$ is the KL-based inter-layer diversity loss, defined in Equation (11). This term penalizes the collapse of augmentation strategies across layers. The corresponding learnable weight is $w_3^2$.
> >
> > The composite loss above is the complete objective used during training. All parameters of the augmented data generator and the downstream model are optimized jointly through backpropagation based on this loss.

---

> > > ### Author Response · Authors · 2025-11-23
> > > **Author Response (Part 3)**
> > >
> > > > W3.1: The paper omits a formal analysis of the computational complexity (both time and memory) of the proposed AutoDA-Timeseries framework. This makes it difficult to assess how the method's training and inference times scale with factors like time series length, dataset size, number of augmentation layers ($K$), or the size of the transformation pool ($n$).
> > >
> > > Thank you for insightfull feedback and suggestions.
> > >
> > > During training, the computational complexity of the augmented model is $O(K\times B\times d\times L)$, and the memory $O(B\times d\times L)$.
> > > During inference, our framework does **not** invoke the augmented model. Only the downstream model is used, so there is zero additional runtime or memory cost introduced by AutoDA-Timeseries at inference time.
> > > We provide a detailed derivation below.
> > >
> > > The computational cost of AutoDA-Timeseries comes from two components: the policy generator (probability and strength generators) and the augmentation operators in the augmentation set $\mathcal{T}$.
> > >
> > > **Time Complexity**. The policy generator takes as input the flattened feature vector in $\mathbb{R}^{C \times d}$ and the probability vector in $\mathbb{R}^{n}$ from the previous layer. Both are length-independent vectors. Therefore, the cost for this part is $O\big(B \times (Cd + n)\big)$, where $B$ is the dataset size, $d$ is the channel dimension, $C$ is the number of feature dimensions, and $n$ is the size of the augmentation set. This cost does not depend on the sequence length $L$ and is significantly smaller than the cost of applying the augmentation operators.
> > >
> > > The dominant cost comes from the augmentation operators. Most transformations used in AutoDA-Timeseries (such as Jittering, Scaling, TimeWarp, and Resample) involve pointwise operations or a single interpolation along the temporal axis. Since the operations are performed over all $L$ time steps and across all $d$ channels, their cost per layer is $O(dL)$.
> > >
> > > Combining the two parts, the total time complexity of $K$ stacked augmentation layers is
> > > $$
> > > \text{Total Cost}
> > > = O(K \times B \times d \times L) + O\big(B \times (Cd + n)\big)
> > > \approx O(K \times B \times d \times L),
> > > $$
> > > because the second term is much smaller than the first.
> > >
> > > **Memory Complexity**. The memory cost consists of three components.
> > > - The policy generator parameters. The MLP weights have size $O(Cd + n)$,
> > > which does not depend on $L$ and is much smaller than the parameter size of downstream models such as Autoformer, VAE, or TCN.
> > > - The intermediate tensors during augmentation. At each layer the model stores the original sequence and the augmented sequence. The extra memory cost is $O(B \times d \times L)$. Augmentation layers operate sequentially rather than in parallel. Therefore, the total additional memory cost remains $O(B \times d \times L)$, the same as a single layer.
> > > - Downstream model memory, which is shared across all methods and does not affect the relative cost of AutoDA-Timeseries.
> > >
> > > > W3.2: The absence of a detailed parameter analysis (e.g., total parameters introduced by the augmentation generator $A_\theta$ relative to the downstream model) also obscures the model's overall size and potential impact on training stability or overfitting.
> > >
> > > Thank you for raising this important point. We provide a detailed parameter analysis to clarify the relative size of the augmentation generator $A_\theta$ and its potential impact on model complexity, stability, and overfitting.
> > >
> > > The table below summarizes the number of parameters in two representative downstream models (RNN, Autoformer) and the corresponding size of $A_\theta$, along with the relative ratio and observed performance gain:

---

> > > > ### Author Response · Authors · 2025-11-23
> > > > **Author Response (Part 4)**
> > > >
> > > > **Table 1. Parameter analysis of the augmentation generator $A_\theta$.**
> > > > | Dataset | Downstream Model | Model Params | $A_\theta$ Params | Ratio ($A_\theta$ / Model)| Gain |
> > > > |---|---|---|---|---|---|
> > > > | ETTh1 | RNN | 2,475,011 | 1,251,939 | 50.58% | 19.64% |
> > > > | ETTh1 | Autoformer | 11,302,596 | 1,251,939 | 11.08% | 14.92% |
> > > > | ETTm2 | RNN | 2,622,499 | 1,399,395 | 53.95% | 23.75% |
> > > > | ETTm2 | Autoformer | 11,450,052 | 1,399,395 | 12.22% | 8.21% |
> > > >
> > > > Based on the results in the table, we make the following observations:
> > > >
> > > > - **The size of $A_\theta$ remains consistent across different downstream models on the same dataset**. Regardless of whether the downstream model is RNN, or Autoformer, the augmentation generator $A_\theta$ contains approximately 1.25M on ETTh1 and 1.39M on ETTm2. This confirms that $A_\theta$ is designed as a stable module whose size does not depend on the downstream architecture.
> > > > - **The parameter ratio varies with model scale but always stays within a reasonable range**. For large models such as Autoformer, the ratio is only 11-12%. For compact models such as RNN, the ratio appears larger (50.58%) only because the backbone itself is very small, rather than because $A_\theta$ is large.
> > > > - **Adding $A_\theta$ consistently improves performance without introducing training instability or overfitting**. Across all datasets and backbones, AutoDA-Timeseries achieves 8.21%-23.75% performance gains, and no degradation or instability is observed.
> > > >
> > > > > Q1: What is the computational complexity of one forward pass through k augmentation layers relative to sequence length L and dimension d?
> > > >
> > > > Thank you for the question. The computational complexity of one forward pass through $K$ augmentation layers is $O(k \cdot dL)$.
> > > >
> > > > Let $L$ denote the sequence length and $d$ the number of channels (feature dimension). Each augmentation layer contains two lightweight neural modules, the probability generator and the intensity generator. Both modules are implemented as small MLPs whose input dimensionality is $O(d)$. Their hidden size and the number of transforms are constants. Therefore, the computational cost of these components is $O(d)$ per layer, and this cost does not scale with the sequence length $L$.
> > > >
> > > > The dominant cost comes from applying the time-series transformations in the augmentation set $\mathcal{T}$. Most transformations used in AutoDA-Timeseries (e.g., Jittering, Scaling, TimeWarp, Resample) involve either pointwise operations or a single interpolation along the temporal axis. Since each operation is applied over all $L$ time steps and across all $d$ channels, their computational complexity is $O(dL)$ per layer.
> > > >
> > > > Putting these together, the total complexity for $k$ augmentation layers is: $O(k \cdot dL)$.
> > > >
> > > > > Q2: (1) How sensitive are results to the learnable loss weights wz; (2) do they collapse to zero (diversity ignored) for some tasks?
> > > >
> > > > Thank you for the constructive question. We clarify both the sensitivity and stability of the learnable loss weights $w_z^2$.
> > > >
> > > > (1) **Sensitivity analysis**. To study whether performance is sensitive to the choice of $w_2^2$ (intra-layer diversity) and $w_3^2$ (inter-layer diversity), we conducted experiments on both classification (ROCKET) and long-term forecasting (RNN with prediction length 96). The weight $w_1^2$ for the task loss is fixed to 1 following standard practice, so the analysis focuses on the diversity-related weights. We varied $w_2^2$ and $w_3^2$ over a wide range and evaluated several datasets in each task. The results are provided in Tables 2-5 below.
> > > >
> > > > Across all experiments, the performance varies only within a narrow range, and the optimal value does not concentrate at the boundaries. This demonstrates that the learnable loss weights $w_2^2$ and $w_3^2$ do not collapse and that the framework is robust to a wide range of initializations.
> > > >
> > > > (2) No, the learnable weights do not collapse to zero. The formulation in Equation (8) prevents any $w_z^2$ from collapsing to zero. When $w_z^2\to0$, the term $\frac{1}{2w_z^2}L_z$ diverges toward infinity, which produces a strong gradient that pushes $w_z^2$ away from zero. In addition, the second term $\ln(1+w_z^2)$ acts as a log-barrier that mathematically prevents collapse. Therefore, the optimization dynamics ensure that none of the weights $w_z^2$ can converge to zero during training.

---

> > > > > ### Author Response · Authors · 2025-11-23
> > > > > **Author Response (Part 5)**
> > > > >
> > > > > **Table 2. Sensitivity of the intra-layer diversity weight $w_2^2$ on ROCKET classification accuracy.**
> > > > > | Dataset | 0.1 | 0.2 | 0.3 | 0.4 | 0.5 |
> > > > > |-------|---|---|---|---|---|
> > > > > | ArticularyWordRecognition | 0.9667 | **0.9800** | 0.9700 | **0.9800** | 0.9733 |
> > > > > | Epilepsy | 0.9493 | **0.9783** | 0.9638 | 0.9638 | **0.9783** |
> > > > > | RacketSports | 0.8684 | 0.8947 | 0.8947 | 0.8947 | **0.9013** |
> > > > >
> > > > > **Table 3. Sensitivity of the inter-layer diversity weight $w_3^2$ on ROCKET classification accuracy.**
> > > > > | Dataset |	1 | 1.1 | 1.2 | 1.3 | 1.4 |
> > > > > |-------|---|---|---|---|---|
> > > > > | ArticularyWordRecognition | 0.9633 | 0.9733 | **0.9800** | **0.9800** | 0.9733 |
> > > > > | Epilepsy | 0.9565 | 0.9638 | 0.9638 | **0.9783** | 0.9638 |
> > > > > | RacketSports | 0.8553 | 0.8684 | 0.8816 | **0.8947** | 0.8816 |
> > > > >
> > > > > **Table 4. Sensitivity of the intra-layer diversity weight $w_2^2$ on RNN long-term forecasting performance (prediction length = 96).**
> > > > > | Dataset | Metric | 0.1 | 0.2 | 0.3 | 0.4 | 0.5 |
> > > > > |-------|---|---|---|---|---|--|
> > > > > | ETTh1 | MSE | 0.4934 | 0.4879 | **0.4849** | 0.4920 | 0.4894 |
> > > > > || MAE | 0.4780 | 0.4743 | **0.4715** | 0.4772 | 0.4761 |
> > > > > | ETTh2 | MSE | 0.3587 | 0.3423 | **0.3336** | 0.3590 | 0.3697 |
> > > > > || MAE | 0.3890 | **0.3692** | 0.3779 | 0.3793 | 0.3901 |
> > > > > | Exchange | MSE | 0.1213 | 0.1109 | 0.1086 | **0.1077** | 0.1220 |
> > > > > || MAE | 0.2535 | 0.2420 | 0.2328 | **0.2320** | 0.2541 |
> > > > >
> > > > > **Table 5. Sensitivity of the inter-layer diversity weight $w_3^2$ on RNN long-term forecasting performance (prediction length = 96).**
> > > > > | Dataset | Metric | 1 | 1.1 | 1.2 | 1.3 | 1.4 |
> > > > > |-------|---|---|---|---|---|--|
> > > > > | ETTh1 | MSE | 0.5011 | **0.4849** | 0.4919 | 0.4927 | 0.5050 |
> > > > > || MAE | 0.4829 | **0.4715** | 0.4822 | 0.4825 | 0.4854 |
> > > > > | ETTh2 | MSE | 0.3569 | 0.3339 | **0.3336** | 0.3423 | 0.3455 |
> > > > > || MAE | 0.3870 | 0.3780 | **0.3779** | 0.3798 | 0.3821 |
> > > > > | Exchange | MSE | **0.1086** | 0.1115 | 0.1248 | 0.1190 | 0.1153 |
> > > > > || MAE | **0.2328** | 0.2535 | 0.2710 | 0.2654 | 0.2587 |
> > > > >
> > > > >
> > > > >
> > > > >
> > > > > > Q3: What specific time series augmentation transformations were included in the set $\mathcal{T}$? How many transformations ($n$) were used?
> > > > >
> > > > > Thank you for the question.
> > > > >
> > > > > We adopt **7** widely used time-series transformations in $\mathcal{T}$. Specifically, the augmentation set includes: Raw (no augmentation applied), Jittering, Scaling, TimeWarp, Resample, FreqWarp, and MagWarp. We have added detailed descriptions of all transformations in Appendix A.
> > > > >
> > > > >
> > > > > > Q4: How was the number of stacked augmentation layers $K$ determined? The hyperparameter study (Fig 8a) shows sensitivity analysis, but what was the value used in the main experiments, and how was it chosen (e.g., validation set)?
> > > > >
> > > > > Thank you for the question.
> > > > >
> > > > > We conducted experiments on the validation set across multiple datasets with $K = \{1, 2, 3, 4, 5\}$, and selected the hyperparameter value that performs well on most datasets, namely $K = 3$. Therefore, in the main experiments, all datasets use $K = 3$.
> > > > > We have provided a concise description of this selection in Appendix A.
> > > > >
> > > > >
> > > > > > Q5: (1) Was the learnable temperature for Gumbel-Softmax shared across layers, or did each layer have its own? (2) Was a specific annealing schedule used, or was it learned purely via backpropagation?
> > > > >
> > > > > Thank you for the question.
> > > > >
> > > > > (1) Each layer has its own learnable temperature.
> > > > >
> > > > > (2) The temperature is learned purely via backpropagation.
> > > > >
> > > > > We have provided a description in Section 3.5.1 of the manuscript.

---

> > > > > > ### Comment · Reviewer_juTF · 2025-11-24
> > > > > > **Thanks for the rebuttal response**
> > > > > >
> > > > > > Thanks to the authors‘ clarity and extensive experiments. The response solves the questions I have mentioned. This paper provides substantial experiments and results. I'm willing to keep the score.

---

> > > > > > > ### Author Response · Authors · 2025-11-25
> > > > > > > **Appreciation and Request for Feedback**
> > > > > > >
> > > > > > > Dear reviewer juTF,
> > > > > > >
> > > > > > > Thank you very much for taking the time and effort to provide a valuable review of our work and acknowledge that our response has addressed your concerns.
> > > > > > >
> > > > > > > We would greatly appreciate it if you could adjust the review score based on your positive feedback on our rebuttal response, and we are more than willing to engage in further discussion if needed.
> > > > > > >
> > > > > > > Sincerely,
> > > > > > >
> > > > > > > The Authors

---

### Official Review · Reviewer_Mpcs · 2025-10-30

**Soundness:** 2
**Presentation:** 3
**Contribution:** 2
**Rating:** 4
**Confidence:** 3

**Summary:**

This paper proposes AutoDA-Timeseries, a general data augmentation method for time series data. It incorporates time series features into data augmentation policy design and proposes an integrated optimization problem which relates the loss function in the downstream tasks and the augmentation probability and intensity. The proposed data augmentation algorithm is tested in different time series tasks, including time series classification, short-term/long-term forecasting, regression, and anomaly detection.

**Strengths:**

1. The paper is well written and easy to follow.
2. It integrates reinforcement learning based data augmentation methods for different types of time series tasks, including time series classification, regression, forecasting and anomaly detection.
3. Generally good review of data augmentation methods.
4. Experiments in different time series tasks are performed to demonstrate the good performance of the proposed method.

**Weaknesses:**

1. The proposed framework is very similar to data augmentation methods in CV, but with some modifications for time series data. For time series data, the unique part seems [3.3 time series feature extraction], and the rest of the proposed method is very similar to existing methods in CV. The novelty of this paper is limited.
2. Compared with other data augmentation methods for time series data, the efficiency of the proposed method is low.
3. Some experiments are not convincing. The good part of experiments is that diverse time series tasks are tested. The weakness is that the selected downstream models are generally weak and not the SOTA methods. For example, for forecasting the RNN and Autoformer are selected, and the performance of both methods are not as good as SOTA. I suspect the proposed data augmentation method may increase the performance of those suboptimal algorithms. It may not be useful for SOTA methods. I highly recommend the authors test its performance of SOTA methods such as PatchTST.

**Questions:**

Check the weakness part.

---

> ### Author Response · Authors · 2025-11-23
> **Author Response (Part 1)**
>
> We thank Reviewer Mpcs for the positive and constructive feedback. Below we provide our detailed responses to the reviewer’s comments.
>
> > It integrates reinforcement learning based data augmentation methods for different types of time series tasks, including time series classification, regression, forecasting and anomaly detection.
>
> Thank you for your comments.
> The misunderstanding likely arises from the conceptual similarity between automated augmentation frameworks in different domains.
> We would like to clarify that our method is **not** a reinforcement learning (RL)–based automated data augmentation (AutoDA) framework.
> AutoDA-Timeseries is fundamentally different from RL-based or search-based AutoDA approaches:
> our method adopts a fully differentiable, end-to-end optimization mechanism to jointly learn augmentation selection probabilities and augmentation strengths.
>
>
> > W1: The proposed framework is very similar to data augmentation methods in CV, but with some modifications for time series data. For time series data, the unique part seems [3.3 time series feature extraction], and the rest of the proposed method is very similar to existing methods in CV. The novelty of this paper is limited.
>
> Thank you for the comments.
>
> The statement that “the proposed method is very similar to existing methods in CV” does not fully reflect our contributions.
> Existing computer vision (CV) AutoDA methods can be categorized into four categories[1][2]:
> - RL- or search-based augmentation frameworks (e.g., AutoAugment (CVPR’2019)[3], Fast-AA (NeurIPS’2019)[4]);
> - Random or uniform augmentation baselines (e.g., RandAugment (CVPR’2020)[7], UniformAugment (arXiv’2020)[8], TrivialAugment (ICCV’2021)[9]);
> - Gradient-based augmentation methods (e.g., AAA (arXiv’2019)[10], OHL-AA (ICCV’2019)[11]);
> - Ensemble-based weighting approaches (e.g., A2Aug (CVPR’2023)[12]).
>
> None of these methods, either in CV or in the time-series domain, provides a *plug-and-play*, *multi-task-aware*, *differentiable*, and *modality-feature–conditioned* *adaptive* augmentation framework.
>
> In contrast, AutoDA-Timeseries offers several capabilities that are not present in existing AutoDA frameworks:
> - Conditioning on time-series modality features to guide augmentation decisions (see Section 3.3);
> - Multi-layer augmentation with end-to-end joint optimization (see Section 3.4);
> - A plug-and-play design that can be jointly optimized with downstream models and applied uniformly across forecasting, classification, regression, and anomaly detection tasks (see Section 3.5.2).
>
> Therefore, our method is not a simple adaptation of CV AutoDA techniques to time series.
> Instead, it introduces new and effective designs in adaptive augmentation, modality-aware feature conditioning, and multi-task compatibility, which are absent in existing CV augmentation frameworks.
>
> > W2: Compared with other data augmentation methods for time series data, the efficiency of the proposed method is low.
>
> Thank you for the reviewer’s concern regarding efficiency.
> We acknowledge that, when compared with RandomAugment, UniformAugment and TrivialAugment, AutoDA-Timeseries incurs higher training cost.
> However, this difference is not specific to our method; rather, it is an inherent characteristic of adaptive augmentation frameworks. RandAugment, TrivialAugment, and UniformAugment rely solely on uniform or random perturbations, which naturally results in higher efficiency but prevents them from learning adaptive augmentations. As a result, their performance is limited and in some cases even worse than NoAug.
>
> More importantly, within the adaptive augmentation methods, our framework achieves a strong balance between efficiency and performance.
> As shown in Tables 1 and 2, AutoDA-Timeseries is consistently faster than existing adaptive baselines such as AutoTCL and A2Aug while also achieving superior accuracy.
> For example, on the MotorImagery dataset, AutoTCL (1077.6 ms/iter) and A2Aug (5000.0 ms/iter) are substantially slower than our method (155.7 ms/iter).
>
> In summary, AutoDA-Timeseries maintains a reasonable computational cost and achieves a more favorable efficiency–performance trade-off than other adaptive augmentation methods.

---

> > ### Author Response · Authors · 2025-11-23
> > **Author Response (Part 2)**
> >
> > **Table 1. Comparison of training efficiency, memory usage, and accuracy across different augmentation methods on the FingerMovements dataset. “∗.” in the method names denotes ∗Augment.**
> > || Ours | AutoTCL | A2Aug | TS2Vec | InfoTS | Rand. | Uniform. | Trivial. | NoAug |
> > |---|---|---|---|---|---|---|---|---|---|
> > | Training Time (ms/iter) | 79.4 | 206.9 | 144.9 | 128.0 | 61.5 | 45.9 | 75.1 | 35.9 | 24.4 |
> > | Memory (MB) | 55.92 | 114.00 | 62.91 | 9.85 | 9.41 | 6.99 | 6.99 | 6.99 | 6.99 |
> > | Accuracy | **0.62** | 0.55 | 0.57 | 0.54 | 0.6 | 0.57 | 0.57 | 0.53 | 0.59 |
> >
> >
> > **Table 2. Comparison of training efficiency, memory usage, and accuracy across different augmentation methods on the MotorImagery dataset. “∗.” in the method names denotes ∗Augment.**
> > || Ours | AutoTCL | A2Aug | TS2Vec | InfoTS | Rand. | Uniform. | Trivial. | NoAug |
> > |---|---|---|---|---|---|---|---|---|---|
> > | Training Time (ms/iter) | 155.7 | 1077.6 | 5000.0 | 588.0 | 86.0 | 61.3 | 112.6 | 47.5 | 23.7 |
> > | Memory (MB) | 115.00 | 107.02 | 63.09 | 10.85 | 10.41 | 7.01 | 7.01 | 7.01 | 7.01 |
> > | Accuracy | **0.67** | 0.60 | 0.62 | 0.62 | 0.5 | 0.61 | 0.62 | 0.64 | 0.61 |
> >
> >
> > > W3: Some experiments are not convincing. The good part of experiments is that diverse time series tasks are tested. The weakness is that the selected downstream models are generally weak and not the SOTA methods. For example, for forecasting the RNN and Autoformer are selected, and the performance of both methods are not as good as SOTA. I suspect the proposed data augmentation method may increase the performance of those suboptimal algorithms. It may not be useful for SOTA methods. I highly recommend the authors test its performance of SOTA methods such as PatchTST.
> >
> > Thank you for the insightful comments.
> >
> > Following the reviewer’s suggestion, we conducted a comprehensive set of experiments using PatchTST, which is widely regarded as a state-of-the-art backbone for long-term time series forecasting. The evaluation spans 6 datasets and 4 prediction lengths (96/192/336/720), resulting in a large-scale and rigorous assessment.
> >
> > The results consistently show that AutoDA-Timeseries yields improvements across nearly all settings, outperforming both NoAug and RandAugment by a clear margin in terms of MSE and MAE. In particular, AutoDA-Timeseries reduces the average MSE by 6.2% relative to NoAug and by 16.6% relative to RandAugment.
> >
> > The findings indicate that our method is not a “suboptimal-algorithms-only” trick.
> > AutoDA-Timeseries operates independently of backbone design and provides adaptive data augmentation that generalizes across diverse model families.

---

> > > ### Author Response · Authors · 2025-11-23
> > > **Author Response (Part 3)**
> > >
> > > **Table 3. Performance of AutoDA-Timeseries on the PatchTST model across multiple datasets and prediction lengths.**
> > > | Dataset | Pred_len | Metric | NoAug | RandAugment | Ours |
> > > |---|---|---|---|---|---|
> > > | ETTh1 | 96 | MSE | 0.4121 | 0.4991 | **0.4093** |
> > > ||| MAE | 0.4323 | 0.4919 | **0.4225** |
> > > || 192 | MSE | 0.4539 | 0.5059 | **0.4440** |
> > > ||| MAE | 0.4504 | 0.4981 | **0.4498** |
> > > || 336 | MSE | 0.4990 | 0.5570 | **0.4893** |
> > > ||| MAE | 0.4732 | 0.5260 | **0.4656** |
> > > || 720 | MSE | 0.5141 | 0.6504 | **0.4995** |
> > > ||| MAE | 0.4892 | 0.6070 | **0.4718** |
> > > | ETTh2 | 96 | MSE | **0.3098** | 0.3763 | 0.3142 |
> > > ||| MAE | **0.3532** | 0.4174 | 0.3632 |
> > > || 192 | MSE | **0.4090** | 0.4596 | 0.4132 |
> > > ||| MAE | **0.4104** | 0.4709 | 0.4297 |
> > > || 336 | MSE | 0.4409 | 0.5060 | **0.4401** |
> > > ||| MAE | 0.4509 | 0.5087 | **0.4489** |
> > > || 720 | MSE | 0.4598 | 0.5455 | **0.4467** |
> > > ||| MAE | 0.4632 | 0.5382 | **0.4536** |
> > > | ETTm1 | 96 | MSE | 0.3609 | 0.4255 | **0.3590** |
> > > ||| MAE | 0.3938 | 0.4469 | **0.3818** |
> > > || 192 | MSE | 0.4098 | 0.4588 | **0.4013** |
> > > ||| MAE | 0.4090	| 0.4589 | **0.4082** |
> > > || 336 | MSE | 0.4224 | 0.4829 | **0.4146** |
> > > ||| MAE | 0.4195 | 0.4794 | **0.4148** |
> > > || 720 | MSE | 0.4789 | 0.5476 | **0.4712** |
> > > ||| MAE | 0.4503 | 0.5122 | **0.4439** |
> > > | ETTm2 | 96 | MSE | **0.1914** | 0.2468 | 0.1915 |
> > > ||| MAE | **0.2760** | 0.3326 | 0.2765 |
> > > || 192 | MSE | **0.2561** | 0.3088 | 0.2569 |
> > > ||| MAE | **0.3149** | 0.3672 | 0.3159 |
> > > || 336 | MSE | 0.3209 | 0.3729 | **0.3155** |
> > > ||| MAE | 0.3549 | 0.4078 | **0.3527** |
> > > || 720 | MSE | 0.4409 | 0.4663 | **0.4220** |
> > > ||| MAE | 0.4219 | 0.4577 | **0.4178** |
> > > | Exchange | 96 | MSE | 0.0929 | 0.1440 | **0.0890** |
> > > ||| MAE | 0.2115 | 0.2660 | **0.2099** |
> > > || 192 | MSE | 0.1953 | 0.2450 | **0.1824** |
> > > ||| MAE | 0.3124 | 0.3660 | **0.3087** |
> > > || 336 | MSE | 0.5037 | 0.4611 | **0.3796** |
> > > ||| MAE | 0.5262 | 0.5238 | **0.4544** |
> > > || 720 | MSE | 1.3924 | 1.2789 | **1.1069** |
> > > ||| MAE | 0.9180 | 0.9001 | **0.8025** |
> > > | Weather | 96 | MSE | 0.1961 | 0.2637 | **0.1732** |
> > > ||| MAE | 0.2391 | 0.3177 | **0.2187** |
> > > || 192 | MSE | 0.2437 | 0.2994 | **0.2234** |
> > > ||| MAE | 0.2745 | 0.3340 | **0.2629** |
> > > || 336 | MSE | 0.2950 | 0.3477 | **0.2734** |
> > > ||| MAE | 0.3095 | 0.3633 | **0.3043** |
> > > || 720 | MSE | 0.3635 | 0.4282 | **0.3514** |
> > > ||| MAE | 0.3527 | 0.4229 | **0.3497** |
> > > ||| Avg. MSE | 0.4026 | 0.4532 | **0.3778** |
> > > ||| Avg. MAE | 0.4045 | 0.4589 | **0.3928** |
> > >
> > > [1] Mumuni A, Mumuni F. Data augmentation: A comprehensive survey of modern approaches[J]. Array, 2022, 16: 100258.
> > >
> > > [2] Yang Z, Sinnott R O, Bailey J, et al. A survey of automated data augmentation algorithms for deep learning-based image classification tasks[J]. Knowledge and Information Systems, 2023, 65(7): 2805-2861.
> > >
> > > [3] Cubuk E D, Zoph B, Mane D, et al. Autoaugment: Learning augmentation strategies from data[C]//Proceedings of the IEEE/CVF conference on computer vision and pattern recognition. 2019: 113-123.
> > >
> > > [4] Lim S, Kim I, Kim T, et al. Fast autoaugment[J]. Advances in neural information processing systems, 2019, 32.
> > >
> > > [5] Zhang X, Wang Q, Zhang J, et al. Adversarial autoaugment[J]. arXiv preprint arXiv:1912.11188, 2019.
> > >
> > > [6] Lin C, Guo M, Li C, et al. Online hyper-parameter learning for auto-augmentation strategy[C]//Proceedings of the IEEE/CVF international conference on computer vision. 2019: 6579-6588.
> > >
> > > [7] Cubuk E D, Zoph B, Shlens J, et al. Randaugment: Practical automated data augmentation with a reduced search space[C]//Proceedings of the IEEE/CVF conference on computer vision and pattern recognition workshops. 2020: 702-703.
> > >
> > > [8] LingChen T C, Khonsari A, Lashkari A, et al. Uniformaugment: A search-free probabilistic data augmentation approach[J]. arXiv preprint arXiv:2003.14348, 2020.
> > >
> > > [9] Müller S G, Hutter F. Trivialaugment: Tuning-free yet state-of-the-art data augmentation[C]//Proceedings of the IEEE/CVF international conference on computer vision. 2021: 774-782.
> > >
> > > [10] Zhang X, Wang Q, Zhang J, et al. Adversarial autoaugment[J]. arXiv preprint arXiv:1912.11188, 2019.
> > >
> > > [11] Lin C, Guo M, Li C, et al. Online hyper-parameter learning for auto-augmentation strategy[C]//Proceedings of the IEEE/CVF international conference on computer vision. 2019: 6579-6588.
> > >
> > > [12] Li L, Li A. A2-aug: Adaptive automated data augmentation[C]//Proceedings of the IEEE/CVF Conference on Computer Vision and Pattern Recognition. 2023: 2267-2274.

---

> ### Comment · Reviewer_Mpcs · 2025-11-27
> **Thanks for the clarification and the added experiments.**
>
> The new experiments for PatchTST look to me. At least in terms of experiment, this paper is strengthed.

---

> ### Author Response · Authors · 2025-11-28
> **Thank You for Your Feedback and Support**
>
> Dear Reviewer Mpcs,
>
> Thank you for your positive feedback and for adjusting the score. We are glad that the additional experiments addressed your concerns, and we sincerely appreciate your thoughtful review and constructive suggestions.
>
> Sincerely,
>
> The Authors

---

### Official Review · Reviewer_CCnY · 2025-11-01

**Soundness:** 3
**Presentation:** 2
**Contribution:** 2
**Rating:** 4
**Confidence:** 4

**Summary:**

The paper proposes AutoDA-Timeseries, a general-purpose automated data augmentation framework tailored specifically for time series data. Unlike existing approaches that either rely on contrastive pretraining or image-based AutoDA techniques, this framework incorporates time series–specific statistical features to guide augmentation policy design, and it jointly optimizes both the augmentation strategy and downstream task model in a single-stage, end-to-end manner. The method adaptively learns both the selection probability and intensity of each augmentation transformation through stacked augmentation layers, balancing exploration and exploitation during training. Extensive experiments across five major time series tasks—classification, long-term and short-term forecasting, regression, and anomaly detection—demonstrate that AutoDA-Timeseries consistently outperforms state-of-the-art baselines, highlighting its robustness and broad applicability.

**Strengths:**

The experimental results presented in the paper are comprehensive and compelling. The authors evaluate the proposed framework across a wide range of time series tasks—including classification, forecasting, regression, and anomaly detection—and consistently demonstrate improvements over strong baselines. The inclusion of diverse downstream models further highlights the generality and robustness of the method. Overall, the experimental evidence strongly supports the effectiveness and broad applicability of the proposed approach.

**Weaknesses:**

W1.

One key limitation of the paper is that it does not discuss or compare with policy-based augmentation approaches, particularly those that leverage reinforcement learning to adaptively optimize augmentation probability and intensity. This family of methods directly learns augmentation policies through reward signals tied to downstream performance and has been shown to be effective in automatically discovering task-specific augmentation strategies in several domains. Since the proposed framework also aims to learn adaptive augmentation behaviors, a discussion of how it conceptually relates to or differs from RL-based policy optimization would be valuable. Additionally, including comparisons with representative policy-based AutoDA baselines would strengthen the empirical claims and situate
the contribution more clearly within the broader landscape of automated augmentation research.

W2.

Another weakness is the gap between the stated motivation and the technical realization. The paper argues that existing AutoDA frameworks overlook time series–specific characteristics such as temporal dependency and autocorrelation (Line 84 onwards), but it remains unclear how the proposed approach explicitly incorporates these properties into the augmentation optimization process. While the method uses descriptive statistical features to condition the policy generator, the paper does not clearly justify why these features are sufficient to capture the essential temporal structures that differentiate time series from other modalities. As a result, it is difficult to assess whether the framework meaningfully adapts augmentation strategies to time series dynamics, or whether it is primarily applying general augmentation patterns with limited modality-aware refinement. Further clarification or ablation focused on which time series characteristics drive policy decisions would strengthen this claim.

W3.

What is the meaning of $z=1,2,3$ in Eq (8)? Does it imply that $K=3$?

**Questions:**

Please seek Weaknesses above.

---

> ### Author Response · Authors · 2025-11-23
> **Author Response (Part 1)**
>
> We thank Reviewer CCnY for the positive and constructive feedback. Below we provide our detailed responses to the reviewer’s comments.
>
> > W1: One key limitation of the paper is that it does not discuss or compare with policy-based augmentation approaches, particularly those that leverage reinforcement learning to adaptively optimize augmentation probability and intensity. This family of methods directly learns augmentation policies through reward signals tied to downstream performance and has been shown to be effective in automatically discovering task-specific augmentation strategies in several domains. Since the proposed framework also aims to learn adaptive augmentation behaviors, a discussion of how it conceptually relates to or differs from RL-based policy optimization would be valuable. Additionally, including comparisons with representative policy-based AutoDA baselines would strengthen the empirical claims and situate the contribution more clearly within the broader landscape of automated augmentation research.
>
> Thank you for the insightful comments.
>
> We provide a clear discussion of the differences between AutoDA-Timeseries and policy-based automated data augmentation (AutoDA) methods that rely on reinforcement learning (RL).
> - **Different augmentation decision variables: continuous and differentiable vs discrete.** In RL-based AutoDA, augmentation decisions are discrete actions, such as selecting a transform from a categorical set or sampling a magnitude from a discrete range. These decisions cannot be optimized via gradient descent and thus require policy gradient or PPO.
> In contrast, AutoDA-Timeseries parameterizes augmentation probability vectors and strength tensors as continuous and differentiable variables. These parameters directly are jointly optimized with the downstream model.
> - **Different optimization mechanisms: backpropagation vs policy gradient.** RL-based approaches[1][2] update augmentation policies using reward signals tied to downstream performance, which requires sampling trajectories, estimating delayed rewards, and updating policies, often resulting in slow convergence.
> By contrast, AutoDA-Timeseries treats augmentation as part of the model, enabling per-batch, low-variance gradient updates without delayed reward signals or buffer sampling, leading to significantly more stable and efficient training.
>
> To further address the reviewer’s suggestion, we conducted direct comparisons with an RL-based AutoDA method (Auto-DrAC[1]) across 26 classification datasets and 6 long-term forecasting datasets.
> Since Auto-DrAC was originally designed for image data, we first performed a rigorous time-series adaptation to ensure a fair comparison:
> - we replaced its augmentation set with the same time-series transformations used in AutoDA-Timeseries;
> - we used identical downstream models, data splits, and training configurations across both methods;
> - the augmentation selection, strength sampling, and policy-gradient optimization in Auto-DrAC strictly follow the original paper.
>
> The results in Tables 1–2 show that our fully differentiable framework consistently outperforms Auto-DrAC on both tasks. For example, AutoDA-Timeseries achieves 8.3% higher average accuracy in classification, and reduces average forecasting MSE by 29.5% relative to Auto-DrAC. These gains validate that our differentiable optimization strategy is not only conceptually different from RL-based methods but also empirically more effective and more stable.
>
> **Table 1. Comparison with RL-based policy augmentation (Auto-DrAC) on classification datasets.**
>
> | Dataset | Ours | Auto-DrAC | NoAug |
> |--|--|--|--|
> | ArticularyWordRecognition | **0.9800** | 0.9633 | 0.9667 |
> | AtrialFibrillation | **0.4667** | 0.3333 | 0.4000 |
> | BasicMotions | **1** | **1** | **1** |
> | Cricket | **1** | 0.9720 | 0.9583 |
> | DuckDuckGeese | **0.7000** | 0.6600 | 0.6600 |
> | EigenWorms | **0.7328** | 0.6641 | 0.6107 |
> | Epilepsy | **0.9783** | 0.9058 | 0.9638 |
> | ERing | **0.9741** | 0.9333 | 0.9444 |
> | EthanolConcentration | **0.3156** | 0.2586 | 0.2928 |
> | FaceDetection | **0.6328** | 0.6215 | 0.6200 |
> | FingerMovements | **0.6500** | 0.6233 | 0.5900 |
> | HandMovementDirection | **0.5541** | 0.4728 | 0.5270 |
> | Handwriting | 0.3588 | 0.3259 | **0.3600** |
> | Heartbeat | **0.7756** | 0.7463 | 0.7610 |
> | Libras | **0.7167** | 0.6944 | 0.6889 |
> | LSST | 0.5933 | 0.5353 | **0.6006** |
> | MotorImagery | **0.6500** | 0.5500 | 0.5800 |
> | NATOPS | **0.9167** | 0.8611 | **0.9167** |
> | PEMS-SF | **0.5607** | 0.4624 | 0.5376 |
> | PenDigits | **0.9711** | 0.8576 | 0.9634 |
> | PhonemeSpectra | 0.1670 | 0.1670 | **0.1837** |
> | RacketSports | **0.8947** | 0.8421 | 0.8750 |
> | SelfRegulationSCP1 | **0.8840** | 0.8669 | 0.8737 |
> | SelfRegulationSCP2 | **0.6111** | 0.5444 | 0.5500 |
> | StandWalkJump | **0.7333** | 0.5333 | 0.5333 |
> | UWaveGestureLibrary | **0.9313** | 0.9188 | 0.8688 |
> | Average Accuracy | **0.7211** | 0.6659 | 0.6856 |

---

> > ### Author Response · Authors · 2025-11-23
> > **Author Response (Part 2)**
> >
> > **Table 2. Comparison with RL-based policy augmentation (Auto-DrAC) on long-term forecasting datasets.**
> >
> > | Dataset | Pred_len | Metric | Ours | Auto-DrAC | NoAug |
> > |--|--|--|--|--|--|
> > | ETTh1 | 96 | MSE | **0.4849** | 0.6571 | 0.6034 |
> > ||| MAE | **0.4715** | 0.6170 | 0.5234 |
> > || 192 | MSE | **0.5536** | 0.7250 | 0.6314 |
> > ||| MAE | **0.5046** | 0.6496 | 0.5394 |
> > || 336 | MSE | **0.5552** | 0.7257 | 0.5591 |
> > ||| MAE | **0.4902** | 0.6315 | 0.4909 |
> > || 720 | MSE | **0.5777** | 0.7496 | 0.6498 |
> > ||| MAE | **0.5161** | 0.6623 | 0.5550 |
> > | ETTh2 | 96 | MSE | **0.3336** | 0.5136 | 0.4103 |
> > ||| MAE | **0.3779** | 0.5373 | 0.4206 |
> > || 192 | MSE | **0.4229** | 0.5627 | 0.6240 |
> > ||| MAE | **0.4238** | 0.5753 | 0.5461 |
> > || 336 | MSE | **0.4340** | 0.5998 | 0.7327 |
> > ||| MAE | **0.4392** | 0.6031 | 0.5923 |
> > || 720 | MSE | **0.4213** | 0.6063 | 0.7419 |
> > ||| MAE | **0.4431** | 0.6097 | 0.5981 |
> > | ETTm1 | 96 | MSE | **0.4714** | 0.6534 | 0.7152 |
> > ||| MAE | **0.4500** | 0.6106 | 0.5315 |
> > || 192 | MSE | **0.5107** | 0.6952 | 0.7856 |
> > ||| MAE | **0.4630** | 0.6242 | 0.5521 |
> > || 336 | MSE | **0.5628** | 0.7530 | 0.8251 |
> > ||| MAE | **0.4881** | 0.6845 | 0.5704 |
> > || 720 | MSE | **0.6071** | 0.7735 | 0.8740 |
> > ||| MAE | **0.5237** | 0.7113 | 0.5959 |
> > | ETTm2 | 96 | MSE | **0.2019** | 0.3947 | 0.2648 |
> > ||| MAE | **0.2850** | 0.4645 | 0.3390 |
> > || 192 | MSE | **0.2601** | 0.4192 | 0.3512 |
> > ||| MAE | **0.3225** | 0.4714 | 0.3868 |
> > || 336 | MSE | **0.3136** | 0.4455 | 0.4352 |
> > ||| MAE | **0.3545** | 0.4749 | 0.4307 |
> > || 720 | MSE | **0.4197** | 0.5479 | 0.5422 |
> > ||| MAE | **0.4165** | 0.5798 | 0.4829 |
> > | Exchange | 96 | MSE | **0.1086** | 0.2310 | 0.1687 |
> > ||| MAE | **0.2328** | 0.3512 | 0.2995 |
> > || 192 | MSE | **0.2049** | 0.3125 | 0.2726 |
> > ||| MAE | **0.3234** | 0.4308 | 0.3835 |
> > || 336 | MSE | **0.3582** | 0.5059 | 0.4378 |
> > ||| MAE | **0.4360** | 0.5620 | 0.4931 |
> > || 720 | MSE | **0.6920** | 1.0717 | 1.0198 |
> > ||| MAE | **0.6493** | 0.8562 | 0.7766 |
> > Weather | 96 | MSE | **0.1736** | 0.2615 | 0.2561 |
> > ||| MAE | **0.2191** | 0.3080 | 0.2801 |
> > || 192 | MSE | **0.2263** | 0.3252 | 0.3021 |
> > ||| MAE | **0.2636** | 0.3595 | 0.3154 |
> > || 336 | MSE | **0.2761** | 0.3868 | 0.3516 |
> > ||| MAE | **0.3050** | 0.4059 | 0.3464 |
> > || 720 | MSE | **0.3536** | 0.4524 | 0.4254 |
> > ||| MAE | **0.3534** | 0.4452 | 0.3916 |
> > ||| Avg. MSE | **0.3968** | 0.5571 | 0.5408 |
> > ||| Avg. MAE | **0.3930** | 0.5527 | 0.5381 |

---

> > > ### Author Response · Authors · 2025-11-23
> > > **Author Response (Part 3)**
> > >
> > > > W2: Another weakness is the gap between the stated motivation and the technical realization. The paper argues that existing AutoDA frameworks overlook time series–specific characteristics such as temporal dependency and autocorrelation (Line 84 onwards), (1) but it remains unclear how the proposed approach explicitly incorporates these properties into the augmentation optimization process. While the method uses descriptive statistical features to condition the policy generator, (2) the paper does not clearly justify why these features are sufficient to capture the essential temporal structures that differentiate time series from other modalities. As a result, (3) it is difficult to assess whether the framework meaningfully adapts augmentation strategies to time series dynamics, or whether it is primarily applying general augmentation patterns with limited modality-aware refinement. Further clarification or ablation focused on which time series characteristics drive policy decisions would strengthen this claim.
> > >
> > > Thank you for the insightful comments.
> > > We provide clarification and explanation in response to the three key concerns raised by the reviewer.
> > >
> > > (1) AutoDA-Timeseries **directly** incorporates these temporal features into the optimization of both augmentation selection probability and augmentation intensity. Specifically, the extracted time-series features are concatenated with the previous layer’s transformation probabilities (see Figure 2) and then passed through neural modules to generate the current layer’s augmentation probability and strength, as formulated in Equations (4) and (5).
> > > In this way, the temporal features serve as explicit decision-making factors, determining which transformations are amplified and which are suppressed.
> > > Furthermore, our ablation study (Figure 7, Table 9) confirms the effectiveness of this feature-driven design: removing temporal features consistently degrades performance, demonstrating that the augmentation policy indeed leverages time-series characteristics.
> > >
> > > (2) Catch22[3] is one of the most widely adopted lightweight feature sets for time-series data.
> > > Multiple independent studies have recognized its discriminative power and coverage of essential temporal structures:
> > > - Survey (DMKD’2021)[4] describes Catch22 as “a set of 22 highly discriminative and low redundancy features”.
> > > - TFB (VLDB’2024)[5] and TSFM-Bench (SIGKDD’2025)[6] use Catch22 to characterize per-variable temporal behavior and analyze essential properties of multivariate time series (MTS).
> > > - COSCI-GAN (NeurIPS’2022)[7] uses Catch22 to evaluate whether synthetic MTS preserve true temporal structures.
> > >
> > > Thus, Catch22 is broadly recognized as a compact feature set capturing key aspects like autocorrelation, temporal dependency, and distributional statistics.
> > > More importantly, our contribution does not rely on Catch22 itself, but on a modality-feature-driven automated data augmentation mechanism. The feature type can be fully replaced or extended, and this aspect represents an innovation that is orthogonal to existing AutoDA methods.
> > >
> > > (3) Our framework meaningfully adapts augmentation strategies to time series dynamics. To verify this claim, we conducted an ablation study based on the three major categories of Catch22 features. Specifically, different subsets of Catch22 capture key dynamic properties of time series:
> > > - Linear and nonlinear autocorrelation features (e.g., CO_f1ecac, CO_FirstMin_ac) describe temporal dependency;
> > > - Distribution-shape features (e.g., DN_HistogramMode_5, DN_HistogramMode_10) reflect whether the sequence contains sparse anomalies, skewness, or kurtosis;
> > > - Differential-based features (e.g., MD_hrv_classic_pnn40, SB_BinaryStats_diff_longstretch0) capture local fluctuations, short-term variation strength, and long-term stationarity.
> > >
> > > We removed each of these feature categories respectively and evaluated performance across 10 UEA datasets using ROCKET as the downstream model. The results show that removing any category consistently leads to a noticeable performance drop. This indicates that temporal-structure features play an essential role in guiding the learned augmentation policy:
> > > the model does not simply apply general augmentation patterns, but instead relies on time-series-specific dynamic attributes to adjust both augmentation selection probabilities and augmentation strengths, thereby achieving true adaptation to time series dynamics.

---

> > > > ### Author Response · Authors · 2025-11-23
> > > > **Author Response (Part 4)**
> > > >
> > > > **Table 3. Ablation on Catch22 Feature Groups.**
> > > > | Dataset | Ours | Remove Autocorrelation | Remove Distribution | Remove Differencing |
> > > > |--|--|--|--|--|
> > > > | ArticularyWordRecognition | **0.9800** | 0.9400 | 0.9467 | **0.9800** |
> > > > | BasicMotions | **1** | **1** | **1** | **1** |
> > > > | Cricket | **1** | 0.9861 | 0.9444 | **1** |
> > > > | Epilepsy | **0.9783** | 0.9275 | 0.9348 | 0.9203 |
> > > > | ERing | **0.9741** | 0.9185 | 0.9296 | 0.9407 |
> > > > | Heartbeat | **0.7756** | 0.7610 | 0.7415 | 0.7512 |
> > > > | RacketSports | **0.8947** | **0.8947** | 0.8421 | 0.8026 |
> > > > | SelfRegulationSCP1 | **0.8840** | 0.8464 | 0.8294 | 0.8328 |
> > > > | UWaveGestureLibrary | **0.9313** | 0.9031 | 0.9125 | 0.9188 |
> > > > | Avg. Accuracy | **0.9353** | 0.9086 | 0.8979 | 0.9052 |
> > > >
> > > >
> > > >
> > > >
> > > >
> > > > > W3: What is the meaning of $z=1, 2, 3$ in Eq (8)? Does it imply that $K = 3$?
> > > >
> > > > Thank you for the question. The notation $z=1, 2, 3$ in Equation (8) does not indicate that $K = 3$ or that the model uses three stacked augmentation layers. Instead, $z$ indexes the three loss components that constitute the composite loss described in Section 3.5.2.
> > > > $$L_{\text{composite}} = \sum_{z=1, 2, 3}[\frac{1}{2w_z^2}L_z + \ln(1+w_z^2)]$$
> > > > - $L_1$ is the task-specific loss (e.g., MSE for forecasting or cross-entropy for classification),
> > > > - $L_2$ is the intra-layer diversity loss, and
> > > > - $L_3$ is the inter-layer diversity loss.
> > > >
> > > > Thus, $z=1, 2, 3$ enumerates the individual losses that are weighted and summed, rather than specifying the number of augmentation layers.
> > > >
> > > > [1] Raileanu R, Goldstein M, Yarats D, et al. Automatic data augmentation for generalization in reinforcement learning[J]. Advances in Neural Information Processing Systems, 2021, 34: 5402-5415.
> > > >
> > > > [2] Cubuk E D, Zoph B, Mane D, et al. Autoaugment: Learning augmentation strategies from data[C]//Proceedings of the IEEE/CVF conference on computer vision and pattern recognition. 2019: 113-123.
> > > >
> > > > [3] Lubba C H, Sethi S S, Knaute P, et al. catch22: CAnonical Time-series CHaracteristics: Selected through highly comparative time-series analysis[J]. Data mining and knowledge discovery, 2019, 33(6): 1821-1852.
> > > >
> > > > [4] Ruiz A P, Flynn M, Large J, et al. The great multivariate time series classification bake off: a review and experimental evaluation of recent algorithmic advances[J]. Data mining and knowledge discovery, 2021, 35(2): 401-449.
> > > >
> > > > [5] Qiu X, Hu J, Zhou L, et al. TFB: Towards Comprehensive and Fair Benchmarking of Time Series Forecasting Methods. Proc. VLDB Endow. 17, 9 (May 2024), 2363–2377.
> > > >
> > > > [6] Li Z, Qiu X, Chen P, et al. Tsfm-bench: A comprehensive and unified benchmark of foundation models for time series forecasting[C]//Proceedings of the 31st ACM SIGKDD Conference on Knowledge Discovery and Data Mining V. 2. 2025: 5595-5606.
> > > >
> > > > [7] Seyfi A, Rajotte J F, Ng R. Generating multivariate time series with COmmon Source CoordInated GAN (COSCI-GAN)[J]. Advances in neural information processing systems, 2022, 35: 32777-32788.

---

> > > > > ### Author Response · Authors · 2025-11-28
> > > > > **Kindly Request for Feedback of Reviewer**
> > > > >
> > > > > Dear Reviewer CCnY,
> > > > >
> > > > > As the rebuttal deadline is approaching soon, we would like to kindly check whether our responses have addressed your concerns. If there are any points that would benefit from further clarification or additional explanation, we would be more than happy to provide them.
> > > > >
> > > > > We sincerely appreciate your time, insightful comments, and thoughtful review.
> > > > >
> > > > > Sincerely,
> > > > >
> > > > > The Authors

---

### Official Review · Reviewer_JEpj · 2025-11-01

**Soundness:** 3
**Presentation:** 3
**Contribution:** 3
**Rating:** 6
**Confidence:** 4

**Summary:**

This paper presents AutoDA-Timeseries, a unified automated data augmentation framework specifically designed for time series tasks. Unlike traditional contrastive representation learning approaches, which adopt a two-stage pipeline, or existing AutoDA methods that are primarily image-oriented, AutoDA-Timeseries integrates both augmentation probability and intensity optimization into a single-stage, end-to-end framework. The method explicitly incorporates time-series–specific characteristics when designing augmentation policies and dynamically adapts them based on model feedback. Extensive experiments across five representative tasks show that AutoDA-Timeseries consistently outperforms strong baselines and demonstrates robustness across diverse datasets and architectures.

**Strengths:**

1. The paper is comprehensive and well-organized, clearly articulating the motivation for domain-specific augmentation design.

2. The end-to-end adaptive policy optimization framework is conceptually appealing and extends AutoDA beyond computer vision to time series effectively.

3. Experiments cover a wide variety of tasks and datasets, showing consistent and convincing improvements over both random and rule-based baselines.

**Weaknesses:**

1. The literature review on both representation learning and AutoDA relies heavily on older references, more recent developments in time-series adaptive augmentation should be cited.

2. In Related Work, the claim that “the learned representations may not always align well with downstream models” does not directly justify why current augmentation strategies are unsuitable. It requires stronger logical grounding.

3. The proposed framework largely follows vision-based AutoAugment paradigms, but does not sufficiently account for time-series-specific characteristics such as temporal continuity, sequential ordering, and dynamic dependencies. Consequently, some learned augmentations may distort temporal patterns or reduce interpretability.

4. The paper does not discuss sensitivity to the initial augmentation distribution, which is crucial to ensure stability and generalization across datasets.

**Questions:**

See Weaknesses.

---

> ### Author Response · Authors · 2025-11-23
> **Author Response (Part 1)**
>
> We thank Reviewer JEpj for the positive and constructive feedback. Below we provide our detailed responses to the reviewer’s comments.
>
> > W1: The literature review on both representation learning and AutoDA relies heavily on older references, more recent developments in time-series adaptive augmentation should be cited.
>
>
> Thank you for your constructive suggestion. Following the reviewer’s suggestion, we have incorporated recent developments in time-series adaptive augmentation into the revised Section 2 (related work).
>
> InfoTS (AAAI’2023)[1] introduces an information-theoretic meta-learner that automatically selects the most suitable transformations for each sample. AutoTCL (ICLR’2024)[2] proposes a parametric factorization-based augmentation mechanism that learns instance-wise informative masks and generates lossless transformations. AutoCL (CIKM’2024)[3] adaptively adjusts augmentation strength through cross-scale temporal consistency constraints. CAAP (arXiv’2024)[4] learns an adversarial augmentation policy that produces task-aware perturbations guided by contrastive objectives. FreRA (KDD’2025)[5] leverages frequency-domain statistics to adaptively decide augmentation direction and intensity. ReAugment (arXiv’2024)[6] uses a variational masked autoencoder (VMAE) to reconstruct masked raw samples and learn their underlying data distribution, then applies reinforcement learning to adjust the VMAE’s latent variable to generate augmented sequences that preserve the original structure.
>
>
> > W2: In Related Work, the claim that “the learned representations may not always align well with downstream models” does not directly justify why current augmentation strategies are unsuitable. It requires stronger logical grounding.
>
> Thank you for your constructive suggestion. We have provided a clearer explanation below and have incorporated the revised explanation into the Section 2.
> Our revisions are summarized as follows:
>
> Despite their effectiveness, most representation learning frameworks adopt a two-stage pipeline. In the first stage, multiple augmented views of the same time series are generated, and an encoder is trained using contrastive objectives to obtain task-agnostic representations. In the second stage, the pretrained encoder is transferred and adapted to downstream models.
> However, these two stages are decoupled:
> the augmentation strategy and representation learning in Stage 1 are optimized entirely for the contrastive objective and cannot perceive feedback from the downstream model in Stage 2, particularly when the downstream model is not explicitly designed to leverage such representations.
> As a result, the learned representations may not always align well with the objectives or architectures of downstream models, which limits the performance gains in practical scenarios.
>
> > W3: The proposed framework largely follows vision-based AutoAugment paradigms, but does not sufficiently account for time-series-specific characteristics such as temporal continuity, sequential ordering, and dynamic dependencies. Consequently, some learned augmentations may distort temporal patterns or reduce interpretability.
>
> We understand the reviewer’s concern regarding *temporal continuity*, *sequential ordering*, and *dynamic dependencies* in time-series data. Our framework accounts for these time-series-specific characteristics and prevents the learned augmentations from distorting temporal patterns.
>
> First, we incorporate Catch22 features into the policy generator so that the augmentation strategy can directly perceive temporal properties. These features include temporal continuity (e.g., MD_hrv_classic_pnn40, SB_MotifThree_quantile_hh), sequential ordering (e.g., CO_trev_1_num, FC_LocalSimple_mean1_tauresrat), and dynamic dependencies (e.g., CO_f1ecac, CO_FirstMin_ac). By exposing these temporal features to the policy network, the learned augmentation strategies naturally favor transformations that preserve temporal structure, enabling feature-driven time-series augmentation.
>
> Second, the augmentation operators used in our framework already include many transformations that preserve temporal smoothness and local continuity, such as jittering, scaling, and magnitude warping, widely used and empirically validated in the time-series literature to maintain local temporal structure[7][8]. Moreover, because our policy learning is fully end-to-end, gradients from the downstream model flow directly into the augmentation module. As a result, the transformations that substantially disrupt temporal patterns will receive lower selection probabilities or reduced strength during training.

---

> ### Author Response · Authors · 2025-11-23
> **Author Response (Part 2)**
>
> > W4: The paper does not discuss sensitivity to the initial augmentation distribution, which is crucial to ensure stability and generalization across datasets.
>
> Thank you for the insightful comments. Our current implementation initializes the augmentation distribution using a uniform prior with an additional raw-transform bonus. To evaluate the sensitivity of our framework to this initialization, we conduct a controlled study on both classification (ROCKET-based, evaluated with *accuracy*) and long-term forecasting (RNN-based, evaluated with *MSE* and *MAE*) settings under four initial augmentation distributions:
> - Uniform distribution + raw-transform bonus (the default setting in our paper)
> - Pure uniform distribution
> - Random distribution sampled from Dirichlet($\alpha=1$), which centers around the uniform distribution with moderate variance
> - Random distribution sampled from Dirichlet($\alpha=2$), which produces samples closer to uniform but still with variability
>
> As shown in Table 1 and 2 below, the final performance differences remain consistently small across all augmentation distributions. These results demonstrate that our framework is insensitive to the choice of the initial augmentation distribution, and that the learned augmentation policy remains stable and robust regardless of how the distribution is initialized. This indicates that the training dynamics of AutoDA-Timeseries are sufficiently strong to overcome any prior biases introduced at initialization.
>
>
> **Table 1. Sensitivity of AutoDA-Timeseries to initial augmentation distributions on classification.**
> | Dataset                  | Uniform + Raw Bonus (Ours) | Uniform Dist. | Random Dist. ($\alpha=1$) | Random Dist. ($\alpha=2$) |
> |--------------------------|-----------------------------|----------------|--------------------------|--------------------------|
> | ArticularyWordRecognition | **0.9800**                      | 0.9500         | 0.9733                   | 0.9633                   |
> | BasicMotions             | **1.0000**                      | **1.0000**         | **1.0000**                   | **1.0000**                   |
> | Cricket                  | **1.0000**                      | 0.9861         | 0.9861                   | **1.0000**                   |
> | Epilepsy                 | **0.9783**                      | 0.9638         | 0.9420                   | 0.9638                   |
> | ERing                    | **0.9741**                      | 0.9222         | 0.9556                   | 0.9296                   |
> | Heartbeat                | **0.7756**                      | **0.7756**         | 0.7415                   | 0.7366                   |
> | RacketSports             | 0.8947                      | 0.8817         | 0.8882                   | **0.9013**                   |
> | SelfRegulationSCP1       | 0.8840                      | 0.8703         | **0.8980**                   | 0.8805                   |
> | UWaveGestureLibrary      | 0.9313                      | 0.9188         | 0.9270                   | **0.9345**                   |
> | Avg                  | **0.9353**                  | 0.9187     | 0.9235               | 0.9233               |

---

> > ### Author Response · Authors · 2025-11-23
> > **Author Response (Part 3)**
> >
> > **Table 2. Sensitivity of AutoDA-Timeseries to initial augmentation distributions on long-term forecasting.**
> > |Dataset | Pred_Len | Metric | Uniform + Raw Bonus (Ours) | Uniform Dist. | Random Dist. ($\alpha=1$) | Random Dist. ($\alpha=2$) |
> > |---------|---------|--------|------------------------|----------|--------------|--------------|
> > |ETTh1| 96  | MSE | **0.4849** | 0.5034 | 0.5098 | 0.5137 |
> > ||| MAE | **0.4715** | 0.4789 | 0.4795 | 0.4803 |
> > || 192 | MSE | 0.5536 | **0.5308** | 0.5387 | 0.5420 |
> > ||| MAE | 0.5046 | **0.4912** | 0.4951 | 0.4956 |
> > || 336 | MSE | **0.5552** | 0.5658 | 0.5728 | 0.5762 |
> > ||| MAE | **0.4902** | 0.5027 | 0.5065 | 0.5069 |
> > || 720 | MSE | 0.5777 | **0.5536** | 0.5692 | 0.5734 |
> > ||| MAE | 0.5161 | **0.5003** | 0.5135 | 0.5139 |
> > |ETTh2| 96  | MSE | **0.3336** | 0.3590 | 0.3559 | 0.3559 |
> > ||| MAE | **0.3779** | 0.3918 | 0.3914 | 0.3914 |
> > || 192 | MSE | **0.4229** | 0.4250 | 0.4273 | 0.4273 |
> > ||| MAE | **0.4238** | 0.4277 | 0.4320 | 0.4321 |
> > || 336 | MSE | **0.4340** | 0.4453 | 0.4597 | 0.4593 |
> > ||| MAE | **0.4392** | 0.4503 | 0.4582 | 0.4582 |
> > || 720 | MSE | **0.4213** | 0.4472 | 0.4546 | 0.4498 |
> > ||| MAE | **0.4431** | 0.4590 | 0.4591 | 0.4505 |
> > |ETTm1| 96  | MSE | **0.4714** | 0.4787 | 0.4761 | 0.4763 |
> > ||| MAE | **0.4500** | 0.4581 | 0.4556 | 0.4539 |
> > || 192 | MSE | 0.5107 | **0.5077** | 0.5081 | 0.5280 |
> > ||| MAE | 0.4630 | **0.4601** | 0.4735 | 0.4691 |
> > || 336 | MSE | 0.5628 | **0.5612** | 0.5615 | 0.6471 |
> > ||| MAE | 0.4881 | **0.4851** | 0.4897 | 0.5084 |
> > || 720 | MSE | 0.6071 | 0.6132 | **0.6023** | 0.6459 |
> > ||| MAE | 0.5237 | 0.5305 | **0.5188** | 0.5584 |
> > |ETTm2| 96  | MSE | **0.2019** | 0.2108 | 0.2156 | 0.2156 |
> > ||| MAE | 0.2850 | **0.2842** | 0.2896 | 0.2856 |
> > || 192 | MSE | **0.2601** | 0.2875 | 0.2827 | 0.2826 |
> > ||| MAE | **0.3225** | 0.3296 | 0.3265 | 0.3265 |
> > || 336 | MSE | **0.3136** | 0.3611 | 0.3480 | 0.3485 |
> > ||| MAE | **0.3545** | 0.3727 | 0.3635 | 0.3639 |
> > || 720 | MSE | **0.4197** | 0.4582 | 0.4328 | 0.4517 |
> > ||| MAE | **0.4165** | 0.4388 | 0.4358 | 0.4368 |
> > | Avg || MSE | **0.4369** | 0.4415 | 0.4435 | 0.4463 |
> > ||| MAE | **0.4356** | 0.4413 | 0.4430 | 0.4457 |
> >
> >
> >
> >
> >
> >
> >
> >
> >
> >
> >
> >
> >
> >
> >
> >
> >
> >
> >
> >
> >
> >
> > [1] Luo D, Cheng W, Wang Y, et al. Time series contrastive learning with information-aware augmentations[C]//Proceedings of the AAAI Conference on Artificial Intelligence. 2023, 37(4): 4534-4542.
> >
> > [2] Zheng X, Wang T, Cheng W, et al. Parametric augmentation for time series contrastive learning[J]. arXiv preprint arXiv:2402.10434, 2024.
> >
> > [3] Jing B, Wang Y, Sui G, et al. Automated contrastive learning strategy search for time series[C]//Proceedings of the 33rd ACM International Conference on Information and Knowledge Management. 2024: 4612-4620.
> >
> > [4] Chang T Y, Dai H, Tseng V S. CAAP: Class-Dependent Automatic Data Augmentation Based On Adaptive Policies For Time Series[J]. arXiv preprint arXiv:2404.00898, 2024.
> >
> > [5] Tian T, Miao C, Qian H. FreRA: a frequency-refined augmentation for contrastive learning on time series classification[C]//Proceedings of the 31st ACM SIGKDD Conference on Knowledge Discovery and Data Mining V. 2. 2025: 2835-2846.
> >
> > [6] Yuan H, Wang Y, Chen Y, et al. ReAugment: Model Zoo-Guided RL for Few-Shot Time Series Augmentation and Forecasting[J]. arXiv preprint arXiv:2409.06282, 2024.
> >
> > [7] Iwana B K, Uchida S. An empirical survey of data augmentation for time series classification with neural networks[J]. Plos one, 2021, 16(7): e0254841.
> >
> > [8] Iglesias G, Talavera E, González-Prieto Á, et al. Data augmentation techniques in time series domain: a survey and taxonomy[J]. Neural Computing and Applications, 2023, 35(14): 10123-10145.

---

> > > ### Author Response · Authors · 2025-11-28
> > > **Kindly Request for Feedback of Reviewer**
> > >
> > > Dear Reviewer JEpj,
> > >
> > > As the rebuttal deadline is approaching soon, we would like to kindly check whether our responses have addressed your concerns. If there are any points that would benefit from further clarification or additional explanation, we would be more than happy to provide them.
> > >
> > > We sincerely appreciate your time, insightful comments, and thoughtful review.
> > >
> > > Sincerely,
> > >
> > > The Authors

---

### Author Response · Authors · 2025-12-04
**The Summary of Rebuttal**

Dear Reviewers, ACs, SACs, and PCs,

Thank you for the thoughtful evaluations and constructive discussion during the rebuttal phase, and we appreciate the AC taking the time to review our submission. We would like to provide a brief summary to assist the AC in understanding how our rebuttal and discussion addressed the reviewer's concerns.

We summarize the score changes observed during the rebuttal period (with supporting evidence available in [the anonymized link](https://anonymous.4open.science/r/AutoDA-Timeseries-R3-Score-1552/README.md)):

| Reviewer | Score Change | Follow-up Comment |
|--|--|--|
| JEpj | 6 $\to$ 6 | No follow-up comment due to platform issue |
| CCnY | 4 $\to$ 4 | No follow-up comment due to platform issue |
| Mpcs | 4 $\to$ 6 | States that the new experiments strengthened the paper |
| juTF | 6 $\to$ 6 | All concerns were resolved and is willing to keep the score |
| Average Score | 5 $\to$ 5.5 |

Reviewers consistently highlighted the following strengths of AutoDA-Timeseries:

- **Clear and well-motivated problem formulation**, identifying the limitations of contrastive learning and CV-based AutoDA for time series.
- **A general-purpose, plug-and-play augmentation framework** applicable across five major time-series tasks.
- **Modality-aware policy generation**, incorporating time-series features to guide augmentation probability and intensity.
- **End-to-end, single-stage optimization**, jointly learning augmentation strategies and downstream models.
- **Strong empirical evidence**, with consistent improvements across numerous datasets and diverse downstream architectures.
- **Scalable and modular design**, demonstrating compatibility with many backbone models and task settings.

During the rebuttal phase, we managed to address the reviewers' concerns through new experiments, theoretical analysis, and comprehensive manuscript revisions.

>Related work expansion, baseline adaptation, and new experiments (JEpj: W1/W2, juTF: W2.1/W2.2, CCnY: W1)

We expanded Section 2 to include recent time-series adaptive augmentation work, clarified the limitations of two-stage contrastive frameworks, and added complete baseline-adaptation details in Appendix B. To support the comparison with RL-based AutoDA, we additionally implemented a carefully adapted RL baseline and evaluated it on 26 classification and 6 forecasting datasets.
>Methodology clarification (JEpj: W3, CCnY: W2/W3, juTF: W1.1/W1.2/W2.3/Q5)

We clarified the methodological components of AutoDA-Timeseries by (i) explaining how Catch22 features capture key temporal properties and are directly used by the policy generator, supported by ablations showing their contribution in Appendix C; (ii) specifying that feature-preserving operators and end-to-end gradient feedback suppress distortive augmentations; (iii) fully detailing the composite loss, correcting ambiguous notations in Section 3.5.
>Sensitivity analysis (JEpj: W4, juTF: Q2)

We conducted controlled experiments on both the learnable loss weights and the initial augmentation distributions. The results (Appendix D) show that performance varies only minimally across four different initialization schemes and across a wide range of weight settings. We further provided analytical derivations demonstrating that the optimization dynamics of the learnable weights remain stable and cannot collapse during training.
>Efficiency, computational complexity, and detailed parameters (Mpcs: W2; juTF: W3.1/W3.2/Q1/Q3/Q4)

We provided a complete analysis of efficiency and computational costs, including full time-memory complexity derivations (Appendix F) and a parameter analysis showing that the augmentation generator introduces only a small and stable overhead. We also added detailed descriptions of the augmentation set $\mathcal{T}$ and the selection of $K$ via validation (Appendix A). Experiments further confirmed that while simple non-adaptive baselines such as RandAugment and UniformAugment are faster, our method achieves a substantially better efficiency-performance trade-off and is faster than adaptive methods like AutoTCL and A2Aug while delivering higher accuracy (Figure 9).
>Novelty clarification and validation on stronger downstream models (Mpcs: W1/W3)

We clarified the novelty of AutoDA-Timeseries by reviewing the four major categories of AutoDA methods in CV and explaining that our framework is fundamentally different: it is a plug-and-play, multi-task-aware, fully differentiable, and conditioned on modality-specific features. Additionally, following the reviewer's suggestion, we conducted new experiments on PatchTST. Across six forecasting datasets, AutoDA-Timeseries consistently improved performance over both NoAug and RandAugment, demonstrating that the gains extend to state-of-the-art downstream models.

We sincerely thank the AC for considering our work and remain willing to provide any further clarification if needed.

Best regards,

Authors of AutoDA-Timeseries

---

### Meta-Review · Area_Chair_5fiE · 2026-01-06

**Summary:**

This paper proposes AutoDA-Timeseries, an automated data augmentation framework specifically designed for time series data. The work is well motivated, identifying clear limitations of both contrastive pretraining–based augmentation and vision-oriented AutoDA methods when applied to time series. By incorporating time-series–specific features into augmentation policy design and jointly optimizing augmentation probability, intensity, and downstream objectives in a single-stage, end-to-end manner, the paper presents a general-purpose and plug-and-play augmentation framework applicable across multiple tasks.

Reviewers consistently acknowledged the breadth and strength of the empirical evaluation, which covers classification, short- and long-term forecasting, regression, and anomaly detection across a large number of datasets and backbone models. The rebuttal was particularly strong: the authors carefully addressed concerns about novelty, efficiency, stability, and applicability to stronger downstream models. Notably, they added extensive comparisons with RL-based AutoDA methods, sensitivity analyses on initialization, efficiency benchmarks, and new experiments on PatchTST, demonstrating that the method is not limited to suboptimal backbones.

At the same time, reviewers noted that parts of the framework are incremental extensions of existing AutoDA paradigms, with novelty primarily stemming from modality-aware conditioning and end-to-end differentiability rather than a fundamentally new theoretical formulation. While the gains are consistent and practically meaningful, the contribution is more engineering-driven than conceptually transformative. Overall, the paper represents a solid, carefully executed, and practically useful contribution that fits well within the ICLR community, and is best suited for acceptance.

**Reviewer Concerns:**

Reviewers initially raised concerns regarding novelty relative to existing CV-based AutoDA methods, efficiency overhead, sensitivity to initialization, and applicability to strong downstream models. These concerns were largely addressed in the rebuttal through additional experiments, ablations, efficiency analyses, and new evaluations on PatchTST. Remaining issues are relatively minor and mostly relate to presentation clarity and positioning.

**Reviewer Scores:**

Reviewer scores were initially mixed but trended positive. During the rebuttal phase, one reviewer explicitly increased their score, citing that the new experiments and clarifications substantially strengthened the paper, while other reviewers maintained their scores but acknowledged that their main concerns had been addressed. Based on the final score distribution, the quality of the rebuttal, and the Area Chair’s assessment, the paper meets the acceptance bar.

---

### Decision · Program_Chairs · 2026-01-26

Accept (Poster)